# Optimal Extragradient-Based Algorithms for Stochastic Variational Inequalities with Separable Structure

**Huizhuo Yuan**$^\diamond$    **Chris Junchi Li**$^\dagger$    **Gauthier Gidel**$^\ddagger$
**Michael I. Jordan**$^{\dagger,\square}$    **Quanquan Gu**$^\diamond$    **Simon S. Du**$^\star$

$^\diamond$ Department of Computer Science, University of California, Los Angeles
`{hzyuan, qgu}@cs.ucla.edu`
$^\dagger$ Department of Electrical Engineering and Computer Sciences, University of California, Berkeley
`{junchili, jordan}@cs.berkeley.edu`
$^\ddagger$ DIRO, Université de Montréal and Mila
`gauthier.gidel@umontreal.ca`
$^\square$ Department of Statistics, University of California, Berkeley
$^\star$ Paul G. Allen School of Computer Science and Engineering, University of Washington
`ssdu@cs.washington.edu`

## Abstract

We consider the problem of solving stochastic monotone variational inequalities with a separable structure using a stochastic first-order oracle. Building on standard extragradient for variational inequalities we propose a novel algorithm—stochastic *accelerated gradient-extragradient* (AG-EG)—for strongly monotone variational inequalities (VIs). Our approach combines the strengths of extragradient and Nesterov acceleration. By showing that its iterates remain in a bounded domain and applying scheduled restarting, we prove that AG-EG has an optimal convergence rate for strongly monotone VIs. Furthermore, when specializing to the particular case of bilinearly coupled strongly-convex-strongly-concave saddle-point problems, including bilinear games, our algorithm achieves fine-grained convergence rates that match the respective lower bounds, with the stochasticity being characterized by an additive statistical error term that is optimal up to a constant prefactor.

## 1   Introduction

The variational inequality (VI) problem plays a central role in a wide range of optimization problems with convex structure, including convex minimization, saddle-point problems, and games [Facchinei and Pang, 2003, Nemirovski, 2004, Nemirovski et al., 2009, Juditsky et al., 2011, Jordan et al., 2023]. A general VI problem aims to find a solution $\boldsymbol{z}^* \in \mathcal{Z}$ that satisfies:

$$\langle \mathcal{W}(\boldsymbol{z}^*), \boldsymbol{z}^* - \boldsymbol{z}\rangle \leq 0, \quad \forall \boldsymbol{z} \in \mathcal{Z}, \tag{1}$$

where $\mathcal{Z}$ is a finite-dimensional closed and convex feasible set and $\mathcal{W}(\cdot)$ is a monotone operator in the following form:

$$\mathcal{W}(\boldsymbol{z}) = \nabla\mathcal{F}(\boldsymbol{z}) + \mathcal{H}(\boldsymbol{z}) + J'(\boldsymbol{z}) \equiv \mathbb{E}_\xi[\nabla\widetilde{\mathcal{F}}(\boldsymbol{z};\xi)] + \mathbb{E}_\zeta[\widetilde{\mathcal{H}}(\boldsymbol{z};\zeta)] + J'(\boldsymbol{z}), \tag{2}$$

where $\mathcal{F}$ is continuously differentiable with $L$-Lipschitz continuous gradient and is $\mu$-strongly convex, $\mathcal{H}$ is an $M$-Lipschitz monotone operator, $J' \in \partial J$ is the subgradient of a simple and convex function, $\xi$ and $\zeta$ are drawn from distributions $\mathcal{D}_\xi$ and $\mathcal{D}_\zeta$, respectively. This formulation captures a separable structure in which $\mathcal{H}$ usually models the competing forces in a system, and $J$ models

a nonsmooth factor. In addition, we consider the stochastic setting where we can only access $\nabla\mathcal{F}$ and $\mathcal{H}$ through their unbiased estimators $\nabla\widetilde{\mathcal{F}}(\boldsymbol{z};\xi)$ and $\widetilde{\mathcal{H}}(\boldsymbol{z};\zeta)$ respectively.

A notable instance of the VI problem (1) with separable structure (2) is the widely studied *bilinearly coupled strongly-convex-strongly-concave saddle-point problem*:

$$\min_{\boldsymbol{x}\in\mathbb{R}^n}\max_{\boldsymbol{y}\in\mathbb{R}^m}\mathscr{F}(\boldsymbol{x},\boldsymbol{y})=F(\boldsymbol{x})+H(\boldsymbol{x},\boldsymbol{y})-G(\boldsymbol{y})\equiv\mathbb{E}_\xi\left[f(\boldsymbol{x};\xi)\right]+\mathbb{E}_\zeta\left[h(\boldsymbol{x},\boldsymbol{y};\zeta)\right]-\mathbb{E}_\xi\left[g(\boldsymbol{y};\xi)\right],$$

(3)

where $H(\boldsymbol{x},\boldsymbol{y})\equiv\boldsymbol{x}^\top\mathbf{B}\boldsymbol{y}-\boldsymbol{x}^\top\mathbf{u}_{\boldsymbol{x}}+\mathbf{u}_{\boldsymbol{y}}^\top\boldsymbol{y}$ is the bilinear coupling function with the coupling matrix $\mathbf{B}\in\mathbb{R}^{n\times m}$. Note that (3) is a special instance of (1) when taking $\mathcal{F}(\boldsymbol{z})=F(\boldsymbol{x})+G(\boldsymbol{y})$, $\mathcal{H}(\boldsymbol{z})=[\nabla_{\boldsymbol{x}}H(\boldsymbol{x},\boldsymbol{y});-\nabla_{\boldsymbol{y}}H(\boldsymbol{x},\boldsymbol{y})]$ and $J=0$. In addition to a wide range of applications in economics, problems of form (3) are becoming increasingly important in machine learning. For instance, (3) appears in reinforcement learning, differentiable games, regularized empirical risk minimization, and robust optimization formulations. It can also be seen as a local approximation of the nonconvex-nonconcave minimax games—e.g., the generative adversarial network (GAN) [Goodfellow et al., 2020]—around a local Nash equilibrium [Mescheder et al., 2017, Nagarajan and Kolter, 2017].

In this paper, we aim to improve the efficiency of solving (1) by utilizing the structural information of the monotone operator in (2). More specifically, we consider the case when $\mathcal{F}$ is strongly monotone, or zero. Although optimal convergence results have been obtained for the monotone VI problem (1) [Chen et al., 2017] as well as the special case of convex-concave saddle-point problem with bilinear coupling (3) [Chen et al., 2014], it remains open how to design an optimal algorithm for the strongly monotone VI problem. Notably, for the special case (3) when $F$ and/or $G$ are strongly convex, several concurrent works have independently obtained the optimal convergence rates [Kovalev et al., 2022, Thekumparampil et al., 2022, Jin et al., 2022, Metelev et al., 2022, Li et al., 2022b]. On the other hand, when both $F$ and $G$ are zero, optimal convergence results have been obtained by Li et al. [2022a] and the accelerated-gradient optimistic gradient approach [Li et al., 2022b]. We defer a more complete overview of related work to the appendix.

## 1.1 Main Contributions

We start with the strongly monotone VI problem in an unbounded feasible set, extending the scope of recent work such as Jin et al. [2022] and going beyond earlier studies that focus on nonstrongly monotone VIs in a bounded feasible set [Juditsky et al., 2011, Chen et al., 2017].[1] We propose a class of algorithms named stochastic *accelerated gradient-extragradient* (AG-EG), which combine Nesterov acceleration with the extragradient method. By employing either a strong-convexity shifting technique or a scheduled restarting scheme, our algorithm achieves convergence rates that match the lower bounds for the general strongly monotone VI problem (1), the special SC-SC saddle-point problem (3), and bilinear games, in both deterministic and stochastic settings, thus providing a unified optimal solution. In sharp contrast to the accelerated mirror-prox (AMP) algorithm proposed by Chen et al. [2017], Jordan et al. [2023], our analysis does not rely on the boundedness of the feasible set $\mathcal{Z}$, which makes our algorithm projection-free. We also extend our algorithm to VIs with bounded feasible set and/or nondifferentiable convex regularization through proximal mapping. We summarize our contributions as follows:

(1) We present a direct approach for separable strongly monotone VIs, where the iteration complexity lower bound due to Zhang et al. [2022] is matched as $\mathcal{O}\left(\sqrt{\frac{L}{\mu}}+\frac{M}{\mu}+\frac{\sigma^2}{\mu^2\varepsilon^2}\right)\log\left(\frac{L}{\mu}\frac{1}{\varepsilon}\right)$, which admits a sharp near-unity coefficient [§2.3, Theorem 2.3]. Here $\sigma^2$ is the weighted, uniform variance bound on the stochastic gradient and stochastic operator.

(2) We also present a stochastic AG-EG algorithm equipped with scheduled restarting, which achieves the sharpest possible iteration complexity of $\mathcal{O}\left(\left(\sqrt{\frac{L}{\mu}}+\frac{M}{\mu}\right)\log\left(\frac{1}{\varepsilon}\right)+\frac{\sigma^2}{\mu^2\varepsilon^2}\right)$ for finding an $\varepsilon$-optimal point. The deterministic part matches the complexity lower bound in Zhang et al. [2022], while the stochastic part matches the optimal statistical error.

---

[1]VIs in an unbounded feasible set is more difficult to solve because existing algorithms and analyses crucially rely on the boundedness of the feasible set.

When specializing the VI problem to bilinearly coupled SC-SC saddle-point problems, our results have the following implications:

**Strongly-convex-strongly-concave (SC-SC) Saddle-Point Problem.** For the class of SC-SC saddle-point problems, the stochastic AG-EG descent-ascent Algorithm 1, equipped with scaling reduction, achieves an iteration complexity of

$$\mathcal{O}\left(\left(\sqrt{\frac{L_F}{\mu_F} \vee \frac{L_G}{\mu_G}} + \sqrt{\frac{\lambda_{\max}(\mathbf{B}^\top \mathbf{B})}{\mu_F \mu_G}}\right) \log\left(\frac{1}{\varepsilon}\right) + \frac{\sigma^2}{\mu_F^2 \varepsilon^2}\right), \tag{4}$$

where $F : \mathbb{R}^n \to \mathbb{R}$ is $L_F$-smooth and $\mu_F$-strongly convex, $G : \mathbb{R}^m \to \mathbb{R}$ is $L_G$-smooth and $\mu_G$-strongly convex. When the optimization problem is deterministic, the complexity upper bound matches the lower bound of Zhang et al. [2022][§3.1, Corollary 2.8].

**Bilinear Games.** For bilinear games ($\nabla f(\boldsymbol{x}; \xi) = \mathbf{0}$ and $\nabla g(\boldsymbol{y}; \xi) = \mathbf{0}$ almost surely), Algorithm 1, equipped with scheduled restarting achieves an iteration complexity of

$$\mathcal{O}\left(\sqrt{\frac{\lambda_{\max}(\mathbf{B}^\top \mathbf{B})}{\lambda_{\min}(\mathbf{B}\mathbf{B}^\top)}} \log\left(\frac{\sqrt[4]{\lambda_{\min}(\mathbf{B}\mathbf{B}^\top)\lambda_{\max}(\mathbf{B}^\top\mathbf{B})}}{\sigma_{\mathrm{Bil}}}\right) + \frac{\sigma_{\mathrm{Bil}}^2}{\lambda_{\min}(\mathbf{B}\mathbf{B}^\top)\varepsilon^2}\right), \tag{5}$$

where $\sigma_{\mathrm{Bil}}^2$ is the variance of the stochastic gradient on the bilinear coupling term. When there is no randomness, this complexity result reduces to $\mathcal{O}\left(\sqrt{\frac{\lambda_{\max}(\mathbf{B}^\top\mathbf{B})}{\lambda_{\min}(\mathbf{B}\mathbf{B}^\top)}} \log\left(\frac{1}{\varepsilon}\right)\right)$ for bilinear games, matching the lower bound of Ibrahim et al. [2020] [§3.2, Corollary 3.3].[2]

**Organization.** The rest of this paper is organized as follows. Section 2 proposes the Accelerated Gradient-Extragradient Descent-Ascent algorithm for strongly monotone VIs, showing that it achieves an accelerated convergence rate, and extending to VIs with bounded domains with proximal operator. Section 3 discusses two specific instances of saddle-point problems, where our proposed AG-EG algorithm has a convergence rate that matches the corresponding lower bounds. Finally, Section 4 summarizes our results and suggests future directions.

**Notation.** Let $\lambda_{\max}(\mathbf{M})$ (resp. $\lambda_{\min}(\mathbf{M})$ be the largest (resp. smallest) eigenvalue of a real symmetric matrix $\mathbf{M}$. Let $a \vee b \equiv \max(a, b)$ (resp. $a \wedge b \equiv \min(a, b)$) denote the maximum (resp. minimum) value of two reals $a, b$. For two nonnegative real sequences $(a_n)$ and $(b_n)$, we write $a_n = \mathcal{O}(b_n)$ or $a_n \lesssim b_n$ (resp. $a_n = \Omega(b_n)$ or $a_n \gtrsim b_n$) to denote $a_n \leq Cb_n$ (resp. $a_n \geq Cb_n$) for all $n \geq 1$ for a positive, numerical constant $C$, and let $a_n \asymp b_n$ if both $a_n \lesssim b_n$ and $a_n \gtrsim b_n$ hold. We also let $a_n = \tilde{\mathcal{O}}(b_n)$ denote $a_n \leq Cb_n$ where $C$ hides a polylogarithmic factor in problem-dependent constants. We let $[\boldsymbol{x}; \boldsymbol{y}] \in \mathbb{R}^{n+m}$ concatenate two vectors $\boldsymbol{x} \in \mathbb{R}^n$ and $\boldsymbol{y} \in \mathbb{R}^m$. Finally for two real symmetric matrices $\mathbf{A}$ and $\mathbf{B}$, we denote $\mathbf{A} \preceq \mathbf{B}$ (resp. $\mathbf{A} \succeq \mathbf{B}$) when $\mathbf{v}^\top(\mathbf{A} - \mathbf{B})\mathbf{v} \leq 0$ (resp. $\mathbf{v}^\top(\mathbf{A} - \mathbf{B})\mathbf{v} \geq 0$) holds for all vectors $\mathbf{v}$.

## 2 Accelerated Gradient-Extragradient Descent-Ascent Algorithm

In this section, we focus on accelerating the extragradient algorithm for the strongly monotone VI problem in (1) with separable structure (2). Our algorithm design draws inspiration from the work of Chen et al. [2017] on the stochastic Accelerated MirrorProx (AMP) algorithm for nonstrongly monotone VIs. The AMP algorithm applies Nesterov-type acceleration on top of the mirror-prox method [Korpelevich, 1976, Nemirovski, 2004] and attains the optimal iteration complexity of $\mathcal{O}\left(\sqrt{\frac{L}{\varepsilon}} + \frac{M}{\varepsilon}\right)$. However, the big-O notation hides the diameter of the feasible set, and the existing theory for the AMP algorithm can only deal with VIs with bounded domain. Our algorithm not only achieves the optimal convergence rates for the strongly monotone VI problem with separable structure but we also remove the dependency on the diameter of the feasible set. Therefore, our algorithm can deal with VIs with unbounded domains.

---

[2]For the function class of bilinear games, we assume that $n = m$ where $\mathbf{B}$ is a nonsingular square matrix, so that $\lambda_{\min}(\mathbf{B}\mathbf{B}^\top) > 0$ and the complexity makes sense. See §3.2 for more on this.

Throughout §2, we maintain conceptual simplicity by presenting all our algorithm designs in the deterministic setting, while presenting the convergence results in the more general stochastic setting. These results can be easily reduced to the deterministic setting when the stochastic noise vanishes.

## 2.1 Setting and Assumptions

In this section, we formally introduce our assumptions. We first state the smoothness and monotonicity assumptions that we impose on $\mathcal{F}$ and $\mathcal{H}$.

**Assumption 2.1 (Monotonicity, strong convexity and smoothness)** *We assume that function $\mathcal{F}(\cdot)$ is continuously differentiable with $L$-Lipschitz continuous gradient and is $\mu$-strongly convex. That is, for any $z, z' \in \mathcal{Z}$,*

$$\tfrac{\mu}{2}\|z - z'\|^2 \leq \mathcal{F}(z) - \mathcal{F}(z') - \nabla\mathcal{F}(z')^\top(z - z') \leq \tfrac{L}{2}\|z - z'\|^2.$$

*Furthermore, operator $\mathcal{H}(\cdot)$ is monotone and $M$-Lipschitz in the sense that for any $z, z' \in \mathcal{Z}$,*

$$\langle \mathcal{H}(z) - \mathcal{H}(z'), z - z' \rangle \geq 0, \quad \|\mathcal{H}(z) - \mathcal{H}(z')\| \leq M\|z - z'\|.$$

Second, we impose assumptions on the noise variance.

**Assumption 2.2 (Unbiased gradients and variance bounds)** *We assume that $z \in \mathcal{Z}$, samples $\xi \sim \mathcal{D}_\xi$ and $\zeta \sim \mathcal{D}_\zeta$ are drawn from given distributions such that the following conditions hold: $\mathbb{E}_\xi[\nabla\tilde{\mathcal{F}}(z; \xi)] = \nabla\mathcal{F}(z)$, $\mathbb{E}_\zeta[\tilde{\mathcal{H}}(z; \zeta)] = \mathcal{H}(z)$, and*

$$\mathbb{E}_\xi\left[\|\nabla\tilde{\mathcal{F}}(z; \xi) - \nabla\mathcal{F}(z)\|^2\right] \leq \sigma_{\mathrm{Str}}^2, \quad \mathbb{E}_\zeta\left[\|\tilde{\mathcal{H}}(z; \zeta) - \mathcal{H}(z)\|^2\right] \leq \sigma_{\mathrm{Bil}}^2. \tag{6}$$

For all results in this work, we suppose that Assumptions 2.1 and 2.2 hold with appropriate parameter settings. Given a desired accuracy $\varepsilon > 0$, our goal is to find an $\varepsilon$-*optimal point* defined as:

**Definition 2.1 ($\varepsilon$-Optimal point)** *A point $z \in \mathcal{Z}$ is called an $\varepsilon$-optimal point for the VI problem in (1) if $\|z - z^*\| \leq \varepsilon$.*

## 2.2 The ExtraGradient (EG) Algorithm

We first consider the case where $\mathcal{Z}$ is the entire space $\mathbb{R}^n$ and the objective is smooth ($J = 0$). The extragradient (EG) algorithm, introduced by Korpelevich [1976], is designed to address cyclic behavior in saddle-point problems by introducing an extrapolated point for gradient evaluation. In the context of VI problems (1), let $z_t$ represents the $t$-th iterate of the EG algorithm. The update rule of EG is as follows:

$$z_{t+1} = z_t - \eta\mathcal{W}(z_t - \eta\mathcal{W}(z_t)), \tag{7}$$

where $\eta > 0$ is the step size. For a $L$-smooth and $\mu$-strongly monotone operator $\mathcal{W}$, Tseng [1995], Mokhtari et al. [2020], Gidel et al. [2019a] have shown that the EG algorithm achieves an iteration complexity of $\mathcal{O}(\kappa \log(1/\varepsilon))$, where $\kappa = L/\mu$ denotes the condition number of the problem.

## 2.3 Accelerating the ExtraGradient Algorithm, Direct Approach

The convergence rate of the EG algorithm is far from optimal for the strongly monotone VI problem in (1) with separable structure (2). Firstly, the update rule in (7) takes $\mathcal{W}$ as a whole without utilizing the separable structure. This prevents us from exploiting the properties of $\nabla\mathcal{F}$. Secondly, in the case of bilinear games, the established lower bound for EG is $\Omega(\sqrt{\kappa}\log(1/\varepsilon))$ rather than $\Omega(\kappa\log(1/\varepsilon))$. This discrepancy highlights the potential for accelerating the EG algorithm in various directions. We first rewrite the EG update rule in (7) as follows:

$$z_{t-\frac{1}{2}} = z_{t-1} - \eta\mathcal{W}(z_{t-1}) = z_{t-1} - \eta\big(\mathcal{H}(z_{t-1}) + \nabla\mathcal{F}(z_{t-1})\big),$$
$$z_t = z_{t-1} - \eta\mathcal{W}(z_{t-\frac{1}{2}}) = z_{t-1} - \eta\big(\mathcal{H}(z_{t-\frac{1}{2}}) + \nabla\mathcal{F}(z_{t-\frac{1}{2}})\big). \tag{8}$$

To accelerate the process based on $\nabla\mathcal{F}$, we consider Nesterov's second acceleration scheme on minimizing a single convex function $\mathcal{F}$ [Tseng, 2008, Lan and Zhou, 2018, Lin et al., 2020c]:

$$z_{t-1}^{\mathrm{md}} = (1 - \alpha_t)z_{t-1}^{\mathrm{ag}} + \alpha_t z_{t-1}, z_t = z_{t-1} - \frac{\eta}{\alpha_t}\nabla\mathcal{F}(z_{t-1}^{\mathrm{md}}), z_t^{\mathrm{ag}} = (1 - \alpha_t)z_{t-1}^{\mathrm{ag}} + \alpha_t z_t, \tag{9}$$

where $\alpha_t$ is the extrapolation step size in the standard three-line Nesterov scheme. Here we adopt the notation $z^{\text{md}}$ and $z^{\text{ag}}$ to indicate the middle point and the aggregated point [Chen et al., 2017], respectively. Next, to achieve acceleration, we replace the gradient of $\nabla\mathcal{F}$ evaluated at both $z_{t-1}$ and $z_{t-\frac{1}{2}}$ in (8) by the gradient evaluated at the extrapolated point $z_{t-1}^{\text{md}}$ in (9). Furthermore, we shift the index of $z^{\text{ag}}$ by $\frac{1}{2}$ to indicate the use of $z_{t-\frac{1}{2}}$ instead of $z_t$ in the $z^{\text{ag}}$ update in (9). In addition, we take into account the $\mu$-strong convexity of $\mathcal{F}$ and shift the gradient of the strongly convex part $\nabla_z\left[\frac{\mu}{2}\|z-z_0\|^2\right] = \mu(z-z_0)$ from $\nabla\mathcal{F}(z)$ to $\mathcal{H}(z)$ as $\mathcal{W}(z) = (\nabla\mathcal{F}(z) - \mu(z-z_0)) + (\mathcal{H}(z) + \mu(z-z_0))$, we obtain the following update rule for a direct version of an accelerated EG algorithm (different step size schemes for $\eta_t$ are required for different algorithmic designs):

$$
\begin{cases}
z_{t-1}^{\text{md}} = (1-\alpha_t)z_{t-\frac{3}{2}}^{\text{ag}} + \alpha_t z_{t-1}, \\
z_{t-\frac{1}{2}} = z_{t-1} - \eta_t\left(\mathcal{H}(z_{t-1}) + \nabla\mathcal{F}(z_{t-1}^{\text{md}}) - \mu(z_{t-1}^{\text{md}} - z_{t-1})\right), \\
z_t = z_{t-1} - \eta_t\left(\mathcal{H}(z_{t-\frac{1}{2}}) + \nabla\mathcal{F}(z_{t-1}^{\text{md}}) - \mu(z_{t-1}^{\text{md}} - z_{t-\frac{1}{2}})\right), \\
z_{t-\frac{1}{2}}^{\text{ag}} = (1-\alpha_t)z_{t-\frac{3}{2}}^{\text{ag}} + \alpha_t z_{t-\frac{1}{2}}.
\end{cases}
\tag{10}
$$

We call the algorithm in (10) the *accelerated gradient-extragradient, direct approach* (AG-EG-Direct), and postpone its full description to Algorithm 2 in §C.1. The final output of the direct approach is $z_T$ after $T$ iterates. The following theorem records the convergence rate and iteration complexity of AG-EG (direct approach).

**Theorem 2.3 (Convergence of stochastic AG-EG, direct approach)** *Suppose Assumptions 2.1 and 2.2 hold. Fix any $r \in (0,1)$, $\beta \in (0,\infty)$, let $\kappa_\beta = \frac{L}{\mu} + \frac{(1+\beta)M^2}{\mu^2}$ and set the step size upper bound $\bar{\alpha} \equiv \frac{r}{1+\sqrt{1+r\kappa_\beta}}$. For any sequence of step sizes $\alpha_t \in (0, \bar{\alpha}]$ and $\eta_t = \frac{\alpha_t}{\mu}$, the iterates of stochastic AG-EG (direct approach) satisfy that for all $t = 1, \ldots, T$, we have*

$$
\mathbb{E}\|z_t - z^*\|^2 \le \|z_0 - z^*\|^2\left(\frac{L}{\mu}+1\right)\prod_{s=1}^{t}(1-\alpha_s) + \frac{3\sigma^2}{\mu^2}\sum_{s=1}^{t}\alpha_s^2\prod_{\tau=s+1}^{t}(1-\alpha_\tau),
\tag{11}
$$

*where we define $\sigma = \frac{1}{\sqrt{3}}\sqrt{\frac{1}{1-r}\sigma_{\text{Str}}^2 + (2+\frac{1}{\beta})\sigma_{\text{Bil}}^2}$.*

In the rest of the paper, we use the same definition $\sigma$ as in Theorem 2.3. The proof of Theorem 2.3 is provided in §D.4. We further note that one possible choice of step size is to let $\alpha_t \equiv \alpha$, such that (11) reduces to

$$
\mathbb{E}\|z_t - z^*\|^2 \le \|z_0 - z^*\|^2\left(\frac{L}{\mu}+1\right)e^{-\alpha t} + \frac{3\sigma^2}{\mu^2}\alpha.
$$

For any given $T \ge 1$, by choosing the optimal $\alpha = \frac{1}{T}\left(1 + \log\left(\frac{\mu^2 T}{3\sigma^2}\left(\frac{L}{\mu}+1\right)\|z_0 - z^*\|^2\right)\right) \wedge \bar{\alpha}$, (11) implies

$$
\mathbb{E}\|z_T - z^*\|^2 \le \|z_0 - z^*\|^2\left(\frac{L}{\mu}+1\right)e^{-\bar{\alpha}T} + \frac{3\sigma^2}{\mu^2 T}\left(1 + \log\left(\frac{\mu^2 T}{3\sigma^2}\left(\frac{L}{\mu}+1\right)\|z_0 - z^*\|^2\right)\right).
$$

Prescribing the desired accuracy $\varepsilon > 0$, the iteration complexity to output an $\varepsilon$-optimal minimax point is [3]

$$
\mathcal{O}\left(\left(\sqrt{\frac{L}{\mu}} + \frac{M}{\mu} + \frac{\sigma^2}{\mu^2\varepsilon^2}\right)\log\left(\left(\frac{L}{\mu}+1\right)\|z_0 - z^*\|^2/\varepsilon^2\right)\right).
$$

We conjecture that the logarithmic factor in the optimal statistical rate $\frac{\sigma^2}{\mu^2\varepsilon^2}$ is removable using a proper diminishing step size, a possibility that we reserve for future study. In the setting of deterministic optimization, setting $\sigma = 0$ and $r \to 1^-$, $\beta \to 0^+$ in Theorem 2.3, we obtain the optimal iteration complexity bound as follows:

$$
\left(1 + \sqrt{1 + \frac{L}{\mu} + \frac{M^2}{\mu^2}}\right)\log\left(\left(\frac{L}{\mu}+1\right)/\varepsilon^2\right).
\tag{12}
$$

---

[3]Throughout this work, we focus on the iteration complexity whereas the required number of queries to the stochastic gradient oracle is three times the iteration complexity.

---

**Algorithm 1** Stochastic AcceleratedGradient-ExtraGradient (AG-EG) Descent-Ascent Algorithm, with Scheduled Restarting

---

**Require:** Initialization $z_0^{[0]}$, total number of epochs $\mathscr{S} \geq 1$, total number of per-epoch iterates $(T_s : s = 1, \ldots, \mathscr{S})$, stepsizes $(\alpha_t, \eta_t : t = 1, 2, \ldots)$.
    **for** $s = 1, 2, \ldots, \mathscr{S}$ **do**
        Set $z_{-\frac{1}{2}}^{\mathrm{ag}} \leftarrow z_0^{[s-1]}, z_0 \leftarrow z_0^{[s-1]}, z_0^{\mathrm{md}} \leftarrow z_0^{[s-1]}$
        **for** $t = 1, 2, \ldots, T_s$ **do**
            Draw samples $\xi_{t-\frac{1}{2}} \sim \mathcal{D}_\xi$ from oracle, and also $\zeta_{t-\frac{1}{2}}, \zeta_t \sim \mathcal{D}_\zeta$ independently from oracle

$$z_{t-\frac{1}{2}} \leftarrow z_{t-1} - \eta_t \left( \tilde{\mathcal{H}}(z_{t-1}; \zeta_{t-\frac{1}{2}}) + \nabla \tilde{\mathcal{F}}(z_{t-1}^{\mathrm{md}}; \xi_{t-\frac{1}{2}}) \right)$$
$$z_{t-\frac{1}{2}}^{\mathrm{ag}} \leftarrow (1 - \alpha_t) z_{t-\frac{3}{2}}^{\mathrm{ag}} + \alpha_t z_{t-\frac{1}{2}}$$
$$z_t \leftarrow z_{t-1} - \eta_t \left( \tilde{\mathcal{H}}(z_{t-\frac{1}{2}}; \zeta_t) + \nabla \tilde{\mathcal{F}}(z_{t-1}^{\mathrm{md}}; \xi_{t-\frac{1}{2}}) \right)$$
$$z_t^{\mathrm{md}} \leftarrow (1 - \alpha_{t+1}) z_{t-\frac{1}{2}}^{\mathrm{ag}} + \alpha_{t+1} z_t$$

        **end for**
        Set $z_0^{[s]} \leftarrow z_{T_s-\frac{1}{2}}^{\mathrm{ag}}$            {//Warm-start using the output of the previous epoch}
    **end for**
    **Output:** $z_0^{[\mathscr{S}]}$

---

**Remark 2.4** *Our complexity bounds fundamentally differs from the previous analysis [Chen et al. 2017, Jordan et al., 2023] for separable smooth (strongly) monotone VIs. The convergence results in previous studies are dependent on the diameter of the domain, whereas our convergence rate is independent of the domain parameters and eliminates the need for projection onto a bounded domain. Moreover, our contributions go beyond those of Chen et al. [2017] by extending the analysis to the strongly monotone case. In comparison with Jordan et al. [2023], we design an algorithm where $\nabla \mathcal{F}$ is strongly monotone and resolve the open problem of extending the analysis to the stochastic case. Additionally, our complexity bound in (12) indicates a near-unity coefficient on the condition-number exponent, improving the corresponding coefficient in Chen et al. [2017, Theorem 15] by an asymptotic factor of* 4.

The direct approach, which reduces to EG when $\nabla \mathcal{F} = 0$ and $\mu = 0$, falls short of attaining optimality within the specific regime of bilinear games. In the next subsection, we will introduce a new algorithm that can overcome this limitation.

## 2.4 Accelerating the ExtraGradient Algorithm with Scheduled Restarting

In this subsection, we solve problem (1) by further accelerating the stochastic EG algorithm. Rather than directly relying on the strong monotonicity of $\nabla \mathcal{F}$, the inner updates of our new algorithm are identical to the updates in (10) with $\mu = 0$. Due to the domain-independent nature of our analysis, we can apply the scheduled restarting technique [O'donoghue and Candes, 2015, Roulet and d'Aspremont, 2017, Renegar and Grimmer, 2022] to the outer loop, accelerating the algorithm from sublinear convergence to linear convergence. In addition, the output of our algorithm is the aggregated point $z_{T-\frac{1}{2}}^{\mathrm{ag}}$ after $T$ iterates. We present the full algorithm in Algorithm 1.

We first present the convergence rate of a single epoch (i.e., the inner loop) of Algorithm 1 in Theorem 2.5. To accommodate more flexibility in the choice of parameters, we introduce three constants $r, \beta$, and $C$ in the theorem statement.

**Theorem 2.5 (Convergence of stochastic AG-EG, one epoch)** *Suppose Assumptions 2.1 and 2.2 hold. For any fixed epoch length $T \geq 1$, any constant $r \in (0, 1)$, $\beta \in (0, \infty)$, $C \in (0, \infty)$, choose step sizes $\alpha_t = \frac{2}{t+1}$ and $\eta_t$ such that*

$$\frac{t}{\eta_t} = \frac{2}{r} L \vee B + \sqrt{\frac{1+\beta}{r}} Mt, \tag{13}$$

*where $B = \frac{\sigma\sqrt{T}(T+1)}{C\sqrt{\mathbb{E}\|z_0 - z^*\|^2}}$. The output $z^{ag}_{T-\frac{1}{2}}$ of a single epoch of Algorithm 1 satisfies*

$$\mathbb{E}\left\|z^{ag}_{T-\frac{1}{2}} - z^*\right\|^2 \le \frac{2}{\mu(T+1)}\left(\frac{2L}{rT} + A\sqrt{\frac{1+\beta}{r}}M\right)\mathbb{E}\|z_0 - z^*\|^2 + \frac{2(\frac{1}{C}+C)\sigma}{\mu\sqrt{T}}\sqrt{\mathbb{E}\|z_0 - z^*\|^2},$$

(14)

*where the prefactor $A \equiv 1 + C^2 B\eta_1 \le 1 + C^2$ reduces to 1 when $\sigma = 0$.*

The proof of Theorem 2.5 is provided in §D.3. We make a few remarks on Theorem 2.5 as follows:

**Remark 2.6** *In the setting of deterministic optimization, by taking $\sigma = 0$, $r \to 1^-$, $\beta \to 0^+$ in our analysis, with step size choice $\eta_t = \frac{t}{2L+Mt}$, we obtain that*

$$\|z^{ag}_{T-\frac{1}{2}} - z^*\|^2 \le \frac{2}{\mu(T+1)}\left(\frac{2L}{T} + M\right)\|z_0 - z^*\|^2,$$

(15)

*In this setting, the algorithm is independent of $B$ and requires no knowledge of $\|z_0 - z^*\|^2$. In the face of stochasticity, we choose $C = 1$ when the initial distance to the optimal point is known. Alternatively, when only an over-estimate $\Gamma_0$ of $\sqrt{\mathbb{E}\|z_0 - z^*\|^2}$ is available, we can set (large enough) $C = \frac{\Gamma_0}{\sqrt{\mathbb{E}\|z_0 - z^*\|^2}} \ge 1$ to obtain*

$$\mathbb{E}\|z^{ag}_{T-\frac{1}{2}} - z^*\|^2 \le \frac{2}{\mu(T+1)}\left(\frac{2L}{rT} + 2\sqrt{\frac{1+\beta}{r}}M\right)\Gamma_0^2 + \frac{4\sigma}{\mu\sqrt{T}}\Gamma_0.$$

(16)

**Remark 2.7** *When the constants are not a concern, the coarse-grained choices of $r = \frac{1}{2}$ and $\beta = 1$ would suffice. Nevertheless, to optimize the constants, the tradeoff between the deviation of $r$ from 1 and $\beta$ from 0 is crucial, as it determines a balance between the stochastic gradient noise variance and the convergence rate coefficients.*

To prepare for our multi-epoch result with the help of scheduled restarting, we perform an induction based on (16) as follows. Supposing that $\mathbb{E}\|z_0^{[s-1]} - z^*\|^2 \le \Gamma_0^2 e^{1-s}$ hold, by taking $r = \frac{1}{2}$ and $\beta = 1$, we have

$$\mathbb{E}\|z_0^{[s]} - z^*\|^2 \lesssim \frac{L}{\mu T_s^2}\Gamma_0^2 e^{1-s} + \frac{M}{\mu T_s}\Gamma_0^2 e^{1-s} + \frac{\sigma}{\mu\sqrt{T_s}}\Gamma_0 e^{\frac{1-s}{2}}.$$

Setting the right-hand side of the above inequality to satisfy $\le \Gamma_0^2 e^{-s}$, and solving for $T_s$, we need the epoch length satisfies $T_s \asymp \sqrt{\frac{L}{\mu}} + \frac{M}{\mu} + \frac{\sigma^2}{\mu^2\Gamma_0^2 e^{1-s}}$. Thus, we can obtain the total iteration complexity as

$$\sum_{s=1}^{S}\left[\sqrt{\frac{L}{\mu}} + \frac{M}{\mu} + \frac{\sigma^2}{\mu^2\Gamma_0^2 e^{1-s}}\right] = \left(\sqrt{\frac{L}{\mu}} + \frac{M}{\mu}\right)S + \frac{\sigma^2}{\mu^2\Gamma_0^2}\cdot\frac{e^S - 1}{e - 1},$$

where $S \equiv \left\lceil \log\frac{\Gamma_0^2}{\varepsilon^2} \right\rceil$. This yields the following multi-epoch iteration complexity bound:

**Corollary 2.8 (Iteration complexity of stochastic AG-EG with scheduled restarting)** *Under the same condition of Theorem 2.5, the stochastic AG-EG with scheduled restarting in Algorithm 1 with epoch length $T_s \asymp \sqrt{\frac{L}{\mu}} + \frac{M}{\mu} + \frac{\sigma^2}{\mu^2\Gamma_0^2 e^{1-s}}$ has a total iteration complexity of*

$$\mathcal{O}\left(\left(\sqrt{\frac{L}{\mu}} + \frac{M}{\mu}\right)\log\left(\frac{1}{\varepsilon}\right) + \frac{\sigma^2}{\mu^2\varepsilon^2}\right).$$

(17)

Note that the hard instance constructed by Zhang et al. [2022] can be modified in a straightforward way to establish a lower bound of $\Omega\left(\left(\sqrt{\frac{L}{\mu}} + \frac{M}{\mu}\right)\log\left(\frac{1}{\varepsilon}\right)\right)$ for our monotone VI (1), demonstrating the optimality of Corollary 2.8 in the deterministic separable setting. An alternative optimality argument proceeds as follows: the first term $\sqrt{\frac{L}{\mu}}$ matches the lower bound for the minimization of a strongly convex function $\mathcal{F}$ [Nesterov, 2004], and the second term $\frac{M}{\mu}$ matches the lower bound for VI for non-strongly monotone operator when $\nabla\mathcal{F} = 0$ [Ouyang and Xu, 2021]. This together gives a lower bound for solving monotone VI (1) via a similar argument by Thekumparampil et al. [2022].

It is worth noting that while both complexity bounds in Corollary 2.8 and Theorem 2.3 match the lower bound in Zhang et al. [2022] for strongly monotone VIs with separable structure, the direct approach in §2.3 reduces to the *last-iterate* independent-sample stochastic extragradient (SEG) algorithm in *bilinear games*. Consequently, the deterministic part ($\sigma = 0$) fails to match the lower bound in Ibrahim et al. [2020]. In the stochastic case with noise variance bounded away from zero, the direct approach in §2.3 can exhibit *nonconvergence behavior* [Hsieh et al., 2020, §3]. The AG-EG algorithm in §2.4 resolves this issue by restarting the *average-iterate* SEG, matching the lower bound results (see §3.2 for more details). In addition, the complexity bound in (17) also eliminates the $\log$ prefactor of the statistical error term $\frac{\sigma^2}{\mu^2 \varepsilon^2}$ compared to Theorem 2.3. The optimality of our algorithm lies in not only the optimization complexity but also the statistical error rate $\frac{\sigma^2}{\mu^2 \epsilon^2}$. Here the $\varepsilon$-optimal point $z$ is defined as $\|\boldsymbol{z} - \boldsymbol{z}^*\| \leq \varepsilon$.[4]

## 2.5 Extension of AG-EG to Proximal Algorithms

In previous subsections, we have focused on the case where the feasible set $\mathcal{Z}$ represents the entire space and the nondifferentiable convex function $J$ is dropped. We now extend the AG-EG algorithm and its analysis to the more general setting that has a bounded feasible set (via Euclidean projection onto the feasible set) as well as a nondifferentiable convex regularization term (via a proximal operator). These settings are useful in various applications, such as the variational inequality on the Lorentz cone where projection onto $\mathcal{Z} = \left\{ (\boldsymbol{x}, t) \in \mathbb{R}^{(n+1)} : \|\boldsymbol{x}\| \leq t \right\}$ is required [Chen et al., 2017], and the two-player game that involves projection onto the probability simplex, among others. To deal with bounded feasible set $\mathcal{Z}$, we adopt a variant of the EG algorithm, where we project the extrapolated point and the main iterates back onto the feasible set $\mathcal{Z}$ of $\mathcal{W}$:

$$\boldsymbol{z}_{t-\frac{1}{2}} = P_{\mathcal{Z}} \left[ \boldsymbol{z}_{t-1} - \eta \mathcal{W}(\boldsymbol{z}_{t-1}) \right] = \arg\min_{\boldsymbol{z} \in \mathcal{Z}} \langle \boldsymbol{z} - \boldsymbol{z}_{t-1}, \eta \mathcal{W}(\boldsymbol{z}_{t-1}) \rangle + \frac{1}{2} \|\boldsymbol{z} - \boldsymbol{z}_{t-1}\|^2,$$

$$\boldsymbol{z}_t = P_{\mathcal{Z}} \left[ \boldsymbol{z}_{t-1} - \eta \mathcal{W}(\boldsymbol{z}_{t-\frac{1}{2}}) \right] = \arg\min_{\boldsymbol{z} \in \mathcal{Z}} \left\langle \boldsymbol{z} - \boldsymbol{z}_{t-1}, \eta \mathcal{W}(\boldsymbol{z}_{t-\frac{1}{2}}) \right\rangle + \frac{1}{2} \|\boldsymbol{z} - \boldsymbol{z}_{t-1}\|^2, \quad (18)$$

where $P_{\mathcal{Z}}(\boldsymbol{z}) = \arg\min_{\boldsymbol{z}' \in \mathcal{Z}} \|\boldsymbol{z} - \boldsymbol{z}'\|^2$ is the Euclidean projection operator. To handle the nondifferentiable simple convex function $J$, we replace the projection operator in (18) by the following proximal mapping defined via a Bregman divergence $\mathcal{B}(\cdot, \cdot)$:

$$\text{prox}_{\boldsymbol{z}}^J(\mathbf{v}) \equiv \arg\min_{\mathbf{u} \in \mathcal{Z}} \langle \mathbf{v}, \mathbf{u} - \boldsymbol{z} \rangle + \mathcal{B}(\boldsymbol{z}, \mathbf{u}) + J(\mathbf{u}). \quad (19)$$

In fact, (18) can be seen as a special case of (19) when choosing the Bregman divergence $\mathcal{B}(\boldsymbol{z}, \mathbf{u}) = \frac{1}{2} \|\boldsymbol{z} - \mathbf{u}\|^2$ and $J(\mathbf{u})$ as the set indicator function of the feasible set $\mathcal{Z}$. Therefore, by substituting the prox-mapping (19) into the AG-EG updates introduced in §2.4, we obtain the more general proximal AG-EG algorithm in Algorithm 3 (See in §C.2), which reduces to Algorithm 1 when $J = 0$, $\mathcal{B}(\boldsymbol{z}, \mathbf{u}) = \frac{1}{2} \|\boldsymbol{z} - \mathbf{u}\|^2$ and $\mathcal{Z} = \mathbb{R}^n$. Moreover, we assume that $\mathcal{B}(\cdot, \cdot)$ is $\mu_{\mathcal{B}}$-strongly convex. Without loss of generality, in contrast to the previous assumption of $\mu$-strong convexity for $\mathcal{F}$, we instead assume that $\mathcal{F}$ is $\mu$-strongly convex with respect to the Bregman divergence $\mathcal{B}(\cdot, \cdot)$ (See, for example, Hazan and Kale [2014], Xu et al. [2018]). Similar to Corollary 2.8, we have the following iteration complexity result, whose proof is deferred to §D.5:

**Corollary 2.9 (Iteration complexity of stochastic proximal AG-EG with scheduled restarting)** *Under the same condition of Theorem 2.5, the stochastic proximal AG-EG with scheduled restarting in Algorithm 3, with epoch length $T_s \asymp \sqrt{\frac{L}{\mu \mu_{\mathcal{B}}}} + \frac{M}{\mu \mu_{\mathcal{B}}} + \frac{\sigma^2 \mathcal{B}(\boldsymbol{z}_0, \boldsymbol{z}^*)}{\mu^2 \mu_{\mathcal{B}} \Gamma_0^2 e^{1-s}}$, has a total iteration complexity of*

$$\mathcal{O}\left( \left( \sqrt{\frac{L}{\mu \mu_{\mathcal{B}}}} + \frac{M}{\mu \mu_{\mathcal{B}}} \right) \log\left(\frac{1}{\varepsilon}\right) + \frac{\sigma^2 \mathcal{B}(\boldsymbol{z}_0, \boldsymbol{z}^*)}{\mu^2 \mu_{\mathcal{B}} \varepsilon^2} \right).$$

For the deterministic case, proximal AG-EG with scheduled restarting has a total iteration complexity of $\mathcal{O}\left( \left( \sqrt{\frac{L}{\mu \mu_{\mathcal{B}}}} + \frac{M}{\mu \mu_{\mathcal{B}}} \right) \log\left(\frac{1}{\varepsilon}\right) \right)$ to output an $\varepsilon$-optimal point of (1).

---

[4]The optimal statistical error rate $\frac{\sigma^2}{\mu^2 T}$ has been achieved by a multistage algorithm in Fallah et al. [2020], where the $\varepsilon$-optimal point is defined by $\|\boldsymbol{z} - \boldsymbol{z}^*\|^2 \leq \varepsilon$. In our paper, the $\varepsilon$-optimal point is defined by $\|\boldsymbol{z} - \boldsymbol{z}^*\| \leq \varepsilon$. Therefore, our statistical error rate can be translated into $\frac{\sigma^2}{\mu^2 T}$ using their definition, which matches their result.

# 3 Implications for Specific Instances

In this section, we discuss the implications of our AG-EG algorithm and its convergence rates when applying to two instances of saddle-point problems.

## 3.1 Strongly-Convex-Strongly-Concave Saddle-Point Problem

For the stochastic bilinearly-coupled SC-SC saddle-point problem (3), we note that the smoothness and strong convexity parameters $L_F$, $L_G$, $\mu_F$, and $\mu_G$ of $F$ and $G$ may differ. To accommodate these variations in curvature information, we employ a scaling reduction technique. This technique enables us to convert the SC-SC with equal strong convexity parameters for $F$ and $G$ by reparametrizing the objective function. The same argument is also applicable to the direct approach.

In lieu of (3), we consider

$$\min_{\hat{\boldsymbol{x}}} \max_{\hat{\boldsymbol{y}}} \hat{\mathscr{F}}(\hat{\boldsymbol{x}}, \hat{\boldsymbol{y}}) = F(\hat{\boldsymbol{x}}) + \hat{H}(\hat{\boldsymbol{x}}, \hat{\boldsymbol{y}}) - \hat{G}(\hat{\boldsymbol{y}}),$$

where $\hat{\mathscr{F}}(\hat{\boldsymbol{x}}, \hat{\boldsymbol{y}}) = \mathscr{F}(\boldsymbol{x}, \boldsymbol{y})$ with the symbolic reparametrization $\hat{\boldsymbol{x}} = \boldsymbol{x}$, $\hat{\boldsymbol{y}} = \sqrt{\frac{\mu_G}{\mu_F}} \boldsymbol{y}$, $\hat{H}(\hat{\boldsymbol{x}}, \hat{\boldsymbol{y}}) = H(\boldsymbol{x}, \boldsymbol{y})$, $\hat{G}(\hat{\boldsymbol{y}}) = G(\boldsymbol{y})$ and also their derivatives $\nabla_{\hat{\boldsymbol{y}}} \hat{H}(\hat{\boldsymbol{x}}, \hat{\boldsymbol{y}}) = \sqrt{\frac{\mu_F}{\mu_G}} \nabla_{\boldsymbol{y}} H(\boldsymbol{x}, \boldsymbol{y})$, $\nabla \hat{G}(\hat{\boldsymbol{y}}) = \sqrt{\frac{\mu_F}{\mu_G}} \nabla G(\boldsymbol{y})$ (the stochastic oracles $\hat{h}, \hat{g}$ follow the same rule). It is straightforward to verify that $\hat{\mathscr{F}}(\hat{\boldsymbol{x}}, \hat{\boldsymbol{y}})$ is $\mu$-strongly-convex-$\mu$-strongly-concave. The essence of our update rules can be summarized by the rescaled updates on $\boldsymbol{y}$:

$$\hat{\boldsymbol{y}}_t = \hat{\boldsymbol{y}}_{t-1} - \eta_t \left( -\nabla_{\hat{\boldsymbol{y}}} h(\hat{\boldsymbol{x}}_{t-\frac{1}{2}}, \hat{\boldsymbol{y}}_{t-\frac{1}{2}}; \zeta_t) + \nabla g(\hat{\boldsymbol{y}}_{t-1}^{\mathrm{md}}; \xi_{t-\frac{1}{2}}) \right)$$

$$\Leftrightarrow \boldsymbol{y}_t = \boldsymbol{y}_{t-1} - \eta_t \cdot \frac{\mu_F}{\mu_G} \left( -\nabla_{\boldsymbol{y}} h(\boldsymbol{x}_{t-\frac{1}{2}}, \boldsymbol{y}_{t-\frac{1}{2}}; \zeta_t) + \nabla g(\hat{\boldsymbol{y}}_{t-1}^{\mathrm{md}}; \xi_{t-\frac{1}{2}}) \right).$$

Therefore, it suffices to analyze Algorithm 3 for $\hat{\mathscr{F}}(\hat{\boldsymbol{x}}, \hat{\boldsymbol{y}})$ and due to this scaling reduction, we only need to prove all results for the case of $\mu_F = \mu_G = \mu$. It is also straightforward to justify corresponding scaling changes as: $L = L_F \vee \frac{\mu_F}{\mu_G} L_G$, $M = \sqrt{\frac{\mu_F}{\mu_G} \lambda_{\max}(\mathbf{B}^\top \mathbf{B})}$, and $\mu = \mu_F$. The following corollary is recovered by reverting the scaling reduction from $\hat{\mathscr{F}}(\hat{\boldsymbol{x}}, \hat{\boldsymbol{y}})$ to $\mathscr{F}(\boldsymbol{x}, \boldsymbol{y})$.

**Corollary 3.1 (Iteration complexity of stochastic AG-EG on SC-SC saddle-point problem)**
*For solving (3), Algorithm 1 with an epoch length $T_s \asymp \sqrt{\frac{L_F}{\mu_F} \vee \frac{L_G}{\mu_G}} + \sqrt{\frac{\lambda_{\max}(\mathbf{B}^\top \mathbf{B})}{\mu_F \mu_G}} + \frac{\sigma^2}{\mu_F^2 \Gamma_0^2 e^{1-s}}$ has a total iteration complexity of*

$$\mathcal{O}\left( \left( \sqrt{\frac{L_F}{\mu_F} \vee \frac{L_G}{\mu_G}} + \sqrt{\frac{\lambda_{\max}(\mathbf{B}^\top \mathbf{B})}{\mu_F \mu_G}} \right) \log\left(\frac{1}{\varepsilon}\right) + \frac{\sigma^2}{\mu_F^2 \varepsilon^2} \right).$$

In the deterministic case, the iteration complexity in Theorem 2.8 matches the lower bound established by Zhang et al. [2022], i.e., $\Omega\left( \left( \sqrt{\frac{L_F}{\mu_F} \vee \frac{L_G}{\mu_G}} + \sqrt{\frac{\lambda_{\max}(\mathbf{B}^\top \mathbf{B})}{\mu_F \mu_G}} \right) \log\left(\frac{1}{\varepsilon}\right) \right)$. Moreover, our algorithm achieves the optimal statistical rate of $\frac{\sigma^2}{\mu_F^2 \varepsilon^2}$ up to a constant prefactor.

**Remark 3.2** *A well-known finding regarding the second scheme of Nesterov acceleration is its connection to the primal-dual method [Lan and Zhou, 2018, Lin et al., 2020c]. This finding has been incorporated into the design of the LPD algorithm [Thekumparampil et al., 2022], where a Chambolle-Pock-style primal-dual method is utilized as an approximation of proximal point methods, instead of the extragradient used in this paper. The LPD algorithm [Thekumparampil et al., 2022] also achieves the optimal complexity for the deterministic bilinearly-coupled saddle-point problem.*

## 3.2 Bilinear Games

In this subsection, we consider the particular case of bilinear games. *We assume $n = m$ such that $\mathbf{B}$ is a nonsingular square matrix, $\nabla f(\boldsymbol{x}; \xi) = \mathbf{0}$ and $\nabla g(\boldsymbol{y}; \xi) = \mathbf{0}$ a.s., so (3) reduces to*

$$\min_{\boldsymbol{x}} \max_{\boldsymbol{y}} \mathscr{F}(\boldsymbol{x}, \boldsymbol{y}) = \mathbb{E}_\zeta [h(\boldsymbol{x}, \boldsymbol{y}; \zeta)] = H(\boldsymbol{x}, \boldsymbol{y}) = \boldsymbol{x}^\top \mathbf{B} \boldsymbol{y} - \boldsymbol{x}^\top \mathbf{u}_{\boldsymbol{x}} + \mathbf{u}_{\boldsymbol{y}}^\top \boldsymbol{y}, \tag{20}$$

and Algorithm 3 reduces to the independent-sample extragradient descent-ascent algorithm for (20). The saddle point $[\boldsymbol{z}^*; \boldsymbol{\omega}_y^\star]$ in this case is the unique solution to the linear equation

$$\begin{bmatrix} \mathbf{0} & \mathbf{B} \\ -\mathbf{B}^\top & \mathbf{0} \end{bmatrix} \begin{bmatrix} \boldsymbol{z}^* \\ \boldsymbol{\omega}_y^\star \end{bmatrix} = \begin{bmatrix} \mathbf{u}_x \\ \mathbf{u}_y \end{bmatrix}, \quad \text{which has a closed-form solution } \begin{bmatrix} \boldsymbol{z}^* \\ \boldsymbol{\omega}_y^\star \end{bmatrix} = \begin{bmatrix} -(\mathbf{B}^\top)^{-1}\mathbf{u}_y \\ \mathbf{B}^{-1}\mathbf{u}_x \end{bmatrix}.$$

Our results imply the following iteration complexity for solving stochastic bilinear games.

**Corollary 3.3 (Iteration complexity of stochastic AG-EG, bilinear games)** *For solving* (20), *choose the step sizes* $\alpha_t = \frac{2}{t+1}$ *and* $\eta_t \equiv \frac{1}{\sqrt{\lambda_{\max}(\mathbf{B}^\top\mathbf{B})}}$, *in which case Algorithm 1 with an epoch length* $T_s \asymp \sqrt{\frac{\lambda_{\max}(\mathbf{B}^\top\mathbf{B})}{\lambda_{\min}(\mathbf{B}\mathbf{B}^\top)}}$ *has the total iteration complexity of*

$$\mathcal{O}\left( \sqrt{\frac{\lambda_{\max}(\mathbf{B}^\top\mathbf{B})}{\lambda_{\min}(\mathbf{B}\mathbf{B}^\top)}} \log\left( \frac{\sqrt[4]{\lambda_{\min}(\mathbf{B}\mathbf{B}^\top)\lambda_{\max}(\mathbf{B}^\top\mathbf{B})}}{\sigma_{\mathrm{Bil}}} \right) + \frac{\sigma_{\mathrm{Bil}}^2}{\lambda_{\min}(\mathbf{B}\mathbf{B}^\top)\varepsilon^2} \right). \tag{21}$$

Note that our choice of the step size is maximal and is independent of the noise. In the deterministic setting, letting $\sigma_{\mathrm{Bil}} \asymp \varepsilon \sqrt[4]{\lambda_{\min}(\mathbf{B}\mathbf{B}^\top)\lambda_{\max}(\mathbf{B}^\top\mathbf{B})}$, the complexity bound in Corollary 3.3 reduces to $\mathcal{O}\left( \sqrt{\frac{\lambda_{\max}(\mathbf{B}^\top\mathbf{B})}{\lambda_{\min}(\mathbf{B}\mathbf{B}^\top)}} \log\left( \frac{1}{\varepsilon} \right) \right)$, which matches the lower bound in Ibrahim et al. [2020]. Notably, Azizian et al. [2020b] proposed an algorithm achieving an upper bound that matches the lower bound in Ibrahim et al. [2020].Li et al. [2022a] also proposed a lower-bound matching SEG algorithm that uses a shared sample in both steps under an unbounded noise assumption. In contrast, our algorithm is in the independent-sample setting with bounded noise variance.

**Remark 3.4** *Standard acceleration techniques do not attain the optimal nonasymptotic convergence rate for bilinear games [Gidel et al. 2019b]. This limitation applies to various algorithms, including the direct approach [§2.3], as well as several other acceleration techniques [Thekumparampil et al. 2022, Kovalev et al. 2022, Jin et al. 2022], all of which fall short of achieving optimal acceleration for bilinear games. Therefore, matching both lower bounds in a single algorithm in the general stochastic setting has been an open problem. While Li et al. [2022b] present an algorithm that achieves both lower bounds in a single algorithm, it relies on the use of optimistic gradients rather than extragradients on the bilinear coupling function. Furthermore, our algorithm and analysis is more general than those in Li et al. [2022b] as we can handle the general variational inequality with proximal operators.*

## 4 Conclusions

We have presented a stochastic extragradient-based acceleration algorithm, AG-EG, for solving stochastic monotone variational inequalities with separable structure. The iteration complexity of our algorithm matches the lower bound and is independent of the size of the feasible set. When specialized to solving the bilinearly coupled saddle-point problem (3), our AG-EG algorithm simultaneously matches lower bounds due to Zhang et al. [2022] and Ibrahim et al. [2020] for strongly-convex-strongly-concave and bilinear games, respectively. To the best of our knowledge, this is the first time that all three lower bounds have been met by a single algorithm. There are some remaining issues to be addressed, however, including the case of one-sided nonstrong convexity, the setting of unbounded noise variance, and the characterization of the full parameter regime dependency on $\lambda_{\min}(\mathbf{B}\mathbf{B}^\top)$. These are left as important directions for future research.

## Acknowledgments and Disclosure of Funding

This work is supported in part by Canada CIFAR AI Chair to GG, by the Mathematical Data Science program of the Office of Naval Research under grant number N00014-18-1-2764 and also the Vannevar Bush Faculty Fellowship program under grant number N00014-21-1-2941 and National Science Foundation (NSF) grant IIS-1901252 to MIJ. This work is also supported in part by the NSF CAREER Award 1906169 to QG and by NSF Award's IIS-2110170 and DMS-2134106 to SSD.

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
