_{\boldsymbol{z}}\left[\frac{\mu}{2}\|\boldsymbol{z} - \boldsymbol{z}_0\|^2\right] = \mu(\boldsymbol{z} - \boldsymbol{z}_0)$ from $\nabla\mathcal{F}(\boldsymbol{z})$ to $\mathcal{H}(\boldsymbol{z})$ as $\mathcal{W}(\boldsymbol{z}) = (\nabla\mathcal{F}(\boldsymbol{z}) - \mu(\boldsymbol{z} - \boldsymbol{z}_0)) + (\mathcal{H}(\boldsymbol{z}) + \mu(\boldsymbol{z} - \boldsymbol{z}_0))$, we obtain the following update rule for a direct version of an accelerated EG algorithm (different step size schemes for $\eta_t$ are required for different algorithmic designs):

$$
\begin{cases}
\boldsymbol{z}_{t-1}^{\mathrm{md}} = (1 - \alpha_t)\boldsymbol{z}_{t-\frac{3}{2}}^{\mathrm{ag}} + \alpha_t \boldsymbol{z}_{t-1}, \\[2mm]
\boldsymbol{z}_{t-\frac{1}{2}} = \boldsymbol{z}_{t-1} - \eta_t \left( \mathcal{H}(\boldsymbol{z}_{t-1}) + \nabla\mathcal{F}(\boldsymbol{z}_{t-1}^{\mathrm{md}}) - \mu(\boldsymbol{z}_{t-1}^{\mathrm{md}} - \boldsymbol{z}_{t-1}) \right), \\[2mm]
\boldsymbol{z}_t = \boldsymbol{z}_{t-1} - \eta_t \left( \mathcal{H}(\boldsymbol{z}_{t-\frac{1}{2}}) + \nabla\mathcal{F}(\boldsymbol{z}_{t-1}^{\mathrm{md}}) - \mu(\boldsymbol{z}_{t-1}^{\mathrm{md}} - \boldsymbol{z}_{t-\frac{1}{2}}) \right), \\[2mm]
\boldsymbol{z}_{t-\frac{1}{2}}^{\mathrm{ag}} = (1 - \alpha_t)\boldsymbol{z}_{t-\frac{3}{2}}^{\mathrm{ag}} + \alpha_t \boldsymbol{

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

The supplementary material is organized as follows. Section A provides specific examples in our minimax optimization setting. Section B compares our work with prior related works. Section C discusses the stochastic AG-EG algorithms in detail. Section D proves the main results. Finally, Section E provides proofs of auxiliary lemmas that support the proofs of main results.

## A Examples

We conduct an overview of some applications in this section.

**Reinforcement learning.** Reinforcement learning problems can be formalized as Markov Decision Processes (MDPs) where, at each step $t = 1, \ldots, n$, the learner receives a four-element tuple, $\{s_t, a_t, r_t, s_{t+1}\}$, where $(s_t, a_t)$ is the current state-action pair, $r_t$ is the reward received upon choosing $a_t$, and $s_{t+1}$ is the next state drawn from a transition distribution. For example, policy evaluation with a linear function approximator can be formalized in terms of the minimization of the *mean squared projected Bellman-Error* (MSPBE) [Dann et al., 2014] based on a set of tuples:

$$\min_{\boldsymbol{\theta}} \frac{1}{2} \|\mathbf{A}\boldsymbol{\theta} - \mathbf{b}\|_{\mathbf{C}^{-1}}^2 + \frac{\rho}{2} \|\boldsymbol{\theta}\|^2, \tag{22}$$

where $\mathbf{A} = \frac{1}{n} \sum_{t=1}^{n} \phi(s_t)(\phi(s_t) - \gamma\phi(s_{t+1}))^\top$, $\mathbf{b} = \frac{1}{n} \sum_{t=1}^{n} r_t\phi(s_t)$, and $\mathbf{C} = \frac{1}{n} \sum_{t=1}^{n} \phi(s_t)\phi(s_t)^\top$ for a given feature mapping $\phi$. To reduce the computational cost incurred by calculating the inverse of matrix $\mathbf{C}$, Du et al. [2017] propose an alternative minimax form of (22):

$$\min_{\boldsymbol{\theta}} \max_{\mathbf{w}} \frac{\rho}{2}\|\boldsymbol{\theta}\|^2 - \mathbf{w}^\top \mathbf{A}\boldsymbol{\theta} - \frac{1}{2}\|\mathbf{w}\|_{\mathbf{C}}^2 + \mathbf{w}^\top \mathbf{b},$$

which falls under the umbrella of problem (3) whenever $\mathbf{C}$ is positive definite.

**Quadratic games.** Another class of examples arises in the setting of bilinear games, where the minimax objective is:

$$\mathscr{F}(\boldsymbol{x}, \boldsymbol{y}) = \frac{1}{2}\boldsymbol{x}^\top \mathbf{M}_F \boldsymbol{x} + \boldsymbol{x}^\top \mathbf{B}\boldsymbol{y} - \frac{1}{2}\boldsymbol{y}^\top \mathbf{M}_G \boldsymbol{y} - \boldsymbol{x}^\top \mathbf{v}_{\boldsymbol{x}} + \mathbf{v}_{\boldsymbol{y}}^\top \boldsymbol{y}, \tag{23}$$

where $\mathbf{M}_F, \mathbf{M}_G$ are real-valued matrices of dimensions $n \times n$ and $m \times m$. This has the form (3) with $F(\boldsymbol{x}) = \frac{1}{2}\boldsymbol{x}^\top \mathbf{M}_F \boldsymbol{x} - \boldsymbol{x}^\top \mathbf{v}_{\boldsymbol{x}}$, $G(\boldsymbol{y}) = \frac{1}{2}\boldsymbol{y}^\top \mathbf{M}_G \boldsymbol{y} - \mathbf{v}_{\boldsymbol{y}}^\top \boldsymbol{y}$ and $H(\boldsymbol{x}, \boldsymbol{y}) \equiv \boldsymbol{x}^\top \mathbf{B}\boldsymbol{y}$. A particular case we will be considering in §3.2 is the case of bilinear games; i.e., where there are no quadratic terms. We provide a detailed analysis of the nonasymptotic convergence in this setting in §3.2 and show that the upper bound on the convergence rate given by our algorithm matches the lower bound of Ibrahim et al. [2020, Theorem 3].

**Regularized empirical risk minimization.** The problem of the minimization of the regularized empirical risk for convex losses and linear predictors is a core problem in classical supervised learning:

$$\min_{\boldsymbol{x} \in \mathbb{R}^d} \mathcal{L}(\mathbf{A}\boldsymbol{x}) + F(\boldsymbol{x}) \equiv \frac{1}{n} \sum_{i=1}^{n} \mathcal{L}_i(\mathbf{a}_i^\top \boldsymbol{x}) + F(\boldsymbol{x}),$$

where $\mathbf{A} = [\mathbf{a}_1, \ldots, \mathbf{a}_n]^\top \in \mathbb{R}^{n \times d}$ consists of feature vectors $\{\mathbf{a}_i\}$, $\mathcal{L}_i(\boldsymbol{y})$ is a univariate convex loss for the $i$th data point, and $F(\boldsymbol{x})$ is a convex regularizer. A standard construction turns this empirical risk minimization problem into a saddle-point problem as follows:

$$\min_{\boldsymbol{x} \in \mathbb{R}^d} \max_{\boldsymbol{y} \in \mathbb{R}^m} F(\boldsymbol{x}) + \boldsymbol{x}^\top \mathbf{A}\boldsymbol{y} - \underbrace{\mathcal{L}^\star(\boldsymbol{y})}_{\text{Legendre dual function of } \mathcal{L}(\boldsymbol{y})} \equiv F(\boldsymbol{x}) + \frac{1}{n} \sum_{i=1}^{n} \boldsymbol{x}_i \boldsymbol{y}^\top \mathbf{a}_i - \frac{1}{n} \sum_{i=1}^{n} \mathcal{L}^\star(\boldsymbol{y}_i).$$

See Zhang and Xiao [2017], Wang and Xiao [2017], Xiao et al. [2019] for in-depth discussions of solving this problem under such a dual form of representation.

## B Related Work

Here we compare our results with related work on saddle-point (minimax) optimization in the machine learning and optimization literature. In Table 1, we compare our AG-EG algorithm with previous work on solving saddle-point optimization problems, in terms of gradient complexity.

**Bilinear games.** In the bilinear game setting, where $L_F = \mu_F = L_G = \mu_G = 0$, a lower bound has been established by Ibrahim et al. [2020]: $\Omega\left(\sqrt{\frac{\lambda_{\max}(\mathbf{B}^\top\mathbf{B})}{\lambda_{\min}(\mathbf{B}\mathbf{B}^\top)}}\log\left(\frac{1}{\varepsilon}\right)\right)$. The study of bilinear game has been initiated by Daskalakis et al. [2018] for understanding saddle-point optimization. They proposed the optimistic gradient descent-ascent (OGDA) algorithm and achieved a sublinear convergence rate. Subsequently, the classical methods of ExtraGradient (EG) and Optimistic Gradient Descent Ascent (OGDA) algorithms were proven to have a linear convergence rate for strongly monotone and Lipschitz operator with $\mathcal{O}\left(\frac{\lambda_{\max}(\mathbf{B}^\top\mathbf{B})}{\lambda_{\min}(\mathbf{B}\mathbf{B}^\top)}\log(\frac{1}{\varepsilon})\right)$ iteration complexity [Gidel et al., 2019b, Mokhtari et al., 2020]. Azizian et al. [2020a] proved that by considering first-order methods with a fixed number of composed gradient evaluations and the last iterate as output (this class of methods is called 1-SCLI and excludes momentum and restarting), the $\mathcal{O}\left(\frac{\lambda_{\max}(\mathbf{B}^\top\mathbf{B})}{\lambda_{\min}(\mathbf{B}\mathbf{B}^\top)}\log(\frac{1}{\varepsilon})\right)$ iteration complexity for EG is optimal. In the absence of strong monotonicity assumption, Loizou et al. [2020] provided the first set of nonasymptotic last-iterate convergence guarantees for smooth games over a noncompact domain from a Hamiltonian viewpoint. The proposed stochastic Hamiltonian gradient method attains convergence in the finite-sum bilinear game setting as well. In a very recent work, Kovalev et al. [2022] derived an $\mathcal{O}\left(\frac{\lambda_{\max}(\mathbf{B}^\top\mathbf{B})}{\lambda_{\min}(\mathbf{B}\mathbf{B}^\top)}\log(\frac{1}{\varepsilon})\right)$ iteration complexity for convex-concave saddle-point problems with bilinear coupling. This is comparable to the rates in Daskalakis et al. [2018], Liang and Stokes [2019], Gidel et al. [2019b], Mokhtari et al. [2020], Mishchenko et al. [2020]. To match the $\Omega\left(\sqrt{\frac{\lambda_{\max}(\mathbf{B}^\top\mathbf{B})}{\lambda_{\min}(\mathbf{B}\mathbf{B}^\top)}}\log(\frac{1}{\varepsilon})\right)$ lower bound provided by Ibrahim et al. [2020], Azizian et al. [2020b] considered EG with momentum. They used a perturbed spectral analysis encompassing Polyak momentum. Nonetheless, Azizian et al. [2020b] only provided accelerated rates in the regime where the condition number is large. Li et al. [2022a] is the first to show that a variant of stochastic extragradient method converges at an accelerated rate for bilinear games with unbounded domain and unbounded stochastic noise using restarted iterate averaging, and matches the lower bound [Ibrahim et al., 2020] in the deterministic setting.

**Smooth and strongly-convex-strongly-concave saddle point problems.** Lower bound has been recently studied by Ouyang and Xu [2021] for smooth convex-concave minimax optimization, and by Zhang et al. [2022] for strongly-convex-strongly-concave saddle-point problems. The latter is of order $\Omega\left(\left(\sqrt{\frac{L_F}{\mu_F}\vee\frac{L_G}{\mu_G}}+\sqrt{\frac{\lambda_{\max}(\mathbf{B}^\top\mathbf{B})}{\mu_F\mu_G}}\right)\log\left(\frac{1}{\varepsilon}\right)\right)$. As for upper bounds, earlier extragradient-based methods [Tseng, 1995] and accelerated dual extrapolation algorithm [Nesterov and Scrimali, 2011] achieve, when restricted to the bilinearly coupled problem, an iteration complexity of $\tilde{\mathcal{O}}\left(\frac{L_F}{\mu_F}\vee\frac{L_G}{\mu_G}+\sqrt{\frac{\lambda_{\max}(\mathbf{B}^\top\mathbf{B})}{\mu_F\mu_G}}\right)$. The same iteration complexity has also been achieved by Gidel et al. [2019a], Mokhtari et al. [2020], Cohen et al. [2021] from a relative Lipschitz viewpoint.[5] Improving upon this result, Lin et al. [2020b] achieved a complexity of $\tilde{\mathcal{O}}\left(\sqrt{\frac{L_F L_G}{\mu_F\mu_G}}+\sqrt{\frac{\lambda_{\max}(\mathbf{B}^\top\mathbf{B})}{\mu_F\mu_G}}\right)$ using proper acceleration methods. Wang and Li [2020] achieved[6] $\tilde{\mathcal{O}}\left(\sqrt{\frac{L_F}{\mu_F}\vee\frac{L_G}{\mu_G}}+\sqrt[4]{\frac{\lambda_{\max}(\mathbf{B}^\top\mathbf{B})}{\mu_F\mu_G}\cdot\frac{L_F L_G}{\mu_F\mu_G}}+\sqrt{\frac{\lambda_{\max}(\mathbf{B}^\top\mathbf{B})}{\mu_F\mu_G}}\right)$ iteration complexity and a Hermitian-skew-based analysis nearly matches Zhang et al. [2022] for the quadratic minimax game. For the same problem, Xie et al. [2021] achieved a complexity of $\tilde{\mathcal{O}}\left(\sqrt[4]{\frac{L_F L_G}{\mu_F\mu_G}\left(\frac{L_F}{\mu_F}\vee\frac{L_G}{\mu_G}\right)}+\sqrt{\frac{\lambda_{\max}(\mathbf{B}^\top\mathbf{B})}{\mu_F\mu_G}}\right)$. These works improve upon Lin et al. [2020b] in a fine-grained fashion where separate Lipschitz constants on different parts of the objective are allowed. In early 2022, three concurrent works Kovalev et al. [2022], Thekumparampil et al. [2022], Jin et al. [2022] study the deterministic problem and independently match the lower bound by Zhang et al. [2022]. The main novelty of this work is that both lower bounds Ibrahim et al. [2020] and Zhang et al. [2022] are achieved in a single algorithm, plus an optimal statistical error rate up

---

[5]Mokhtari et al. [2020] report an $\tilde{\mathcal{O}}\left(\frac{L_F\vee L_G+\sqrt{\lambda_{\max}(\mathbf{B}^\top\mathbf{B})}}{\mu_F\wedge\mu_G}\right)$ complexity, but the mentioned complexity can be obtained via a scaling-reduction argument: consider $\mu_F=\mu_G$ case first, then consider the general case by rescaling the $y$ variable by a factor of $\sqrt{\frac{\mu_G}{\mu_F}}$.

[6]Note the cross term here, $\tilde{\mathcal{O}}\left(\sqrt[4]{\frac{\lambda_{\max}(\mathbf{B}^\top\mathbf{B})}{\mu_F\mu_G}\cdot\frac{L_F L_G}{\mu_F\mu_G}}\right)$, cannot be absorbed into the summation of the remaining terms.

to a constant prefactor in the stochastic setting. Recently, an independent work by Li et al. [2022b] also proposed a single algorithm that can achieve the optimal rates for both settings. However, their algorithm is based on optimistic gradient, and is less general than the variational inequality setting studied in this paper.

| Method \ Setting | Bilinearly-coupled SC-SC | Bilinear Game | Stochastic VI |
|---|---|---|---|
| EG / OGDA [Mokhtari et al., 2020] [Cohen et al., 2021] | $\frac{L_F}{\mu_F} \vee \frac{L_G}{\mu_G} + \sqrt{\frac{\lambda_{\max}(\mathbf{B}^\top\mathbf{B})}{\mu_F\mu_G}}$ | $\frac{\lambda_{\max}(\mathbf{B}^\top\mathbf{B})}{\lambda_{\min}(\mathbf{B}\mathbf{B}^\top)}$ | ✓ |
| Minimax-APPA [Lin et al., 2020b] | $\sqrt{\frac{L_F L_G}{\mu_F\mu_G}} + \sqrt{\frac{\lambda_{\max}(\mathbf{B}^\top\mathbf{B})}{\mu_F\mu_G}}$ | — | ✗ |
| Proximal Best Response [Wang and Li, 2020] | $\sqrt{\frac{L_F}{\mu_F} \vee \frac{L_G}{\mu_G}} + \sqrt[4]{\frac{\lambda_{\max}(\mathbf{B}^\top\mathbf{B})}{\mu_F\mu_G} \cdot \frac{L_F L_G}{\mu_F\mu_G}} + \sqrt{\frac{\lambda_{\max}(\mathbf{B}^\top\mathbf{B})}{\mu_F\mu_G}}$ | — | ✗ |
| DIPPA [Xie et al., 2021] | $\sqrt[4]{\frac{L_F L_G}{\mu_F\mu_G}\left(\frac{L_F}{\mu_F} \vee \frac{L_G}{\mu_G}\right)} + \sqrt{\frac{\lambda_{\max}(\mathbf{B}^\top\mathbf{B})}{\mu_F\mu_G}}$ | — | ✗ |
| LPD [Thekumparampil et al., 2022] | $\sqrt{\frac{L_F}{\mu_F} \vee \frac{L_G}{\mu_G}} + \sqrt{\frac{\lambda_{\max}(\mathbf{B}^\top\mathbf{B})}{\mu_F\mu_G}}$ | $\frac{\lambda_{\max}(\mathbf{B}^\top\mathbf{B})}{\lambda_{\min}(\mathbf{B}\mathbf{B}^\top)}$ | ✗ |
| APDG [Kovalev et al., 2022] | $\sqrt{\frac{L_F}{\mu_F} \vee \frac{L_G}{\mu_G}} + \sqrt{\frac{\lambda_{\max}(\mathbf{B}^\top\mathbf{B})}{\mu_F\mu_G}}$ | $\frac{\lambda_{\max}(\mathbf{B}^\top\mathbf{B})}{\lambda_{\min}(\mathbf{B}\mathbf{B}^\top)}$ | ✗ |
| PD-EG [Jin et al., 2022] | $\sqrt{\frac{L_F}{\mu_F} \vee \frac{L_G}{\mu_G}} + \sqrt{\frac{\lambda_{\max}(\mathbf{B}^\top\mathbf{B})}{\mu_F\mu_G}}$ | — | ✗ |
| EG+Momentum [Azizian et al., 2020b] | — | $\sqrt{\frac{\lambda_{\max}(\mathbf{B}^\top\mathbf{B})}{\lambda_{\min}(\mathbf{B}\mathbf{B}^\top)}}$ | ✗ |
| SEG with Restarting [Li et al., 2022a] | — | $\sqrt{\frac{\lambda_{\max}(\mathbf{B}^\top\mathbf{B})}{\lambda_{\min}(\mathbf{B}\mathbf{B}^\top)}}$ | ✓ |
| AG-EG-Direct (this work) | $\sqrt{\frac{L_F}{\mu_F} \vee \frac{L_G}{\mu_G}} + \sqrt{\frac{\lambda_{\max}(\mathbf{B}^\top\mathbf{B})}{\mu_F\mu_G}}$ | $\frac{\lambda_{\max}(\mathbf{B}^\top\mathbf{B})}{\lambda_{\min}(\mathbf{B}\mathbf{B}^\top)}$ | ✓ |
| AG-EG with Restarting (this work) | $\sqrt{\frac{L_F}{\mu_F} \vee \frac{L_G}{\mu_G}} + \sqrt{\frac{\lambda_{\max}(\mathbf{B}^\top\mathbf{B})}{\mu_F\mu_G}}$ | $\sqrt{\frac{\lambda_{\max}(\mathbf{B}^\top\mathbf{B})}{\lambda_{\min}(\mathbf{B}\mathbf{B}^\top)}}$ | ✓ |
| Lower Bound [Zhang et al., 2022] [Ibrahim et al., 2020] | $\Omega\left(\left(\sqrt{\frac{L_F}{\mu_F} \vee \frac{L_G}{\mu_G}} + \sqrt{\frac{\lambda_{\max}(\mathbf{B}^\top\mathbf{B})}{\mu_F\mu_G}}\right)\log\left(\frac{1}{\varepsilon}\right)\right)$ | $\tilde{\Omega}\left(\sqrt{\frac{\lambda_{\max}(\mathbf{B}^\top\mathbf{B})}{\lambda_{\min}(\mathbf{B}\mathbf{B}^\top)}}\log\left(\frac{1}{\varepsilon}\right)\right)$ | — |

| Reference | Stochastic variational inequality | No bounded domain assumption | No bounded noise assumption |
|---|---|---|---|
| [Korpelevich, 1976] [Juditsky et al., 2011] [Hsieh et al., 2020] | $\frac{L \vee M}{\varepsilon}\mathcal{D}^2 + \frac{\sigma^2}{\varepsilon^2}\mathcal{D}^2$ | ✗ | ✗ |
| [Li et al., 2022a] for bilinear games | $\frac{L \vee M}{\varepsilon}\Gamma_0^2 + \frac{\sigma^2}{\varepsilon^2}\Gamma_0^2$ | ✗ | ✓ |
| [Chen et al., 2017] [Lan and Ouyang, 2021] | $\sqrt{\frac{L}{\varepsilon}}\mathcal{D} + \frac{M}{\varepsilon}\mathcal{D}^2 + \frac{\sigma^2}{\varepsilon^2}\mathcal{D}^2$ | ✗ | ✗ |
| This work | $\sqrt{\frac{L}{\varepsilon}}\Gamma_0 + \frac{M}{\varepsilon}\Gamma_0^2 + \frac{\sigma^2}{\varepsilon^2}\Gamma_0^2$ | ✓ | ✗ |
| Lower Bound [Zhang et al., 2022] [Ouyang and Xu, 2021] | $\Omega\left(\sqrt{\frac{L}{\varepsilon}}\mathcal{D} + \frac{M}{\varepsilon}\mathcal{D}^2 + \frac{\sigma^2}{\varepsilon^2}\mathcal{D}^2\right)$ | ✗ | ✗ |

Table 1: A comparison of the first-order gradient complexities of our proposed algorithm with selected prevailing algorithms in terms of gradient complexity for solving a variety of saddle-point problems. Upper tabular: comparison of several cases such as general bilinearly-coupled SC-SC, bilinear games for finding an $\varepsilon$-optimal point, as well as a column indicating whether the stochastic variational inequality (VI) case is discussed. Lower tabular: complexities for stochastic VI for finding a point of $\varepsilon$ primal-dual gap, as well as columns of domain/noise assumptions (note that $\Gamma_0 \leq \mathcal{D}$). The row in red background is the convergence result presented in this paper. The "—" indicates that the complexity does not apply to the given case. A polylogarithm factor in each upper bound in the table is ignored.

**Stochastic minimax optimization.** Stochastic minimax optimization has been studied intensively as a special case of the variational inequalities. It is widely assumed in the classical literature on stochastic variational inequality [Juditsky et al., 2011] that the set of parameters and the variance of the stochastic estimate of the vector field are bounded. Chen et al. [2017] extended the analysis of Juditsky et al. [2011] and achieved an accelerated convergence rates for a class of variational

**Algorithm 2** Stochastic AcceleratedGradient-ExtraGradient (AG-EG) Descent-Ascent Algorithm, Direct Approach

---

**Require:** Initialization $z_0$, total number of iterates $T$, step sizes $(\alpha_t, \eta_t : t = 1, 2, \dots)$

1: Set $z_{-\frac{1}{2}}^{\mathrm{ag}} \leftarrow z_0$, $z_0^{\mathrm{md}} \leftarrow z_0$

2: **for** $t = 1, 2, \dots, T$ **do**

3:    Draw samples $\xi_{t-\frac{1}{2}} \sim \mathcal{D}_\xi$ from oracle, and also $\zeta_{t-\frac{1}{2}}, \zeta_t \sim \mathcal{D}_\zeta$ independently from oracle

4:    $z_{t-\frac{1}{2}} \leftarrow z_{t-1} - \eta_t \left( \tilde{\mathcal{H}}(z_{t-1}; \zeta_{t-\frac{1}{2}}) + \nabla \tilde{\mathcal{F}}(z_{t-1}^{\mathrm{md}}; \xi_{t-\frac{1}{2}}) - \mu(z_{t-1}^{\mathrm{md}} - z_{t-1}) \right)$

5:    $z_{t-\frac{1}{2}}^{\mathrm{ag}} \leftarrow (1 - \alpha_t) z_{t-\frac{3}{2}}^{\mathrm{ag}} + \alpha_t z_{t-\frac{1}{2}}$

6:    $z_t \leftarrow z_{t-1} - \eta_t \left( \tilde{\mathcal{H}}(z_{t-\frac{1}{2}}; \zeta_t) + \nabla \tilde{\mathcal{F}}(z_{t-1}^{\mathrm{md}}; \xi_{t-\frac{1}{2}}) - \mu(z_{t-1}^{\mathrm{md}} - z_{t-\frac{1}{2}}) \right)$

7:    $z_t^{\mathrm{md}} \leftarrow (1 - \alpha_{t+1}) z_{t-\frac{1}{2}}^{\mathrm{ag}} + \alpha_{t+1} z_t$

8: **end for**

9: **Output:** $z_T$

---

inequalities. Iusem et al. [2017] proposed an analysis of stochastic extragradient using large batches to reduce the variance. Mertikopoulos et al. [2018] showed almost sure convergence of Stochastic EG to a strictly coherent solution (a.k.a., star-strict monotone variational inequality problem). In a similar vein, Ryu et al. [2019] showed that stochastic gradient descent ascent (SGDA) with anchoring almost surely converges to strictly convex-concave saddle points. Fallah et al. [2020] developed a multistage variant of SGDA and stochastic optimistic gradient descent ascent with constant learning rate decay schedule. We improve upon their rates since their iteration complexity depends on a significantly larger condition number than our method and is infinite in the absence of strong convexity and strong concavity. They achieved the optimal dependency on the noise variance but suboptimal dependency on the condition number. Hsieh et al. [2020] developed a double step size extragradient method and proved the last-iterate convergence rates under an error bound condition similar to star-strong monotonicity. Kotsalis et al. [2020] proposed a simple and optimal scheme for a class of generalized strongly monotone (stochastic) variational inequalities. Due to the unconstrained nature of stochastic bilinear games, these two assumptions do not hold in this case because the noise increases with the value of the parameters. Mishchenko et al. [2020] showed that stochastic extragradients can be computed under a different step size, which removes the bounded domain assumption, while still requiring the bounded noise assumption. They also discussed the advantages of using the same mini-batch for the two stochastic gradients in stochastic extragradient. In another vein, Jelassi et al. [2020] focused on stochastic extragradient in games with a large number of players. They proposed an extragradient algorithm that randomly updates a small subset of the players at each iteration. Yan et al. [2019, 2020], Rafique et al. [2021] studied the nonsmooth setting and obtained fast rates. More recent works consider minimax optimization problems without convexity and/or concavity, where the goal is to find first-order and second-order stationary points [Lin et al., 2020a, Guo et al., 2020, Chen et al., 2021, Yang et al., 2022, Luo et al., 2022, Sebbouh et al., 2022]. One interesting direction is to extend our algorithm to these settings and obtain a fine-grained complexity bound with optimal rates.

## C    Algorithms

In this section we provide delayed algorithms for the AG-EG (direct approach) and the AG-EG with bounded domain and proximal operator.

### C.1    Stochastic AG-EG, Direct Approach

The full algorithm for AG-EG, direct approach is shown in Algorithm 2.

### C.2    Stochastic AG-EG, with Restarting and Projection

The full algorithm for AG-EG, with restarting and projection is shown in algorithm 3.

**Algorithm 3** Stochastic AcceleratedGradient-ExtraGradient (AG-EG) Descent-Ascent Algorithm, with Scheduled Restarting

---

**Require:** Initialization $z_0^{[0]}$, total number of epochs $\mathscr{S} \geq 1$, total number of per-epoch iterates $(T_s : s = 1, \ldots, \mathscr{S})$, step sizes $(\alpha_t, \eta_t : t = 1, 2, \ldots)$, ratio of strong-convexity params. $\mathscr{R} = \frac{\mu_G}{\mu_F}$
  **for** $s = 1, 2, \ldots, \mathscr{S}$ **do**
    Set $z_{-\frac{1}{2}}^{\mathrm{ag}} \leftarrow z_0^{[s-1]}, z_0 \leftarrow z_0^{[s-1]}, z_0^{\mathrm{md}} \leftarrow z_0^{[s-1]}$
    **for** $t = 1, 2, \ldots, T_s$ **do**
      Draw samples $\xi_{t-\frac{1}{2}} \sim \mathcal{D}_\xi$ from oracle, and also $\zeta_{t-\frac{1}{2}}, \zeta_t \sim \mathcal{D}_\zeta$ independently from oracle

$$z_{t-\frac{1}{2}} \leftarrow \mathrm{prox}_{z_{t-1}}^{\eta_t J} \left( \eta_t \tilde{\mathcal{H}}(z_{t-1}; \zeta_{t-\frac{1}{2}}) + \eta_t \nabla \tilde{\mathcal{F}}(z_{t-1}^{\mathrm{md}}; \xi_{t-\frac{1}{2}}) \right)$$

$$z_{t-\frac{1}{2}}^{\mathrm{ag}} \leftarrow (1 - \alpha_t) z_{t-\frac{3}{2}}^{\mathrm{ag}} + \alpha_t z_{t-\frac{1}{2}}$$

$$z_t \leftarrow \mathrm{prox}_{z_{t-1}}^{\eta_t J} \left( \eta_t \tilde{\mathcal{H}}(z_{t-\frac{1}{2}}; \zeta_t) + \eta_t \nabla \tilde{\mathcal{F}}(z_{t-1}^{\mathrm{md}}; \xi_{t-\frac{1}{2}}) \right)$$

$$z_t^{\mathrm{md}} \leftarrow (1 - \alpha_{t+1}) z_{t-\frac{1}{2}}^{\mathrm{ag}} + \alpha_{t+1} z_t$$

    **end for**
    Set $z_0^{[s]} \leftarrow z_{t_s - \frac{1}{2}}^{\mathrm{ag}}$                 {//Warm-start using the output of the previous epoch}
  **end for**
  **Output:** $z_0^{[\mathscr{S}]}$

---

# D  Proofs of Main Results

In this section we present the proofs of our main results. §D.1 illustrates the scaling reduction argument used in the instance of bilinearly-coupled saddle-point problem. §D.2 provides auxiliary lemmas. With a slight adjustment of their presentation order §D.3 proves Theorem 2.5, §D.4 proves Theorem 2.3, §D.5 proves Corollary 2.9 and finally §D.6 proves Corollary 3.3. Throughout the section, we assume that the Bregman divergence $\mathcal{B}(\cdot, \cdot)$ is $\mu_\mathcal{B}$-strongly convex.

## D.1  Scaling Reduction Argument

Here we illustrate the scaling reduction argument that reduces our analysis of our AG-EG Algorithm 1 under bilinearly-coupled saddle-point problem to the one with equal strong-convexity parameters of $F$ and $G$ using a reparameterized objective function; the same argument applies to Algorithm 2 and we omit the details. The idea is in fact analogous to mirror descent-ascent with respect to a Bregman divergence, and our goal here is to detail this argument for our analysis.

In lieu to (3) we consider

$$\min_{\hat{x}} \max_{\hat{y}} \ \hat{\mathscr{F}}(\hat{x}, \hat{y}) = F(\hat{x}) + \hat{H}(\hat{x}, \hat{y}) - \hat{G}(\hat{y}),$$

where we have $\hat{\mathscr{F}}(\hat{x}, \hat{y}) = \mathscr{F}(x, y)$ with the symbolic reparameterization $\hat{x} = x$, $\hat{y} = \sqrt{\frac{\mu_G}{\mu_F}} y$, $\hat{H}(\hat{x}, \hat{y}) = H(x, y)$, $\hat{h}(\hat{x}, \hat{y}; \zeta) = h(x, y; \zeta)$, $\hat{G}(\hat{y}) = G(y)$, $\hat{g}(\hat{y}; \xi) = g(y; \xi)$ and also their derivatives

$$\nabla_{\hat{y}} \hat{H}(\hat{x}, \hat{y}) = \sqrt{\frac{\mu_F}{\mu_G}} \nabla_y H(x, y), \qquad \nabla_{\hat{y}} \hat{h}(\hat{x}, \hat{y}; \zeta) = \sqrt{\frac{\mu_F}{\mu_G}} \nabla_y h(x, y; \zeta),$$

and

$$\nabla \hat{G}(\hat{y}) = \sqrt{\frac{\mu_F}{\mu_G}} \nabla G(y), \qquad \nabla \hat{g}(\hat{y}; \xi) = \sqrt{\frac{\mu_F}{\mu_G}} \nabla g(y; \xi).$$

It is straightforward to verify $\hat{\mathscr{F}}(\hat{x}, \hat{y})$ is arguably $\mu$-strongly-convex-$\mu$-strongly-concave. The essence of our update rules is captured by 8 lines corresponding to Lines 5–8 in Algorithm 1, which becomes:

$$\hat{x}_{t-\frac{1}{2}} = \hat{x}_{t-1} - \eta_t \left( \nabla f(\hat{x}_{t-1}^{\mathrm{md}}; \xi_{t-\frac{1}{2}}) + \nabla_{\hat{x}} h(\hat{x}_{t-1}, \hat{y}_{t-1}; \zeta_{t-\frac{1}{2}}) \right), \tag{24a}$$

$$\hat{y}_{t-\frac{1}{2}} = \hat{y}_{t-1} - \eta_t \left( -\nabla_{\hat{y}} h(\hat{x}_{t-1}, \hat{y}_{t-1}; \zeta_{t-\frac{1}{2}}) + \nabla g(\hat{y}_{t-1}^{\mathrm{md}}; \xi_{t-\frac{1}{2}}) \right), \tag{24b}$$

$$\hat{\boldsymbol{x}}^{\text{ag}}_{t-\frac{1}{2}} = (1-\alpha_t)\hat{\boldsymbol{x}}^{\text{ag}}_{t-\frac{3}{2}} + \alpha_t\hat{\boldsymbol{x}}_{t-\frac{1}{2}}, \tag{24c}$$

$$\hat{\boldsymbol{y}}^{\text{ag}}_{t-\frac{1}{2}} = (1-\alpha_t)\hat{\boldsymbol{y}}^{\text{ag}}_{t-\frac{3}{2}} + \alpha_t\hat{\boldsymbol{y}}_{t-\frac{1}{2}}, \tag{24d}$$

$$\hat{\boldsymbol{x}}_t = \hat{\boldsymbol{x}}_{t-1} - \eta_t\left(\nabla f(\hat{\boldsymbol{x}}^{\text{md}}_{t-1}; \xi_{t-\frac{1}{2}}) + \nabla_{\hat{\boldsymbol{x}}} h(\hat{\boldsymbol{x}}_{t-\frac{1}{2}}, \hat{\boldsymbol{y}}_{t-\frac{1}{2}}; \zeta_t)\right), \tag{24e}$$

$$\hat{\boldsymbol{y}}_t = \hat{\boldsymbol{y}}_{t-1} - \eta_t\left(-\nabla_{\hat{\boldsymbol{y}}} h(\hat{\boldsymbol{x}}_{t-\frac{1}{2}}, \hat{\boldsymbol{y}}_{t-\frac{1}{2}}; \zeta_t) + \nabla g(\hat{\boldsymbol{y}}^{\text{md}}_{t-1}; \xi_{t-\frac{1}{2}})\right), \tag{24f}$$

$$\hat{\boldsymbol{x}}^{\text{md}}_t = (1-\alpha_{t+1})\hat{\boldsymbol{x}}^{\text{ag}}_{t-\frac{1}{2}} + \alpha_{t+1}\hat{\boldsymbol{x}}_t, \tag{24g}$$

$$\hat{\boldsymbol{y}}^{\text{md}}_t = (1-\alpha_{t+1})\hat{\boldsymbol{y}}^{\text{ag}}_{t-\frac{1}{2}} + \alpha_{t+1}\hat{\boldsymbol{y}}_t. \tag{24h}$$

The rest translations are also straightforward, represented by

$$\hat{\boldsymbol{x}}_{t-\frac{1}{2}} = \hat{\boldsymbol{x}}_{t-1} - \eta_t\left(\nabla f(\hat{\boldsymbol{x}}^{\text{md}}_{t-1}; \xi_{t-\frac{1}{2}}) + \nabla_{\hat{\boldsymbol{x}}} h(\hat{\boldsymbol{x}}_{t-1}, \hat{\boldsymbol{y}}_{t-1}; \zeta_{t-\frac{1}{2}})\right)$$

$$\Leftrightarrow \boldsymbol{x}_{t-\frac{1}{2}} = \boldsymbol{x}_{t-1} - \eta_t\left(\nabla f(\boldsymbol{x}^{\text{md}}_{t-1}; \xi_{t-\frac{1}{2}}) + \nabla_{\boldsymbol{x}} h(\boldsymbol{x}_{t-1}, \boldsymbol{y}_{t-1}; \zeta_{t-\frac{1}{2}})\right),$$

as well as

$$\hat{\boldsymbol{y}}_t = \hat{\boldsymbol{y}}_{t-1} - \eta_t\left(-\nabla_{\hat{\boldsymbol{y}}} h(\hat{\boldsymbol{x}}_{t-\frac{1}{2}}, \hat{\boldsymbol{y}}_{t-\frac{1}{2}}; \zeta_t) + \nabla g(\hat{\boldsymbol{y}}^{\text{md}}_{t-1}; \xi_{t-\frac{1}{2}})\right)$$

$$\Leftrightarrow \boldsymbol{y}_t = \boldsymbol{y}_{t-1} - \eta_t \cdot \frac{\mu_F}{\mu_G}\left(-\nabla_{\boldsymbol{y}} h(\boldsymbol{x}_{t-\frac{1}{2}}, \boldsymbol{y}_{t-\frac{1}{2}}; \zeta_t) + \nabla g(\hat{\boldsymbol{y}}^{\text{md}}_{t-1}; \xi_{t-\frac{1}{2}})\right).$$

It is also straightforward to justify that Assumptions 2.1 and 2.2 are rediscovered by reverting the scaling reduction from $\hat{\mathscr{F}}(\hat{\boldsymbol{x}}, \hat{\boldsymbol{y}})$ to $\mathscr{F}(\boldsymbol{x}, \boldsymbol{y})$. Therefore, it suffices to analyze Algorithm 1 for $\hat{\mathscr{F}}(\hat{\boldsymbol{x}}, \hat{\boldsymbol{y}})$ and due to this scaling reduction, we only need to prove all results for the case of $\frac{\mu_F}{\mu_G} = 1$. To keep the notations simple, till the rest of this work we slightly abuse the notations and remove the hats in all symbols.

## D.2 Auxiliary Lemmas

We first state the following basic lemma to handle the inner-product induced terms for extragradient analysis:

**Lemma D.1** . *Given $\boldsymbol{\theta}, \boldsymbol{\varphi}_1, \boldsymbol{\varphi}_2 \in \mathcal{Z}$, a simple and convex function $J(\cdot)$, and also $\boldsymbol{\delta}_1, \boldsymbol{\delta}_2$ that satisfies*

$$\boldsymbol{\varphi}_1 = \text{prox}^J_{\boldsymbol{\theta}}(\boldsymbol{\delta}_1), \qquad \boldsymbol{\varphi}_2 = \text{prox}^J_{\boldsymbol{\theta}}(\boldsymbol{\delta}_2), \tag{25}$$

*then for any $\mathbf{z} \in \mathcal{Z}$ we have*

$$\langle \boldsymbol{\delta}_2, \boldsymbol{\varphi}_1 - \boldsymbol{z} \rangle + J(\boldsymbol{\varphi}_1) - J(\boldsymbol{z}) \le \frac{1}{2\mu_{\mathcal{B}}}\|\boldsymbol{\delta}_2 - \boldsymbol{\delta}_1\|^2 + \mathcal{B}(\boldsymbol{\theta}, \boldsymbol{z}) - \mathcal{B}(\boldsymbol{\varphi}_2, \boldsymbol{z}) - \mathcal{B}(\boldsymbol{\theta}, \boldsymbol{\varphi}_1). \tag{26}$$

*Furthermore, when taking $J = 0$, $\mathcal{Z} = \mathbb{R}^d$ and $\mathcal{B}(\boldsymbol{z}, \mathbf{u}) = 1/2\|\boldsymbol{z} - \mathbf{u}\|^2$, (26) reduces to:*

$$\langle \boldsymbol{\delta}_2, \boldsymbol{\varphi}_1 - \boldsymbol{z} \rangle \le \frac{1}{2}\|\boldsymbol{\delta}_2 - \boldsymbol{\delta}_1\|^2 + \frac{1}{2}\left[\|\boldsymbol{\theta} - \boldsymbol{z}\|^2 - \|\boldsymbol{\varphi}_2 - \boldsymbol{z}\|^2 - \|\boldsymbol{\theta} - \boldsymbol{\varphi}_1\|^2\right]. \tag{27}$$

Proof of Lemma D.1 is provided in §E.1. Lemma D.1 is standard and commonly adopted in extragradient-based analysis; see Lemma 2 of [Chen et al., 2017] for one with similar flavor.

En route to our proofs of Theorems 2.5 and 2.3, we first introduce some notations. Let $\tilde{\boldsymbol{z}} \in \mathcal{Z}$ and let the *pointwise primal-dual gap* function be

$$V(\boldsymbol{z} \mid \tilde{\boldsymbol{z}}) = \mathcal{F}(\boldsymbol{z}) - \mathcal{F}(\tilde{\boldsymbol{z}}) + \langle \mathcal{H}(\tilde{\boldsymbol{z}}), \boldsymbol{z} - \tilde{\boldsymbol{z}} \rangle. \tag{28}$$

We prove that this quantity is lower bounded by a positive quadratic function:

**Lemma D.2** *For $L$-smooth and $\mu$-strongly convex $\mathcal{F}(\boldsymbol{z})$, simple and convex $J$, and for any $\boldsymbol{z} \in \mathcal{Z}$ we have*

$$V(\boldsymbol{z} \mid \boldsymbol{z}^*) = \mathcal{F}(\boldsymbol{z}) - \mathcal{F}(\boldsymbol{z}^*) + \langle \nabla\mathcal{H}(\boldsymbol{z}^*), \boldsymbol{z} - \boldsymbol{z}^* \rangle + J(\boldsymbol{z}) - J(\boldsymbol{z}^*) \ge \frac{\mu}{2}\|\boldsymbol{z} - \boldsymbol{z}^*\|^2. \tag{29}$$

Proof of Lemma D.2 is provided in §E.2. Our final auxiliary lemma on the key properties on step sizes writes as follows:

**Lemma D.3** *Our step size choice* (13) *satisfies (i)* $\eta_t \leq \frac{t}{B}$; *(ii)* $\left(\frac{t}{\eta_t} : t \geq 1\right)$ *is a nonnegative, nondecreasing arithmetic sequence with common difference* $\sqrt{\frac{1+\beta}{r}}M$; *(iii)* $M\eta_t \leq 1$, *and (iv) the step size condition*

$$r - \frac{2L}{t+1}\eta_t - (1+\beta)M^2\eta_t^2 \geq 0. \tag{30}$$

Proof of Lemma D.3 is provided in §E.3.

### D.3 Proof of Theorem 2.5

*Proof.*[Proof of Theorem 2.5]

We first introduce some notations. Denote the incurred stochastic noise terms as

$$\boldsymbol{\Delta}_{\mathcal{F}}^{t-\frac{1}{2}} \equiv \nabla\tilde{\mathcal{F}}(\boldsymbol{z}_{t-1}^{\text{md}};\xi_{t-\frac{1}{2}}) - \nabla\mathcal{F}(\boldsymbol{z}_{t-1}^{\text{md}}), \qquad \boldsymbol{\Delta}_{\mathcal{H}}^{t-\frac{1}{2}} \equiv \tilde{\mathcal{H}}(\boldsymbol{z}_{t-1};\zeta_{t-\frac{1}{2}}) - \mathcal{H}(\boldsymbol{z}_{t-1}),$$
$$\boldsymbol{\Delta}_{\mathcal{H}}^{t} \equiv \tilde{\mathcal{H}}(\boldsymbol{z}_{t-\frac{1}{2}};\zeta_t) - H(\boldsymbol{z}_{t-\frac{1}{2}}). \tag{31}$$

For our martingale analysis we adopt the filtrations $\mathcal{F}_t^{\xi} \equiv \sigma\left(\xi_s : s = \frac{1}{2}, \frac{3}{2}, \ldots, s \leq t\right)$ and $\mathcal{F}_t^{\zeta} \equiv \sigma\left(\zeta_s : s = \frac{1}{2}, 1, \frac{3}{2}, \ldots, s \leq t\right)$, and also $\mathcal{F}_t \equiv \sigma(\mathcal{F}_t^{\xi} \cup \mathcal{F}_t^{\zeta})$ be the $\sigma$-algebra generated by the union of $\mathcal{F}_t^{\xi}$ and $\mathcal{F}_t^{\zeta}$. We are ready for the proof which proceeds as the following steps:

**Step 1.** Estimating the primal-dual gap function difference sequence. We provide the following Lemma (D.4), whose proof is in §E.4:

**Lemma D.4** *For arbitrary* $\tilde{\boldsymbol{z}} \in \mathcal{Z}$ *the iterates of Algorithm 1 satisfy for* $t = 1, \ldots, T$, *almost surely*

$$V(\boldsymbol{z}_{t-\frac{1}{2}}^{ag} \mid \tilde{\boldsymbol{z}}) - (1 - \alpha_t)V(\boldsymbol{z}_{t-\frac{3}{2}}^{ag} \mid \tilde{\boldsymbol{z}})$$
$$\leq \alpha_t\langle\nabla\mathcal{F}(\boldsymbol{z}_{t-1}^{md}) + \mathcal{H}(\boldsymbol{z}_{t-\frac{1}{2}}), \boldsymbol{z}_{t-\frac{1}{2}} - \tilde{\boldsymbol{z}}\rangle + \frac{\alpha_t^2 L}{2}\left\|\boldsymbol{z}_{t-\frac{1}{2}} - \boldsymbol{z}_{t-1}\right\|^2. \tag{32}$$

Note the proof only relies on the interpolation updates in our algorithm as in Lines 6 and 8, and hence this result holds in a per-trajectory (almost-sure) fashion.

**Step 2.** We target to prove the following lemma, the complete proof is in §E.5

**Lemma D.5** *For our choice of* $\eta_t$ *that satisfies, for a given* $r \in (0,1)$, (30) *of Lemma D.3(iv) that* $r - \frac{2L}{t+1}\eta_t - (1+\beta)M^2\eta_t^2 \geq 0$, *we have for any* $\tilde{\boldsymbol{z}} \in \mathbb{R}^n$, $\tilde{\boldsymbol{y}} \in \mathbb{R}^m$ *and* $t = 1, \ldots, T$ *that*

$$t(t+1)\mathbb{E}[V(\boldsymbol{z}_{t-\frac{1}{2}}^{ag} \mid \tilde{\boldsymbol{z}})] - (t-1)t\mathbb{E}[V(\boldsymbol{z}_{t-\frac{3}{2}}^{ag} \mid \tilde{\boldsymbol{z}})]$$
$$\leq \frac{t}{\eta_t}\mathbb{E}\left[\|\boldsymbol{z}_{t-1} - \tilde{\boldsymbol{z}}\|^2 - \|\boldsymbol{z}_t - \tilde{\boldsymbol{z}}\|^2\right] + \left(\frac{1}{1-r}\sigma_{\text{Str}}^2 + (2 + \frac{1}{\beta})\sigma_{\text{Bil}}^2\right)t\eta_t, \tag{33}$$

Now for a given $1 \leq \mathcal{T} \leq T$, we finish the proof by telescope the above recursion for $t = 1, \ldots, \mathcal{T}$. We conclude from our choice of step size as in (13) that satisfies (30) so by denoting $\sigma \equiv \frac{1}{\sqrt{3}}\sqrt{\frac{1}{1-r}\sigma_{\text{Str}}^2 + (2 + \frac{1}{\beta})\sigma_{\text{Bil}}^2}$, we have by Lemma D.3(i)

$$\left(\frac{1}{1-r}\sigma_{\text{Str}}^2 + (2 + \frac{1}{\beta})\sigma_{\text{Bil}}^2\right)\sum_{t=1}^{\mathcal{T}} t\eta_t = 3\sigma^2\sum_{t=1}^{\mathcal{T}} t\eta_t \leq 3\sigma^2 \cdot \frac{1}{B}\sum_{t=1}^{\mathcal{T}} t^2$$
$$= 3\sigma^2 \cdot \frac{C\sqrt{\mathbb{E}[\|\boldsymbol{z}_0 - \tilde{\boldsymbol{z}}\|^2]}}{\sigma[T(T+1)^2]^{1/2}} \cdot \frac{\mathcal{T}(\mathcal{T}+\frac{1}{2})(\mathcal{T}+1)}{3} = \frac{\mathcal{T}(\mathcal{T}+\frac{1}{2})(\mathcal{T}+1)}{[T(T+1)^2]^{1/2}} \cdot \sigma C\sqrt{\mathbb{E}[\|\boldsymbol{z}_0 - \tilde{\boldsymbol{z}}\|^2]},$$

where $B \equiv \frac{\sigma[T(T+1)^2]^{1/2}}{C\sqrt{\mathbb{E}[\|\boldsymbol{z}_0 - \tilde{\boldsymbol{z}}\|^2]}}$. Finally by summing over $t = 1, \ldots, \mathcal{T}$, we have

$$\mathcal{T}(\mathcal{T}+1)\mathbb{E}[V(\boldsymbol{z}_{\mathcal{T}-\frac{1}{2}}^{\mathrm{ag}} \mid \tilde{\boldsymbol{z}})]$$

$$\leq \sum_{t=1}^{\mathcal{T}} \frac{t}{\eta_t} \mathbb{E}\left[\|\boldsymbol{z}_{t-1} - \tilde{\boldsymbol{z}}\|^2 - \|\boldsymbol{z}_t - \tilde{\boldsymbol{z}}\|^2\right] + \left(\tfrac{1}{1-r}\sigma_{\mathrm{Str}}^2 + (2 + \tfrac{1}{\beta})\sigma_{\mathrm{Bil}}^2\right) \sum_{t=1}^{\mathcal{T}} t\eta_t$$

$$= \frac{1}{\eta_1}\mathbb{E}\|\boldsymbol{z}_0 - \tilde{\boldsymbol{z}}\|^2 + \sum_{t=2}^{\mathcal{T}} \left(\frac{t}{\eta_t} - \frac{t-1}{\eta_{t-1}}\right)\mathbb{E}\|\boldsymbol{z}_{t-1} - \tilde{\boldsymbol{z}}\|^2 - \frac{\mathcal{T}}{\eta_{\mathcal{T}}}\mathbb{E}\|\boldsymbol{z}_{\mathcal{T}} - \tilde{\boldsymbol{z}}\|^2$$

$$+ \frac{\mathcal{T}(\mathcal{T}+\frac{1}{2})(\mathcal{T}+1)}{[T(T+1)^2]^{1/2}} \cdot C\sigma\sqrt{\mathbb{E}\|\boldsymbol{z}_0 - \tilde{\boldsymbol{z}}\|^2}.$$

Following the above derivations and apply Lemma D.3(ii) we obtain $\frac{t}{\eta_t} - \frac{t-1}{\eta_{t-1}} = \sqrt{\frac{1+\beta}{r}}M$. Rearranging the terms along with Jensen's inequality, and noting that

$$\frac{\mathcal{T}(\mathcal{T}+\frac{1}{2})(\mathcal{T}+1)}{[T(T+1)^2]^{1/2}} \leq \frac{T(T+\frac{1}{2})(T+1)}{[T(T+1)^2]^{1/2}} \leq [T(T+1)^2]^{1/2}$$

proves the following inequality (34).

$$\mathcal{T}(\mathcal{T}+1)\mathbb{E}[V(\boldsymbol{z}_{\mathcal{T}-\frac{1}{2}}^{\mathrm{ag}} \mid \tilde{\boldsymbol{z}})] + \frac{\mathcal{T}}{\eta_{\mathcal{T}}}\mathbb{E}\|\boldsymbol{z}_{\mathcal{T}} - \tilde{\boldsymbol{z}}\|^2$$

$$\leq \frac{1}{\eta_1}\mathbb{E}\|\boldsymbol{z}_0 - \tilde{\boldsymbol{z}}\|^2 + \sqrt{\frac{1+\beta}{r}}M\sum_{t=2}^{\mathcal{T}}\mathbb{E}\|\boldsymbol{z}_{t-1} - \tilde{\boldsymbol{z}}\|^2 + [T(T+1)^2]^{1/2} \cdot C\sigma\sqrt{\mathbb{E}\|\boldsymbol{z}_0 - \tilde{\boldsymbol{z}}\|^2}. \tag{34}$$

**Step 3. Bounded Iterates** We conduct the following "bootstrapping" argument to arrive at our final theorem. Starting from the recursion (34) we have by setting $\tilde{\boldsymbol{z}} = \boldsymbol{z}^*$, Lemma D.2 implies that its first summand on the left hand $\mathcal{T}(\mathcal{T}+1)\mathbb{E}[V(\boldsymbol{z}_{\mathcal{T}-\frac{1}{2}}^{\mathrm{ag}} \mid \boldsymbol{z}^*)]$ is nonnegative, and hence we can drop it and have for any $\mathcal{T} = 1, \ldots, T$

$$\frac{\mathcal{T}}{\eta_{\mathcal{T}}}\mathbb{E}\|\boldsymbol{z}_{\mathcal{T}} - \boldsymbol{z}^*\|^2 \tag{35}$$

$$\leq \frac{1}{\eta_1}\mathbb{E}\|\boldsymbol{z}_0 - \boldsymbol{z}^*\|^2 + \sqrt{\frac{1+\beta}{r}}M\sum_{t=2}^{\mathcal{T}}\mathbb{E}\|\boldsymbol{z}_{t-1} - \boldsymbol{z}^*\|^2 + [T(T+1)^2]^{1/2} \cdot C\sigma\sqrt{\mathbb{E}\|\boldsymbol{z}_0 - \boldsymbol{z}^*\|^2}$$

$$= (\tfrac{2}{r}L \vee B)\mathbb{E}\|\boldsymbol{z}_0 - \boldsymbol{z}^*\|^2 + \sqrt{\frac{1+\beta}{r}}M\underbrace{\sum_{t=1}^{\mathcal{T}}\mathbb{E}\|\boldsymbol{z}_{t-1} - \boldsymbol{z}^*\|^2}_{\equiv \mathcal{Q}_{\mathcal{T}-1}} + \underbrace{[T(T+1)^2]^{1/2} \cdot C\sigma\sqrt{\mathbb{E}\|\boldsymbol{z}_0 - \boldsymbol{z}^*\|^2}}_{\mathcal{R}_0}. \tag{36}$$

Converting (36) to a version of partial sum $\mathcal{Q}_{\mathcal{T}-1} \equiv \sum_{t=1}^{\mathcal{T}}\mathbb{E}\|\boldsymbol{z}_{t-1} - \boldsymbol{z}^*\|^2$ that for all $\mathcal{T} = 1, \ldots, T$

$$\frac{\mathcal{T}}{\eta_{\mathcal{T}}}\mathbb{E}\|\boldsymbol{z}_{\mathcal{T}} - \boldsymbol{z}^*\|^2 = \frac{\mathcal{T}}{\eta_{\mathcal{T}}}(\mathcal{Q}_{\mathcal{T}} - \mathcal{Q}_{\mathcal{T}-1}) \leq \sqrt{\frac{1+\beta}{r}}M\mathcal{Q}_{\mathcal{T}-1} + \underbrace{\mathcal{R}_0 + (\tfrac{2}{r}L \vee B)\mathcal{Q}_0}_{\mathcal{D}_0}. \tag{37}$$

(37) is equivalently written as

$$\frac{\mathcal{T}}{\eta_{\mathcal{T}}}\mathcal{Q}_{\mathcal{T}} \leq \frac{\mathcal{T}+1}{\eta_{\mathcal{T}+1}}\mathcal{Q}_{\mathcal{T}-1} + \mathcal{D}_0.$$

From here and onwards, we denote $\kappa_t \equiv \frac{t}{\eta_t} = \frac{2}{r}L \vee B + \sqrt{\frac{1+\beta}{r}}Mt$ for each $t = 1, \ldots, T$. Dividing both sides of the above display by $\kappa_{\mathcal{T}}\kappa_{\mathcal{T}+1} = \frac{\mathcal{T}}{\eta_{\mathcal{T}}} \cdot \frac{\mathcal{T}+1}{\eta_{\mathcal{T}+1}}$ gives

$$\frac{\mathcal{Q}_{\mathcal{T}}}{\kappa_{\mathcal{T}+1}} \leq \frac{\mathcal{Q}_{\mathcal{T}-1}}{\kappa_{\mathcal{T}}} + \frac{\mathcal{D}_0}{\kappa_{\mathcal{T}} \cdot \kappa_{\mathcal{T}+1}}.$$

Telescoping up from $1, \ldots, \mathcal{T} - 1$ for $1 \leq \mathcal{T} \leq T$ yields

$$\frac{\mathcal{Q}_{\mathcal{T}-1}}{\kappa_{\mathcal{T}}} \leq \frac{\mathcal{Q}_0}{\kappa_1} + \sum_{t=1}^{\mathcal{T}-1} \frac{\mathcal{D}_0}{\kappa_t \cdot \kappa_{t+1}} \leq \frac{\mathcal{Q}_0}{\kappa_1} + \mathcal{D}_0 \sum_{t=1}^{\mathcal{T}-1} \frac{1}{\kappa_t \cdot \kappa_{t+1}},$$

where we applied Lemma D.3(ii) that for all $t = 1, \ldots, \mathcal{T} - 1$ we have $\kappa_{t+1} - \kappa_t = \sqrt{\frac{1+\beta}{r}} M$. This yields

$$\sqrt{\frac{1+\beta}{r}} M \sum_{t=1}^{\mathcal{T}-1} \frac{1}{\kappa_t \cdot \kappa_{t+1}} = \sum_{t=1}^{\mathcal{T}-1} \left[ \frac{1}{\kappa_t} - \frac{1}{\kappa_{t+1}} \right] = \frac{1}{\kappa_1} - \frac{1}{\kappa_{\mathcal{T}}},$$

and hence

$$\sqrt{\frac{1+\beta}{r}} M \frac{\mathcal{Q}_{\mathcal{T}-1}}{\kappa_{\mathcal{T}}} \leq \sqrt{\frac{1+\beta}{r}} M \frac{\mathcal{Q}_0}{\kappa_1} + \mathcal{D}_0 \sqrt{\frac{1+\beta}{r}} M \sum_{t=1}^{\mathcal{T}-1} \frac{1}{\kappa_t \cdot \kappa_{t+1}} = \sqrt{\frac{1+\beta}{r}} M \frac{\mathcal{Q}_0}{\kappa_1} + \mathcal{D}_0 \left( \frac{1}{\kappa_1} - \frac{1}{\kappa_{\mathcal{T}}} \right).$$

Next, we rearrange the above quantity and derive

$$\frac{\sqrt{\frac{1+\beta}{r}} M \mathcal{Q}_0 + \mathcal{D}_0}{\kappa_1} - \frac{\mathcal{D}_0}{\kappa_{\mathcal{T}}} = \frac{\sqrt{\frac{1+\beta}{r}} M \mathcal{Q}_0 + \left( \mathcal{R}_0 + (\frac{2}{r} L \vee B) \mathcal{Q}_0 \right)}{\kappa_1} - \frac{\mathcal{D}_0}{\kappa_{\mathcal{T}}} = \mathcal{Q}_0 + \frac{\mathcal{R}_0}{\kappa_1} - \frac{\mathcal{D}_0}{\kappa_{\mathcal{T}}}.$$

Plugging this into (37) we have for all iterates $1 \leq \mathcal{T} \leq T$

$$\mathbb{E}\|z_{\mathcal{T}} - z^*\|^2 \leq \sqrt{\frac{1+\beta}{r}} M \frac{\mathcal{Q}_{\mathcal{T}-1}}{\kappa_{\mathcal{T}}} + \frac{\mathcal{D}_0}{\kappa_{\mathcal{T}}} \leq \mathcal{Q}_0 + \frac{\mathcal{R}_0}{\kappa_1} \leq \left( 1 + \frac{C\sigma[T(T+1)^2]^{1/2}}{\kappa_1 \sqrt{\mathcal{Q}_0}} \right) \mathcal{Q}_0$$

$$= \underbrace{\left( 1 + C^2 B \eta_1 \right)}_{A} \mathbb{E}\|z_0 - z^*\|^2, \tag{38}$$

where the prefactor $A$ lies in $[1, 1 + C^2]$ and reduces to 1 when the argument is set as 0.

Now we drop the second summand on the left hand of (34) with $z^* = z^*$, $\mathcal{T} = T$. Combining with (38) gives

$$\mathcal{T}(\mathcal{T}+1)\mathbb{E}[V(z^{\mathrm{ag}}_{\mathcal{T}-\frac{1}{2}} \mid z^*)]$$

$$\leq \frac{1}{\eta_1} \mathbb{E}\|z_0 - z^*\|^2 + \sqrt{\frac{1+\beta}{r}} M \sum_{t=2}^{\mathcal{T}} \mathbb{E}\|z_{t-1} - z^*\|^2 + [T(T+1)^2]^{1/2} \cdot C\sigma \sqrt{\mathbb{E}\|z_0 - z^*\|^2}$$

$$\leq \left( \frac{2}{r} L \vee B + \sqrt{\frac{1+\beta}{r}} M \right) \mathbb{E}\|z_0 - z^*\|^2$$

$$+ \sqrt{\frac{1+\beta}{r}} M (T-1) \cdot A \cdot \mathbb{E}\|z_0 - z^*\|^2 + C\sigma[T(T+1)^2]^{1/2} \sqrt{\mathbb{E}\|z_0 - z^*\|^2}$$

$$\leq \left( \frac{2}{r} L + A \sqrt{\frac{1+\beta}{r}} M T \right) \mathbb{E}\|z_0 - z^*\|^2 + (\frac{1}{C} + C)\sigma[T(T+1)^2]^{1/2} \sqrt{\mathbb{E}\|z_0 - z^*\|^2}.$$

Using (29) in Lemma D.2 again lower bounds the left hand in the last display as

$$T(T+1)\mathbb{E}[V(z^{\mathrm{ag}}_{T-\frac{1}{2}} \mid z^*)] \geq \frac{\mu}{2} T(T+1)\mathbb{E}\left\| z^{\mathrm{ag}}_{T-\frac{1}{2}} - z^* \right\|^2 \geq 0.$$

Dividing both sides by $\frac{\mu}{2} T(T+1)$ concludes

$$\mathbb{E}\left\| z^{\mathrm{ag}}_{T-\frac{1}{2}} - z^* \right\|^2 \leq \frac{2 \left( \frac{2}{r} L + A \sqrt{\frac{1+\beta}{r}} M T \right)}{\mu T(T+1)} \mathbb{E}\|z_0 - z^*\|^2 + \frac{2(\frac{1}{C} + C)\sigma}{\mu T^{1/2}} \sqrt{\mathbb{E}\|z_0 - z^*\|^2},$$

and hence concludes (14) and the whole proof of Theorem 2.5.

## D.4 Proof of Theorem 2.3

We overload function notations $\mathcal{F}, \mathcal{H}$ to the new group accordingly where $\mathcal{F}(z) \leftarrow \mathcal{F}(z) - \frac{\mu_\star}{2}\|z - z_0\|^2$ is non-strongly convex and $\mathcal{H}(z) \leftarrow \mathcal{H}(z) + \mu_\star(z - z_0)$. For convenience we repeat the iterates of Algorithm 2 as

$$z_{t-\frac{1}{2}} = z_{t-1} - \eta_t \left( \nabla \tilde{\mathcal{F}}(z_{t-1}^{\mathrm{md}}; \xi_{t-\frac{1}{2}}) + \tilde{\mathcal{H}}(z_{t-1}; \zeta_{t-\frac{1}{2}}) - \mu(z_{t-1}^{\mathrm{md}} - z_{t-1}) \right),$$

$$z_{t-\frac{1}{2}}^{\mathrm{ag}} = (1 - \alpha_t)z_{t-\frac{3}{2}}^{\mathrm{ag}} + \alpha_t z_{t-\frac{1}{2}},$$

$$z_t = z_{t-1} - \eta_t \left( \nabla \tilde{\mathcal{F}}(z_{t-1}^{\mathrm{md}}; \xi_{t-\frac{1}{2}}) + \tilde{\mathcal{H}}(z_{t-\frac{1}{2}}; \zeta_t) - \mu(z_{t-1}^{\mathrm{md}} - z_{t-\frac{1}{2}}) \right),$$

$$z_t^{\mathrm{md}} = (1 - \alpha_{t+1})z_{t-\frac{1}{2}}^{\mathrm{ag}} + \alpha_{t+1}z_t,$$

with the initialization $z_0 = z_0^{\mathrm{md}} = z_{-\frac{1}{2}}^{\mathrm{ag}} \in \mathbb{R}^{n+m}$. We continue to assume the noise-related setting as in (31). Our proof proceeds in the following steps:

**Step 1.** We prove the following generalization of Lemma D.4, whose proof is in §E.6:

**Lemma D.6** *For arbitrary* $\tilde{z} \in \mathbb{R}^{n+m}$ *and* $\alpha_t \in (0, 1]$ *the iterates of Algorithm 2 satisfy almost surely*

$$V(z_{t-\frac{1}{2}}^{\mathrm{ag}} \mid \tilde{z}) - (1 - \alpha_t)V(z_{t-\frac{3}{2}}^{\mathrm{ag}} \mid \tilde{z})$$

$$\leq \alpha_t \langle \nabla \mathcal{F}(z_{t-1}^{\mathrm{md}}) + \mathcal{H}(z_{t-\frac{1}{2}}), z_{t-\frac{1}{2}} - \tilde{z} \rangle + \frac{\alpha_t^2 L}{2} \left\| z_{t-\frac{1}{2}} - z_{t-1} \right\|^2 - \alpha_t \mu_\star \left\| z_{t-\frac{1}{2}} - \tilde{z} \right\|^2.$$
$$\tag{39}$$

**Step 2.** Analogous to Step 2 in the proof of Theorem 2.5 in §D.3 we conclude for all $z \in \mathbb{R}^n$, $z \in \mathbb{R}^m$,

$$\eta_t \mathbb{E}\langle \nabla \tilde{\mathcal{F}}(z_{t-1}^{\mathrm{md}}; \xi_{t-\frac{1}{2}}) + \tilde{\mathcal{H}}(z_{t-\frac{1}{2}}; \zeta_t), z_{t-\frac{1}{2}} - \tilde{z} \rangle$$

$$\leq \frac{1}{2} \left( \mathbb{E}[\|z_{t-1} - \tilde{z}\|^2] - \mathbb{E}[\|z_t - \tilde{z}\|^2] \right) - \frac{1 - (1 + \beta)M^2\eta_t^2}{2} \mathbb{E}\left[ \left\| z_{t-\frac{1}{2}} - z_{t-1} \right\|^2 \right]$$

$$+ \frac{\eta_t^2}{2}(2 + \frac{1}{\beta})\sigma_{\mathrm{Bil}}^2.$$

To show this, note that

$$\eta_t \langle \nabla \tilde{\mathcal{F}}(z_{t-1}^{\mathrm{md}}; \xi_{t-\frac{1}{2}}) + \tilde{\mathcal{H}}(z_{t-\frac{1}{2}}; \zeta_t), z_{t-\frac{1}{2}} - \tilde{z} \rangle$$

$$\leq \frac{1}{2} \left( \|z_{t-1} - \tilde{z}\|^2 - \|z_t - \tilde{z}\|^2 - \|z_{t-\frac{1}{2}} - z_{t-1}\|^2 \right) + \frac{\eta_t^2}{2}\|\tilde{\mathcal{H}}(z_{t-\frac{1}{2}}; \zeta_t) - \tilde{\mathcal{H}}(z_{t-1}; \zeta_{t-\frac{1}{2}})\|^2.$$

To handle the stochastic terms, Young's inequality combined with the martingale structure, along with the definition of $M$, indicates

$$\mathbb{E}\left\| \tilde{\mathcal{H}}(z_{t-\frac{1}{2}}; \zeta_t) - \tilde{\mathcal{H}}(z_{t-1}; \zeta_{t-\frac{1}{2}}) \right\|^2 = \mathbb{E}\left\| \mathcal{H}(z_{t-\frac{1}{2}}) - \mathcal{H}(z_{t-1}) - \mathbf{\Delta}_{\mathcal{H}}^{t-\frac{1}{2}} \right\|^2 + \mathbb{E}\left\| \mathbf{\Delta}_{\mathcal{H}}^t \right\|^2$$

$$\leq (1 + \beta)M^2\mathbb{E}\|z_{t-\frac{1}{2}} - z_{t-1}\|^2 + (1 + \frac{1}{\beta})\mathbb{E}\left\| \mathbf{\Delta}_{\mathcal{H}}^{t-\frac{1}{2}} \right\|^2 + \mathbb{E}\|\mathbf{\Delta}_{\mathcal{H}}^t\|^2.$$

Combining the last three displays gives

$$\eta_t \mathbb{E}\langle \nabla \tilde{\mathcal{F}}(z_{t-1}^{\mathrm{md}}; \xi_{t-\frac{1}{2}}) + \tilde{\mathcal{H}}(z_{t-\frac{1}{2}}; \zeta_t), z_{t-\frac{1}{2}} - \tilde{z} \rangle$$

$$\leq \frac{1}{2} \left( \mathbb{E}[\|z_{t-1} - \tilde{z}\|^2] - \mathbb{E}[\|z_t - \tilde{z}\|^2] \right) - \frac{1 - (1 + \beta)M^2\eta_t^2}{2} \mathbb{E}\left[ \left\| z_{t-\frac{1}{2}} - z_{t-1} \right\|^2 \right]$$

$$+ \frac{\eta_t^2}{2} \left( (1 + \frac{1}{\beta})\mathbb{E}\left\| \mathbf{\Delta}_{\mathcal{H}}^{t-\frac{1}{2}} \right\|^2 + \mathbb{E}\|\mathbf{\Delta}_{\mathcal{H}}^t\|^2 \right).$$
$$\tag{40}$$

Combining this with Lemma D.6, we have

$$\mathbb{E}[V(\boldsymbol{z}_{t-\frac{1}{2}}^{\mathrm{ag}} \mid \tilde{\boldsymbol{z}})] - (1-\alpha_t)\mathbb{E}[V(\boldsymbol{z}_{t-\frac{3}{2}}^{\mathrm{ag}} \mid \tilde{\boldsymbol{z}})]$$

$$\leq \alpha_t \mathbb{E}\langle \nabla \mathcal{F}(\boldsymbol{z}_{t-1}^{\mathrm{md}}) + \mathcal{H}(\boldsymbol{z}_{t-\frac{1}{2}}), \boldsymbol{z}_{t-\frac{1}{2}} - \tilde{\boldsymbol{z}}\rangle + \frac{\alpha_t^2 L}{2}\mathbb{E}\left[\left\|\boldsymbol{z}_{t-\frac{1}{2}} - \boldsymbol{z}_{t-1}\right\|^2\right] - \alpha_t \mu_\star \mathbb{E}\left[\left\|\boldsymbol{z}_{t-\frac{1}{2}} - \tilde{\boldsymbol{z}}\right\|^2\right]$$

$$= \alpha_t \mathbb{E}\langle \nabla \tilde{\mathcal{F}}(\boldsymbol{z}_{t-1}^{\mathrm{md}}; \xi_{t-\frac{1}{2}}) + \tilde{\mathcal{H}}(\boldsymbol{z}_{t-\frac{1}{2}}; \zeta_t), \boldsymbol{z}_{t-\frac{1}{2}} - \tilde{\boldsymbol{z}}\rangle$$

$$- \alpha_t \mathbb{E}\langle \boldsymbol{\Delta}_{\mathcal{F}}^{t-\frac{1}{2}} + \boldsymbol{\Delta}_{\mathcal{H}}^{t}, \boldsymbol{z}_{t-\frac{1}{2}} - \tilde{\boldsymbol{z}}\rangle + \frac{\alpha_t^2 L}{2}\mathbb{E}\left[\left\|\boldsymbol{z}_{t-\frac{1}{2}} - \boldsymbol{z}_{t-1}\right\|^2\right] - \alpha_t \mu_\star \mathbb{E}\left[\left\|\boldsymbol{z}_{t-\frac{1}{2}} - \tilde{\boldsymbol{z}}\right\|^2\right]$$

$$\leq \frac{\alpha_t}{\eta_t}\left(\frac{1}{2}\left(\mathbb{E}[\|\boldsymbol{z}_{t-1} - \tilde{\boldsymbol{z}}\|^2] - \mathbb{E}[\|\boldsymbol{z}_t - \tilde{\boldsymbol{z}}\|^2]\right) - \frac{1-(1+\beta)M^2\eta_t^2}{2}\mathbb{E}\left[\left\|\boldsymbol{z}_{t-\frac{1}{2}} - \boldsymbol{z}_{t-1}\right\|^2\right]\right.$$

$$\left. + \frac{\eta_t^2}{2}(2+\tfrac{1}{\beta})\sigma_{\mathrm{Bil}}^2\right) - \alpha_t \mathbb{E}\langle \boldsymbol{\Delta}_{\mathcal{F}}^{t-\frac{1}{2}} + \boldsymbol{\Delta}_{\mathcal{H}}^{t}, \boldsymbol{z}_{t-\frac{1}{2}} - \tilde{\boldsymbol{z}}\rangle + \frac{\alpha_t^2 L}{2}\mathbb{E}\left[\left\|\boldsymbol{z}_{t-\frac{1}{2}} - \boldsymbol{z}_{t-1}\right\|^2\right]$$

$$- \alpha_t \mu_\star \mathbb{E}\left[\left\|\boldsymbol{z}_{t-\frac{1}{2}} - \tilde{\boldsymbol{z}}\right\|^2\right].$$

Continuing this estimation gives (note Young's inequality applies, and $\mathbb{E}\langle \boldsymbol{\Delta}_{\mathcal{F}}^{t-\frac{1}{2}} + \boldsymbol{\Delta}_{\mathcal{H}}^{t}, \boldsymbol{z}_{t-\frac{1}{2}} - \tilde{\boldsymbol{z}}\rangle = \mathbb{E}\langle \boldsymbol{\Delta}_{\mathcal{F}}^{t-\frac{1}{2}}, \boldsymbol{z}_{t-\frac{1}{2}} - \boldsymbol{z}_{t-1}\rangle$)

$$\mathbb{E}[V(\boldsymbol{z}_{t-\frac{1}{2}}^{\mathrm{ag}} \mid \tilde{\boldsymbol{z}})] - (1-\alpha_t)\mathbb{E}[V(\boldsymbol{z}_{t-\frac{3}{2}}^{\mathrm{ag}} \mid \tilde{\boldsymbol{z}})]$$

$$\leq \frac{\alpha_t}{2\eta_t}\left(\mathbb{E}[\|\boldsymbol{z}_{t-1} - \tilde{\boldsymbol{z}}\|^2] - \mathbb{E}[\|\boldsymbol{z}_t - \tilde{\boldsymbol{z}}\|^2]\right) - \frac{\alpha_t}{2\eta_t}\left(r - \alpha_t L\eta_t - (1+\beta)M^2\eta_t^2\right)\mathbb{E}\left[\left\|\boldsymbol{z}_{t-\frac{1}{2}} - \boldsymbol{z}_{t-1}\right\|^2\right]$$

$$+ \frac{\alpha_t \eta_t}{2}(2+\tfrac{1}{\beta})\sigma_{\mathrm{Bil}}^2 - \frac{\alpha_t(1-r)}{2\eta_t}\mathbb{E}\left[\left\|\boldsymbol{z}_{t-\frac{1}{2}} - \boldsymbol{z}_{t-1}\right\|^2\right]$$

$$- \alpha_t \mathbb{E}\langle \boldsymbol{\Delta}_{\mathcal{F}}^{t-\frac{1}{2}}, \boldsymbol{z}_{t-\frac{1}{2}} - \boldsymbol{z}_{t-1}\rangle - \alpha_t \mu_\star \mathbb{E}\left[\left\|\boldsymbol{z}_{t-\frac{1}{2}} - \tilde{\boldsymbol{z}}\right\|^2\right]$$

$$\leq \frac{\alpha_t}{2\eta_t}\left(\mathbb{E}[\|\boldsymbol{z}_{t-1} - \tilde{\boldsymbol{z}}\|^2] - \mathbb{E}[\|\boldsymbol{z}_t - \tilde{\boldsymbol{z}}\|^2]\right) - \frac{\alpha_t}{2\eta_t}\left(r - \alpha_t L\eta_t - (1+\beta)M^2\eta_t^2\right)\mathbb{E}\left[\left\|\boldsymbol{z}_{t-\frac{1}{2}} - \boldsymbol{z}_{t-1}\right\|^2\right]$$

$$+ \frac{\alpha_t \eta_t}{2}(2+\tfrac{1}{\beta})\sigma_{\mathrm{Bil}}^2 + \frac{\alpha_t \eta_t}{2(1-r)}\mathbb{E}\left\|\boldsymbol{\Delta}_{\mathcal{F}}^{t-\frac{1}{2}}\right\|^2 - \alpha_t \mu_\star \mathbb{E}\left[\left\|\boldsymbol{z}_{t-\frac{1}{2}} - \tilde{\boldsymbol{z}}\right\|^2\right]$$

$$\leq \frac{\alpha_t}{2\eta_t}\left(\mathbb{E}[\|\boldsymbol{z}_{t-1} - \tilde{\boldsymbol{z}}\|^2] - \mathbb{E}[\|\boldsymbol{z}_t - \tilde{\boldsymbol{z}}\|^2]\right) - \frac{\alpha_t}{2\eta_t}\left(r - \alpha_t L\eta_t - (1+\beta)M^2\eta_t^2\right)\mathbb{E}\left[\left\|\boldsymbol{z}_{t-\frac{1}{2}} - \boldsymbol{z}_{t-1}\right\|^2\right]$$

$$- \alpha_t \mu_\star \mathbb{E}\left[\left\|\boldsymbol{z}_{t-\frac{1}{2}} - \tilde{\boldsymbol{z}}\right\|^2\right] + \frac{\alpha_t \eta_t}{2}\left(\tfrac{1}{1-r}\sigma_{\mathrm{Str}}^2 + (2+\tfrac{1}{\beta})\sigma_{\mathrm{Bil}}^2\right).$$

By applying Young's inequality, it yields

$$\mathbb{E}[V(\boldsymbol{z}_{t-\frac{1}{2}}^{\mathrm{ag}} \mid \tilde{\boldsymbol{z}})] - (1-\alpha_t)\mathbb{E}[V(\boldsymbol{z}_{t-\frac{3}{2}}^{\mathrm{ag}} \mid \tilde{\boldsymbol{z}})] - \frac{\alpha_t \eta_t}{2}\left(\tfrac{1}{1-r}\sigma_{\mathrm{Str}}^2 + (2+\tfrac{1}{\beta})\sigma_{\mathrm{Bil}}^2\right)$$

$$+ \frac{\alpha_t}{2\eta_t}\left(r - \alpha_t L\eta_t - (1+\beta)M^2\eta_t^2\right)\mathbb{E}\left[\left\|\boldsymbol{z}_{t-\frac{1}{2}} - \boldsymbol{z}_{t-1}\right\|^2\right]$$

$$\leq \frac{\alpha_t}{2\eta_t}\left(\mathbb{E}[\|\boldsymbol{z}_{t-1} - \tilde{\boldsymbol{z}}\|^2] - \mathbb{E}[\|\boldsymbol{z}_t - \tilde{\boldsymbol{z}}\|^2]\right) - \alpha_t \mu_\star \mathbb{E}\left[\left\|\boldsymbol{z}_{t-\frac{1}{2}} - \tilde{\boldsymbol{z}}\right\|^2\right]$$

$$\leq \frac{\alpha_t}{2\eta_t}\left((1-\alpha_t)\mathbb{E}[\|\boldsymbol{z}_{t-1} - \tilde{\boldsymbol{z}}\|^2] - \mathbb{E}[\|\boldsymbol{z}_t - \tilde{\boldsymbol{z}}\|^2]\right) + \frac{\alpha_t^2}{2\eta_t}\mathbb{E}[\|\boldsymbol{z}_{t-1} - \tilde{\boldsymbol{z}}\|^2] - \alpha_t \mu_\star \mathbb{E}\left[\left\|\boldsymbol{z}_{t-\frac{1}{2}} - \tilde{\boldsymbol{z}}\right\|^2\right]$$

$$\leq \frac{\alpha_t}{2\eta_t}\left((1-\alpha_t)\mathbb{E}[\|\boldsymbol{z}_{t-1} - \tilde{\boldsymbol{z}}\|^2] - \mathbb{E}[\|\boldsymbol{z}_t - \tilde{\boldsymbol{z}}\|^2]\right) + \eta_t \mu_\star^2 \mathbb{E}\left[\left\|\boldsymbol{z}_{t-\frac{1}{2}} - \boldsymbol{z}_{t-1}\right\|^2\right].$$

Setting $\eta_t = \frac{\alpha_t}{\mu_\star}$ we have

$$\mathbb{E}[V(\boldsymbol{z}_{t-\frac{1}{2}}^{\mathrm{ag}} \mid \tilde{\boldsymbol{z}})] + \frac{\mu_\star}{2}\mathbb{E}[\|\boldsymbol{z}_t - \tilde{\boldsymbol{z}}\|^2] - (1-\alpha_t)\left(\mathbb{E}[V(\boldsymbol{z}_{t-\frac{3}{2}}^{\mathrm{ag}} \mid \tilde{\boldsymbol{z}})] + \frac{\mu_\star}{2}\mathbb{E}[\|\boldsymbol{z}_{t-1} - \tilde{\boldsymbol{z}}\|^2]\right)$$

$$\leq -\frac{\mu_\star}{2}\left(r - 2\alpha_t - \left(\frac{L}{\mu_\star} + \frac{(1+\beta)M^2}{\mu_\star^2}\right)\alpha_t^2\right)\mathbb{E}\left[\left\|\boldsymbol{z}_{t-\frac{1}{2}} - \boldsymbol{z}_{t-1}\right\|^2\right]$$

$$+ \frac{\alpha_t^2}{2\mu_\star}\left(\tfrac{1}{1-r}\sigma_{\mathrm{Str}}^2 + (2+\tfrac{1}{\beta})\sigma_{\mathrm{Bil}}^2\right).$$

**Step 3.** By the definition $\alpha_t$ we have $r - 2\alpha_t - \left(\tfrac{L}{\mu_\star} + \tfrac{(1+\beta)M^2}{\mu_\star^2}\right)\alpha_t^2 \geq 0$, so we obtain regularity condition $\alpha_t \leq \bar{\alpha} = \dfrac{r}{1+\sqrt{1+r\left(\tfrac{L}{\mu_\star} + \tfrac{(1+\beta)M^2}{\mu_\star^2}\right)}}$ of Theorem 2.3. Since we assumed both $F$ and $G$ are nonstrongly convex and $H$ is a $\mu_\star$-strongly-convex-$\mu_\star$-strongly-concave isotropic quadratic, this implies

$$\mathbb{E}[V(z_{t-\frac{1}{2}}^{\mathrm{ag}} \mid \tilde{z})] + \frac{\mu_\star}{2}\mathbb{E}[\|z_t - \tilde{z}\|^2] \leq (1-\alpha_t)\left(\mathbb{E}[V(z_{t-\frac{3}{2}}^{\mathrm{ag}} \mid \tilde{z})] + \frac{\mu_\star}{2}\mathbb{E}[\|z_{t-1} - \tilde{z}\|^2]\right) + \frac{3\alpha_t^2}{2\mu_\star}\sigma^2.$$

Plugging in $\tilde{z} = z^*$ gives
$$\mathbb{E}[V(\tilde{z} \mid z^*)] = \mathcal{F}(\tilde{z}) - \mathcal{F}(z^*) + \langle \mathcal{H}(z^*), \tilde{z} - z^*\rangle \geq \langle \nabla\mathcal{F}(z^*) + \mathcal{H}(z^*), \tilde{z} - z^*\rangle = 0,$$

and also
$$\mathbb{E}[V(\tilde{z} \mid z^*)] \leq \langle \nabla\mathcal{F}(z^*) + \mathcal{H}(z^*), \tilde{z} - z^*\rangle + \tfrac{L}{2}\|\tilde{z} - z^*\|^2 = \tfrac{L}{2}\|\tilde{z} - z^*\|^2,$$

so (by the fact that $z_{-\frac{1}{2}}^{\mathrm{ag}} = z_0$ and $z_{-\frac{1}{2}}^{\mathrm{ag}} = z_0$)

$$\frac{\mu_\star}{2}\mathbb{E}[\|z_t - z^*\|^2] \leq \mathbb{E}[V(z_{t-\frac{1}{2}}^{\mathrm{ag}} \mid z^*)] + \frac{\mu_\star}{2}\mathbb{E}[\|z_t - z^*\|^2]$$

$$\leq \left(V(z_{-\frac{1}{2}}^{\mathrm{ag}} \mid z^*) + \frac{\mu_\star}{2}\|z_0 - z^*\|^2\right)\prod_{\tau=1}^{t}(1-\alpha_\tau) + \sum_{\tau=1}^{t}\frac{3\alpha_\tau^2}{2\mu_\star}\left[\prod_{\tau'=\tau+1}^{t}(1-\alpha_{\tau'})\right]\sigma^2$$

$$\leq \|z_0 - z^*\|^2\frac{L+\mu_\star}{2}\prod_{\tau=1}^{t}(1-\alpha_\tau) + \frac{3\sigma^2}{2\mu_\star}\sum_{\tau=1}^{t}\alpha_\tau^2\prod_{\tau'=\tau+1}^{t}(1-\alpha_{\tau'}).$$

Dividing both sides by $\frac{\mu_\star}{2}$ gives (11) and our theorem.

### D.5 Proof of Corollary 2.9

The proof of Corollary 2.9 mostly follows the proof of Theorem 2.5 and Corollary 2.8, except that we modify some steps to adapt to the proximal operator. The proof is as follows:

**Step 1.** Estimating the primal-dual gap function difference sequence. We have the following Lemma (D.7), whose proof is in §E.7:

**Lemma D.7** *For arbitrary $\tilde{z} \in \mathcal{Z}$ the iterates of Algorithm 1 satisfy for $t = 1, \ldots, T$, almost surely*

$$V(z_{t-\frac{1}{2}}^{\mathrm{ag}} \mid \tilde{z}) - (1-\alpha_t)V(z_{t-\frac{3}{2}}^{\mathrm{ag}} \mid \tilde{z})$$

$$\leq \alpha_t\langle \nabla\mathcal{F}(z_{t-1}^{\mathrm{md}}) + \mathcal{H}(z_{t-\frac{1}{2}}), z_{t-\frac{1}{2}} - \tilde{z}\rangle + \tfrac{\alpha_t^2 L}{2}\left\|z_{t-\frac{1}{2}} - z_{t-1}\right\|^2 + \alpha_t\left(J(z_{t-\frac{1}{2}}) - J(\tilde{z})\right). \tag{41}$$

Note the proof only relies on the interpolation updates in our algorithm as in Lines 6 and 8, and hence this result holds in a per-trajectory (almost-sure) fashion.

**Step 2.** We target to prove the following lemma, the complete proof is in §E.8:

**Lemma D.8** *For our choice of $\eta_t$ that satisfies, for a given $r \in (0,1)$, that*

$$r\mu_{\mathcal{B}} - \frac{2L}{t+1}\eta_t - \frac{(1+\beta)M^2\eta_t^2}{\mu_{\mathcal{B}}} \geq 0, \tag{42}$$

*we have for any $\tilde{z} \in \mathbb{R}^n$, $\tilde{y} \in \mathbb{R}^m$ and $t = 1, \ldots, T$ that*

$$t(t+1)\mathbb{E}[V(z_{t-\frac{1}{2}}^{\mathrm{ag}} \mid \tilde{z})] - (t-1)t\mathbb{E}[V(z_{t-\frac{3}{2}}^{\mathrm{ag}} \mid \tilde{z})]$$

$$\leq \frac{2t}{\eta_t}\left(\mathbb{E}[\mathcal{B}(z_{t-1}, \tilde{z})] - \mathbb{E}[\mathcal{B}(z_t, \tilde{z})]\right) + \left(\tfrac{1}{1-r}\sigma_{\mathrm{Str}}^2 + (2+\tfrac{1}{\beta})\sigma_{\mathrm{Bil}}^2\right)\frac{t\eta_t}{\mu_{\mathcal{B}}}, \tag{43}$$

We note that (43) in Lemma D.8 only differs with (33) in Lemma D.5 by the use of Bregman distance $\mathcal{B}$ and a factor of $1/\mu_{\mathcal{B}}$ on the variance term. Following similar derivations as in the proof of Theorem 2.5, we telescope the above recursion for $t = 1, \ldots, \mathcal{T}$ and choose the step size as

$$\mu_{\mathcal{B}} \cdot \tfrac{t}{\eta_t} = \tfrac{2}{r}L \vee B + \sqrt{\tfrac{1+\beta}{r}}Mt, \tag{44}$$

with $B = \frac{\sigma\sqrt{T(T+1)}}{C\sqrt{\frac{2}{\mu_{\mathcal{B}}}\mathcal{B}(z_0,\tilde{z})}}$ that satisfies (42). By denoting $\sigma \equiv \frac{1}{\sqrt{3}}\sqrt{\frac{1}{1-r}\sigma_{\mathrm{Str}}^2 + (2+\frac{1}{\beta})\sigma_{\mathrm{Bil}}^2}$, we have by Lemma D.3(i) and the same derivative as in §D.3

$$\left(\tfrac{1}{1-r}\sigma_{\mathrm{Str}}^2 + (2+\tfrac{1}{\beta})\sigma_{\mathrm{Bil}}^2\right)\sum_{t=1}^{\mathcal{T}} t\eta_t \leq 3\sigma^2 \cdot \frac{\mu_{\mathcal{B}}}{B}\sum_{t=1}^{\mathcal{T}} t^2$$

$$= \frac{\mathcal{T}(\mathcal{T}+\frac{1}{2})(\mathcal{T}+1)}{[T(T+1)^2]^{1/2}} \cdot \sigma\mu_{\mathcal{B}}C\sqrt{\frac{2}{\mu_{\mathcal{B}}}\mathbb{E}\mathcal{B}(z_0,\tilde{z})},$$

Finally by summing over $t = 1, \ldots, \mathcal{T}$, we have

$$\mathcal{T}(\mathcal{T}+1)\mathbb{E}[V(z_{\mathcal{T}-\frac{1}{2}}^{\mathrm{ag}} \mid \tilde{z})]$$

$$\leq \sum_{t=1}^{\mathcal{T}} \frac{2t}{\eta_t}\left(\mathbb{E}[\mathcal{B}(z_{t-1},\tilde{z})] - \mathbb{E}[\mathcal{B}(z_t,\tilde{z})]\right) + \left(\tfrac{1}{1-r}\sigma_{\mathrm{Str}}^2 + (2+\tfrac{1}{\beta})\sigma_{\mathrm{Bil}}^2\right)\sum_{t=1}^{\mathcal{T}}\frac{t\eta_t}{\mu_{\mathcal{B}}}$$

$$= \frac{2}{\eta_1}\mathcal{B}(z_0,\tilde{z}) + 2\sum_{t=2}^{\mathcal{T}}\left(\frac{t}{\eta_t} - \frac{t-1}{\eta_{t-1}}\right)\mathbb{E}\mathcal{B}(z_{t-1},\tilde{z}) - \frac{2\mathcal{T}}{\eta_{\mathcal{T}}}\mathbb{E}\mathcal{B}(z_{\mathcal{T}},\tilde{z})$$

$$+ \frac{\mathcal{T}(\mathcal{T}+\frac{1}{2})(\mathcal{T}+1)}{[T(T+1)^2]^{1/2}} \cdot C\sigma\sqrt{\frac{2}{\mu_{\mathcal{B}}}\mathbb{E}\mathcal{B}(z_0,\tilde{z})}$$

$$= \frac{2}{\eta_1}\mathcal{B}(z_0,\tilde{z}) + 2\sqrt{\tfrac{1+\beta}{r}}\frac{M}{\mu_{\mathcal{B}}}\sum_{t=2}^{\mathcal{T}}\mathbb{E}\mathcal{B}(z_{t-1},\tilde{z}) - \frac{2\mathcal{T}}{\eta_{\mathcal{T}}}\mathbb{E}\mathcal{B}(z_{\mathcal{T}},\tilde{z})$$

$$+ [T(T+1)^2]^{1/2} \cdot C\sigma\sqrt{\frac{2}{\mu_{\mathcal{B}}}\mathbb{E}\mathcal{B}(z_0,\tilde{z})}.$$

Rearranging the terms proves the following inequality (45).

$$\mathcal{T}(\mathcal{T}+1)\mathbb{E}[V(z_{\mathcal{T}-\frac{1}{2}}^{\mathrm{ag}} \mid \tilde{z})] + \frac{2\mathcal{T}}{\eta_{\mathcal{T}}}\mathbb{E}\mathcal{B}(z_{\mathcal{T}},\tilde{z})$$

$$\leq \frac{2}{\eta_1}\mathcal{B}(z_0,\tilde{z}) + 2\sqrt{\tfrac{1+\beta}{r}}\frac{M}{\mu_{\mathcal{B}}}\sum_{t=2}^{\mathcal{T}}\mathbb{E}\mathcal{B}(z_{t-1},\tilde{z}) + [T(T+1)^2]^{1/2} \cdot C\sigma\sqrt{\frac{2}{\mu_{\mathcal{B}}}\mathbb{E}\mathcal{B}(z_0,\tilde{z})}. \tag{45}$$

The same bootstrapping argument gives

$$\mathbb{E}\mathcal{B}(z_{t-1},z^*) \leq \left(1 + C^2 B\eta_1\right)\mathbb{E}\mathcal{B}(z_0,z^*),$$

which further derives

$$\mathcal{T}(\mathcal{T}+1)\mathbb{E}[V(z_{\mathcal{T}-\frac{1}{2}}^{\mathrm{ag}} \mid z^*)] \leq \frac{2}{\mu_{\mathcal{B}}}\left(\tfrac{2}{r}L + A\sqrt{\tfrac{1+\beta}{r}}MT\right)\mathbb{E}\mathcal{B}(z_0,z^*)$$

$$+ (\tfrac{1}{C} + C)\sigma[T(T+1)^2]^{1/2}\sqrt{\frac{2}{\mu_{\mathcal{B}}}\mathbb{E}\mathcal{B}(z_0,z^*)}$$

Again we can lower bounds the left hand in the last display as

$$T(T+1)\mathbb{E}[V(z_{T-\frac{1}{2}}^{\mathrm{ag}} \mid z^*)] \geq \frac{\mu}{2} \cdot T(T+1)\mathbb{E}\mathcal{B}\left(z_{T-\frac{1}{2}}^{\mathrm{ag}}, z^*\right) \geq 0.$$

Dividing both sides by $\frac{\mu}{2}T(T+1)$ concludes

$$\mathbb{E}\mathcal{B}\left(z_{T-\frac{1}{2}}^{\mathrm{ag}}, z^*\right) \leq \frac{4\left(\tfrac{2}{r}L + A\sqrt{\tfrac{1+\beta}{r}}MT\right)}{\mu\mu_{\mathcal{B}}T(T+1)}\mathbb{E}\mathcal{B}(z_0,z^*) + \frac{2(\tfrac{1}{C}+C)\sigma}{\mu T^{1/2}}\sqrt{\frac{2}{\mu_{\mathcal{B}}}\mathbb{E}\mathcal{B}(z_0,z^*)},$$

The rest of the proof follows the same bounded iterates argument and the restarting argument exactly as in the previous proof of Theorem 2.5 with only a difference in a factor of $\mu_{\mathcal{B}}$. Similar derivatives gives us a total iteration complexity of

$$\mathcal{O}\left(\left(\sqrt{\frac{L}{\mu\mu_{\mathcal{B}}}} + \frac{M}{\mu\mu_{\mathcal{B}}}\right)\log\left(\frac{1}{\varepsilon}\right) + \frac{\sigma^2\mathcal{B}(\boldsymbol{z}_0,\boldsymbol{z}^*)}{\mu^2\mu_{\mathcal{B}}\varepsilon^2}\right)$$

with epoch length $T_s \asymp \sqrt{\frac{L}{\mu\mu_{\mathcal{B}}}} + \frac{M}{\mu\mu_{\mathcal{B}}} + \frac{\sigma^2\mathcal{B}(\boldsymbol{z}_0,\boldsymbol{z}^*)}{\mu^2\mu_{\mathcal{B}}\Gamma_0^2 e^{1-s}}$.

### D.6 Proof of Corollary 3.3

Before the proof we first adopt the scaling reduction argument as in §D.1, to argue that we only need to prove the result for the case of bilinear games centered at zero, i.e., $F(\boldsymbol{x}) = 0 = G(\boldsymbol{y})$ we have $L = \mu = \mu_F = 0$. We set the iteration symbol $\mathbf{z} \equiv \begin{bmatrix} \hat{\boldsymbol{x}} \\ \hat{\boldsymbol{y}} \end{bmatrix} = \begin{bmatrix} \boldsymbol{x} - \boldsymbol{z}^* \\ \boldsymbol{y} - \boldsymbol{\omega}_{\boldsymbol{y}}^\star \end{bmatrix}$ and also $\hat{\mathscr{F}}(\hat{\boldsymbol{x}}, \hat{\boldsymbol{y}}) = \hat{\boldsymbol{x}}^\top \mathbf{B}\hat{\boldsymbol{y}}$, with $\hat{\mathscr{F}}(\hat{\boldsymbol{x}}, \hat{\boldsymbol{y}})$ being equal to $\mathscr{F}(\boldsymbol{x}, \boldsymbol{y})$ defined as in (20) up to an additive constant. Our scaling-reduction argument hence applies.

*Proof.*[Proof of Corollary 3.3] From the update rule we have

$$\mathbf{z}_{t-\frac{1}{2}} = \mathbf{z}_{t-1} - \eta\mathbf{J}\mathbf{z}_{t-1} + \eta\varepsilon_{t-\frac{1}{2}}, \tag{46a}$$

$$\mathbf{z}_{t-\frac{1}{2}}^{\text{ag}} = \frac{t-1}{t+1}\mathbf{z}_{t-\frac{3}{2}}^{\text{ag}} + \frac{2}{t+1}\mathbf{z}_{t-\frac{1}{2}}, \tag{46b}$$

$$\mathbf{z}_t = \mathbf{z}_{t-1} - \eta\mathbf{J}\mathbf{z}_{t-\frac{1}{2}} + \eta\varepsilon_t. \tag{46c}$$

Note the $[\boldsymbol{x}_t^{\text{md}}; \boldsymbol{y}_t^{\text{md}}]$ sequence becomes irrelevant in this update; $\mathbf{J} \equiv \begin{bmatrix} \mathbf{0} & \mathbf{B} \\ -\mathbf{B}^\top & \mathbf{0} \end{bmatrix}$ is skew-symmetric with $\mathbf{J}^\top = -\mathbf{J}$, so $\mathbf{J}^2 = -\mathbf{J}^\top\mathbf{J}$ is symmetric and negative semidefinite. We proceed with the proof in steps:

**Step 1.** We target to show the last-iterate bound

$$\mathbb{E}\|\mathbf{z}_t\|^2 \le \mathbb{E}\|\mathbf{z}_0\|^2 + 2t\eta^2\sigma_{\text{Bil}}^2 \tag{47}$$

Note (46a) and (46c) together gives

$$\mathbf{z}_t = \left(\mathbf{I} - \eta\mathbf{J} + \eta^2\mathbf{J}^2\right)\mathbf{z}_{t-1} - \eta^2\mathbf{J}\varepsilon_{t-\frac{1}{2}} + \eta\varepsilon_t \tag{48}$$

Taking squared norm on both sides of (48), we have when $\eta \le \frac{1}{\sqrt{\lambda_{\max}(\mathbf{B}^\top\mathbf{B})}}$, $\mathbf{z}_t$ does not expand in Euclidean norm (noiseless), so

$$\mathbb{E}\|\mathbf{z}_t\|^2 = \mathbb{E}\left[(\mathbf{z}_{t-1})^\top\left(\mathbf{I} + \eta^2\mathbf{J}^2 + \eta^4\mathbf{J}^4\right)\mathbf{z}_{t-1}\right] + \mathbb{E}\left\|-\eta^2\mathbf{J}\varepsilon_{t-\frac{1}{2}} + \eta\varepsilon_t\right\|^2$$

$$\le \mathbb{E}\|\mathbf{z}_{t-1}\|^2 + \mathbb{E}\left\|\eta^2\mathbf{J}\varepsilon_{t-\frac{1}{2}}\right\|^2 + \mathbb{E}\|\eta\varepsilon_t\|^2 \tag{49}$$

$$\le \mathbb{E}\|\mathbf{z}_{t-1}\|^2 + \eta^2\left(1 + \eta^2\lambda_{\max}(\mathbf{B}^\top\mathbf{B})\right)\sigma_{\text{Bil}}^2 \le \mathbb{E}\|\mathbf{z}_{t-1}\|^2 + 2\eta^2\sigma_{\text{Bil}}^2.$$

Recursively applying the above concludes (47).

**Step 2.** We start from the update rule (46b) which implies $(t+1)t\mathbf{z}_{t-\frac{1}{2}}^{\text{ag}} = t(t-1)\mathbf{z}_{t-\frac{3}{2}}^{\text{ag}} + 2t\mathbf{z}_{t-\frac{1}{2}}$ holds for $t = 1, \ldots, T$, so

$$(T+1)T\mathbf{z}_{T-\frac{1}{2}}^{\text{ag}} = 2\sum_{t=1}^T t\mathbf{z}_{t-\frac{1}{2}} \quad \Rightarrow \quad \mathbf{z}_{T-\frac{1}{2}}^{\text{ag}} = \frac{2}{(T+1)T}\sum_{t=1}^T t\mathbf{z}_{t-\frac{1}{2}}.$$

Using this to analyze our algorithm:

$$t\mathbf{z}_t - (t-1)\mathbf{z}_{t-1} - \mathbf{z}_{t-1} = t(\mathbf{z}_t - \mathbf{z}_{t-1}) = -\eta\mathbf{J}\left[t\mathbf{z}_{t-\frac{1}{2}}\right] + \eta t\varepsilon_t,$$

so telescoping gives

$$T\mathbf{z}_T - \sum_{t=1}^T \mathbf{z}_{t-1} = -\eta\mathbf{J}\sum_{t=1}^T t\mathbf{z}_{t-\frac{1}{2}} + \eta\sum_{t=1}^T t\varepsilon_t,$$

which yields

$$\mathbf{z}_{T-\frac{1}{2}}^{\text{ag}} = \frac{2}{(T+1)T} \sum_{t=1}^{T} t\mathbf{z}_{t-\frac{1}{2}} = \frac{2}{-\eta(T+1)T} \mathbf{J}^{-1} \left( T\mathbf{z}_T - \sum_{t=1}^{T} \mathbf{z}_{t-1} - \eta \sum_{t=1}^{T} t\boldsymbol{\varepsilon}_t \right). \quad (50)$$

Obviously the least singular value of the matrix $\mathbf{J}$ can be lower bounded as $\sigma_{\min}(\mathbf{J}) \geq \sqrt{\lambda_{\min}(\mathbf{B}\mathbf{B}^{\top})}$. We conclude from (50) along with Young's inequality that

$$\lambda_{\min}(\mathbf{B}\mathbf{B}^{\top})\mathbb{E}\left\|\mathbf{z}_{T-\frac{1}{2}}^{\text{ag}}\right\|^2 \leq \mathbb{E}\left\|\mathbf{J}\mathbf{z}_{T-\frac{1}{2}}^{\text{ag}}\right\|^2$$

$$= (1+\gamma)\frac{4}{\eta^2(T+1)^2T^2}\mathbb{E}\left\|\sum_{t=1}^{T}(\mathbf{z}_T - \mathbf{z}_{t-1})\right\|^2 + (1+\tfrac{1}{\gamma})\frac{4}{\eta^2(T+1)^2T^2}\mathbb{E}\left\|\eta\sum_{t=1}^{T}t\boldsymbol{\varepsilon}_t\right\|^2$$

$$\equiv (1+\gamma)\mathbf{I} + (1+\tfrac{1}{\gamma})\mathbf{II},$$

where applying the last-iterate bound (47) together with some elementary estimates leads to

$$\mathbf{I} \leq \frac{4}{\eta^2(T+1)^2T^2} \cdot T\sum_{t=1}^{T}\left[2\mathbb{E}\|\mathbf{z}_T\|^2 + 2\mathbb{E}\|\mathbf{z}_{t-1}\|^2\right]$$

$$\leq \frac{4}{\eta^2(T+1)^2T^2} \cdot T\sum_{t=1}^{T}\left[4\mathbb{E}\|\mathbf{z}_0\|^2 + 4(T+t-1)\eta^2\sigma_{\text{Bil}}^2\right]$$

$$\leq \frac{16\mathbb{E}\|\mathbf{z}_0\|^2 + 24\eta^2\sigma_{\text{Bil}}^2 T}{\eta^2(T+1)^2} \leq \frac{16\lambda_{\max}(\mathbf{B}^{\top}\mathbf{B})\mathbb{E}\|\mathbf{z}_0\|^2}{(T+1)^2} + \frac{24\sigma_{\text{Bil}}^2}{T+1},$$

and, using the property of square-integrable martingales,

$$\mathbf{II} \leq \frac{4}{\eta^2(T+1)^2T^2}\mathbb{E}\left\|\eta\sum_{t=1}^{T}t\boldsymbol{\varepsilon}_t\right\|^2 = \frac{4}{\eta^2(T+1)^2T^2} \cdot \eta^2\sum_{t=1}^{T}t^2\mathbb{E}\|\boldsymbol{\varepsilon}_t\|^2$$

$$\leq \frac{4\sigma_{\text{Bil}}^2}{\eta^2(T+1)^2T^2} \cdot \eta^2\frac{T(T+\frac{1}{2})(T+1)}{3} \leq \frac{4\sigma_{\text{Bil}}^2}{3T}.$$

To summarize we have for arbitrary $\gamma \in (0, \infty)$

$$\lambda_{\min}(\mathbf{B}\mathbf{B}^{\top})\mathbb{E}\left\|\mathbf{z}_{T-\frac{1}{2}}^{\text{ag}}\right\|^2 \leq (1+\gamma)\left(\frac{16\lambda_{\max}(\mathbf{B}^{\top}\mathbf{B})\mathbb{E}\|\mathbf{z}_0\|^2}{(T+1)^2} + \frac{24\sigma_{\text{Bil}}^2}{T+1}\right) + (1+\tfrac{1}{\gamma})\frac{4\sigma_{\text{Bil}}^2}{3T}.$$

Optimizing $\gamma$ gives along with $\sqrt{a+b} \leq \sqrt{a} + \sqrt{b}$ for nonnegatives $a$ and $b$:

$$\sqrt{\lambda_{\min}(\mathbf{B}\mathbf{B}^{\top})}\sqrt{\mathbb{E}\left\|\mathbf{z}_{T-\frac{1}{2}}^{\text{ag}}\right\|^2} \leq \sqrt{\frac{16\lambda_{\max}(\mathbf{B}^{\top}\mathbf{B})\mathbb{E}\|\mathbf{z}_0\|^2}{(T+1)^2} + \frac{24\sigma_{\text{Bil}}^2}{T+1}} + \sqrt{\frac{4\sigma_{\text{Bil}}^2}{3T}}$$

$$\leq \sqrt{\frac{16\lambda_{\max}(\mathbf{B}^{\top}\mathbf{B})\mathbb{E}\|\mathbf{z}_0\|^2}{(T+1)^2}} + \sqrt{\frac{24\sigma_{\text{Bil}}^2}{T+1}} + \sqrt{\frac{4\sigma_{\text{Bil}}^2}{3T}} \leq \frac{4\sqrt{\lambda_{\max}(\mathbf{B}^{\top}\mathbf{B})}}{T+1}\sqrt{\mathbb{E}\|\mathbf{z}_0\|^2} + \frac{7\sigma_{\text{Bil}}}{\sqrt{T}}.$$

Dividing both sides by $\sqrt{\lambda_{\min}(\mathbf{B}\mathbf{B}^{\top})}$ and taking squares conclude the result.

## E  Proof of Auxiliary Lemmas

### E.1  Proof of Lemma D.1

The analysis in this subsection is partially motivated by Lemma 2 of Chen et al. [2017].

*Proof.*[Proof of Lemma D.1] We first introduce the following lemma on the operator $\mathcal{P}$:

**Lemma E.1 (Lemma 2 in Ghadimi and Lan 2012 and Lemma 1 in Chen et al. 2017)** *If* $\phi = \mathcal{P}_{\boldsymbol{\theta}}(\boldsymbol{\delta})$ *for arbitrarily chosen* $\boldsymbol{\theta}, \boldsymbol{\delta} \in \mathbb{R}^d$, *then for* $\forall \mathbf{z} \in \mathcal{Z}$, *we have the following inequality*

$$\langle \boldsymbol{\delta}, \phi - \mathbf{z} \rangle + J(\phi) - J(\mathbf{z}) \leq \mathcal{B}(\boldsymbol{\theta}, \mathbf{z}) - \mathcal{B}(\boldsymbol{\theta}, \phi) - V(\phi, \mathbf{z})$$

By applying Lemma E.1 to (25), we have for any $\mathbf{z} \in \mathcal{Z}$

$$\langle \boldsymbol{\delta}_1, \boldsymbol{\varphi}_1 - \mathbf{z} \rangle + J(\boldsymbol{\varphi}_1) - J(\mathbf{z}) \leq \mathcal{B}(\boldsymbol{\theta}, \mathbf{z}) - \mathcal{B}(\boldsymbol{\theta}, \boldsymbol{\varphi}_1) - \mathcal{B}(\boldsymbol{\varphi}_1, \mathbf{z}), \tag{51}$$

$$\langle \boldsymbol{\delta}_2, \boldsymbol{\varphi}_2 - \mathbf{z} \rangle + J(\boldsymbol{\varphi}_2) - J(\mathbf{z}) \leq \mathcal{B}(\boldsymbol{\theta}, \mathbf{z}) - \mathcal{B}(\boldsymbol{\theta}, \boldsymbol{\varphi}_2) - \mathcal{B}(\boldsymbol{\varphi}_2, \mathbf{z}). \tag{52}$$

Specifically, letting $\mathbf{z} = \boldsymbol{\varphi}_2$ in (51) we have

$$\langle \boldsymbol{\delta}_1, \boldsymbol{\varphi}_1 - \boldsymbol{\varphi}_2 \rangle + J(\boldsymbol{\varphi}_1) - J(\boldsymbol{\varphi}_2) = \mathcal{B}(\boldsymbol{\theta}, \boldsymbol{\varphi}_2) - \mathcal{B}(\boldsymbol{\theta}, \boldsymbol{\varphi}_1) - \mathcal{B}(\boldsymbol{\varphi}_1, \boldsymbol{\varphi}_2). \tag{53}$$

Now, combining inequalities (52) and (53) we have

$$\langle \boldsymbol{\delta}_2, \boldsymbol{\varphi}_2 - \mathbf{z} \rangle + \langle \boldsymbol{\delta}_1, \boldsymbol{\varphi}_1 - \boldsymbol{\varphi}_2 \rangle + J(\boldsymbol{\varphi}_1) - J(\mathbf{z}) \leq \mathcal{B}(\boldsymbol{\theta}, \mathbf{z}) - \mathcal{B}(\boldsymbol{\varphi}_2, \mathbf{z}) - \mathcal{B}(\boldsymbol{\theta}, \boldsymbol{\varphi}_1) - \mathcal{B}(\boldsymbol{\varphi}_1, \boldsymbol{\varphi}_2),$$

which in turn gives

$$\langle \boldsymbol{\delta}_2, \boldsymbol{\varphi}_1 - \mathbf{z} \rangle + J(\boldsymbol{\varphi}_1) - J(\mathbf{z})$$
$$\leq \langle \boldsymbol{\delta}_2 - \boldsymbol{\delta}_1, \boldsymbol{\varphi}_1 - \boldsymbol{\varphi}_2 \rangle + \mathcal{B}(\boldsymbol{\theta}, \mathbf{z}) - \mathcal{B}(\boldsymbol{\varphi}_2, \mathbf{z}) - \mathcal{B}(\boldsymbol{\theta}, \boldsymbol{\varphi}_1) - \mathcal{B}(\boldsymbol{\varphi}_1, \boldsymbol{\varphi}_2).$$

An application of the Young and Cauchy-Schwartz inequalities gives

$$
\begin{aligned}
&\langle \boldsymbol{\delta}_2, \boldsymbol{\varphi}_1 - \mathbf{z} \rangle + J(\boldsymbol{\varphi}_1) - J(\mathbf{z}) \\
&\leq \|\boldsymbol{\delta}_2 - \boldsymbol{\delta}_1\| \|\boldsymbol{\varphi}_1 - \boldsymbol{\varphi}_2\| + \mathcal{B}(\boldsymbol{\theta}, \mathbf{z}) - \mathcal{B}(\boldsymbol{\varphi}_2, \mathbf{z}) - \mathcal{B}(\boldsymbol{\theta}, \boldsymbol{\varphi}_1) - \mathcal{B}(\boldsymbol{\varphi}_1, \boldsymbol{\varphi}_2) \\
&\leq \frac{1}{2\mu_{\mathcal{B}}} \|\boldsymbol{\delta}_2 - \boldsymbol{\delta}_1\|^2 + \frac{\mu_{\mathcal{B}}}{2} \|\boldsymbol{\varphi}_1 - \boldsymbol{\varphi}_2\|^2 \\
&\quad + \mathcal{B}(\boldsymbol{\theta}, \mathbf{z}) - \mathcal{B}(\boldsymbol{\varphi}_2, \mathbf{z}) - \mathcal{B}(\boldsymbol{\theta}, \boldsymbol{\varphi}_1) - \mathcal{B}(\boldsymbol{\varphi}_1, \boldsymbol{\varphi}_2) \\
&\leq \frac{1}{2\mu_{\mathcal{B}}} \|\boldsymbol{\delta}_2 - \boldsymbol{\delta}_1\|^2 + \mathcal{B}(\boldsymbol{\theta}, \mathbf{z}) - \mathcal{B}(\boldsymbol{\varphi}_2, \mathbf{z}) - \mathcal{B}(\boldsymbol{\theta}, \boldsymbol{\varphi}_1).
\end{aligned}
\tag{54}
$$

In the last inequality, we uses the fact that

$$\frac{\mu_{\mathcal{B}}}{2} \|\boldsymbol{\varphi}_1 - \boldsymbol{\varphi}_2\|^2 \leq \mathcal{B}(\boldsymbol{\varphi}_1, \boldsymbol{\varphi}_2).$$

This establishes (27) and hence Lemma D.1.

## E.2  Proof of Lemma D.2

*Proof.*[Proof of Lemma D.2]

Since $\mathcal{F}(\mathbf{z})$ is $L$-smooth and $\mu$-strongly convex. For the rest of this proof, we observe that the saddle definition of $\mathbf{z}^*$ satisfies the first-order stationary condition for problem (3):

$$\nabla \mathcal{F}(\mathbf{z}^*) + \mathcal{H}(\mathbf{z}^*) + J'(\mathbf{z}^*) = 0. \tag{55}$$

Furthermore, we have

$$
\begin{aligned}
&\mathcal{F}(\mathbf{z}) - \mathcal{F}(\mathbf{z}^*) + \langle \mathcal{H}(\mathbf{z}^*), \mathbf{z} - \mathbf{z}^* \rangle + J(\mathbf{z}) - J(\mathbf{z}^*) \\
&\geq \langle \nabla \mathcal{F}(\mathbf{z}^*), \mathbf{z} - \mathbf{z}^* \rangle + \frac{\mu}{2} \|\mathbf{z} - \mathbf{z}^*\|^2 + \langle \mathcal{H}(\mathbf{z}^*), \mathbf{z} - \mathbf{z}^* \rangle + \langle J'(\mathbf{z}^*), \mathbf{z} - \mathbf{z}^* \rangle \\
&= \langle \nabla \mathcal{F}(\mathbf{z}^*) + \mathcal{H}(\mathbf{z}^*) + J'(\mathbf{z}^*), \mathbf{z} - \mathbf{z}^* \rangle + \frac{\mu}{2} \|\mathbf{z} - \mathbf{z}^*\|^2 = \frac{\mu}{2} \|\mathbf{z} - \mathbf{z}^*\|^2,
\end{aligned}
$$

where in both of the two displays, the inequality holds due to the $\mu$-strong convexity of $\mathcal{F}$, and the equality holds due to the first-order stationary condition (55). This completes the proof.

## E.3  Proof of Lemma D.3

*Proof.*[Proof of Lemma D.3] Items (i)—(iii) are straightforward. For the proof of (30) in item (iv), we note that $\eta_t = \bar{\eta}_t(\sigma; T, C, r, \beta) \leq \dfrac{t}{\frac{2}{r}L + \sqrt{\frac{1+\beta}{r}}Mt} \leq \dfrac{1}{\sqrt{\frac{1+\beta}{r}}M}$ which gives

$$r - \frac{2L}{t+1}\eta_t - (1+\beta)M^2\eta_t^2 \geq \frac{r}{t}\left(t - \left(\frac{2}{r}L + \sqrt{\frac{1+\beta}{r}}Mt\right)\eta_t\right) \geq 0,$$

and hence completes the proof.

### E.4 Proof of Lemma D.4

*Proof.*[Proof of Lemma D.4] From the convexity and $L$-smoothness of $F$ as in Assumption 2.1, we know that for arbitrary $\tilde{z}$:

$$\mathcal{F}(z^{\mathrm{ag}}_{t-\frac{1}{2}}) - \mathcal{F}(\tilde{z}) = \mathcal{F}(z^{\mathrm{ag}}_{t-\frac{1}{2}}) - \mathcal{F}(z^{\mathrm{md}}_{t-1}) - \left(\mathcal{F}(\tilde{z}) - \mathcal{F}(z^{\mathrm{md}}_{t-1})\right)$$

$$\leq \langle \nabla\mathcal{F}(z^{\mathrm{md}}_{t-1}), z^{\mathrm{ag}}_{t-\frac{1}{2}} - z^{\mathrm{md}}_{t-1}\rangle + \tfrac{L}{2}\left\|z^{\mathrm{ag}}_{t-\frac{1}{2}} - z^{\mathrm{md}}_{t-1}\right\|^2 - \langle \nabla\mathcal{F}(z^{\mathrm{md}}_{t-1}), \tilde{z} - z^{\mathrm{md}}_{t-1}\rangle.$$

Taking $\tilde{z} = z^{\mathrm{ag}}_{t-\frac{3}{2}}$ in the above inequality, we have

$$\mathcal{F}(z^{\mathrm{ag}}_{t-\frac{1}{2}}) - \mathcal{F}(z^{\mathrm{ag}}_{t-\frac{3}{2}}) = \mathcal{F}(z^{\mathrm{ag}}_{t-\frac{1}{2}}) - \mathcal{F}(z^{\mathrm{md}}_{t-1}) - \left(\mathcal{F}(z^{\mathrm{ag}}_{t-\frac{3}{2}}) - \mathcal{F}(z^{\mathrm{md}}_{t-1})\right)$$

$$\leq \langle \nabla\mathcal{F}(z^{\mathrm{md}}_{t-1}), z^{\mathrm{ag}}_{t-\frac{1}{2}} - z^{\mathrm{md}}_{t-1}\rangle + \tfrac{L}{2}\left\|z^{\mathrm{ag}}_{t-\frac{1}{2}} - z^{\mathrm{md}}_{t-1}\right\|^2 - \langle \nabla\mathcal{F}(z^{\mathrm{md}}_{t-1}), z^{\mathrm{ag}}_{t-\frac{3}{2}} - z^{\mathrm{md}}_{t-1}\rangle.$$

Multiplying the first display by $\alpha_t$ and the second display by $(1 - \alpha_t)$ and adding them up, we have

$$\mathcal{F}(z^{\mathrm{ag}}_{t-\frac{1}{2}}) - (1-\alpha_t)\mathcal{F}(z^{\mathrm{ag}}_{t-\frac{3}{2}}) - \alpha_t\mathcal{F}(\tilde{z})$$

$$\leq \langle \nabla\mathcal{F}(z^{\mathrm{md}}_{t-1}), z^{\mathrm{ag}}_{t-\frac{1}{2}} - z^{\mathrm{md}}_{t-1}\rangle + \tfrac{L}{2}\left\|z^{\mathrm{ag}}_{t-\frac{1}{2}} - z^{\mathrm{md}}_{t-1}\right\|^2 - \langle \nabla\mathcal{F}(z^{\mathrm{md}}_{t-1}), (1-\alpha_t)z^{\mathrm{ag}}_{t-\frac{3}{2}} + \alpha_t\tilde{z} - z^{\mathrm{md}}_{t-1}\rangle$$

$$\leq \langle \nabla\mathcal{F}(z^{\mathrm{md}}_{t-1}), \alpha_t(z_{t-\frac{1}{2}} - z_{t-1})\rangle + \tfrac{L}{2}\|\alpha_t(z_{t-\frac{1}{2}} - z_{t-1})\|^2 - \langle \nabla\mathcal{F}(z^{\mathrm{md}}_{t-1}), \alpha_t(\tilde{z} - z_{t-1})\rangle$$

$$= \alpha_t\langle \nabla\mathcal{F}(z^{\mathrm{md}}_{t-1}), z_{t-\frac{1}{2}} - \tilde{z}\rangle + \tfrac{\alpha_t^2 L}{2}\|z_{t-\frac{1}{2}} - z_{t-1}\|^2,$$

$$\tag{56}$$

where we applied the fact from our update rules that $z^{\mathrm{ag}}_{t-\frac{1}{2}} - z^{\mathrm{md}}_{t-1} = \alpha_t(z_{t-\frac{1}{2}} - z_{t-1})$.

On the other hand, due to Line (6) in Algorithm 1 we have

$$\langle \mathcal{H}(\tilde{z}), z^{\mathrm{ag}}_{t-\frac{1}{2}} - \tilde{z}\rangle - (1-\alpha_t)\langle \mathcal{H}(\tilde{z}), z^{\mathrm{ag}}_{t-\frac{3}{2}} - \tilde{z}\rangle = \langle \mathcal{H}(\tilde{z}), z^{\mathrm{ag}}_{t-\frac{1}{2}} - \tilde{z} - (1-\alpha_t)(z^{\mathrm{ag}}_{t-\frac{3}{2}} - \tilde{z})\rangle$$

$$= \alpha_t\langle \mathcal{H}(\tilde{z}), z_{t-\frac{1}{2}} - \tilde{z}\rangle.$$

Further, due to our monotonicity assumption on $\mathcal{H}$ we have

$$\langle \mathcal{H}(\tilde{z}), z_{t-\frac{1}{2}} - \tilde{z}\rangle \leq \langle \mathcal{H}(z_{t-\frac{1}{2}}), z_{t-\frac{1}{2}} - \tilde{z}\rangle.$$

Combining the above two displays together yields

$$\langle \mathcal{H}(\tilde{z}), z^{\mathrm{ag}}_{t-\frac{1}{2}} - \tilde{z}\rangle - (1-\alpha_t)\langle \mathcal{H}(\tilde{z}), z^{\mathrm{ag}}_{t-\frac{3}{2}} - \tilde{z}\rangle \leq \alpha_t\langle \mathcal{H}(z_{t-\frac{1}{2}}), z_{t-\frac{1}{2}} - \tilde{z}\rangle. \tag{57}$$

Now, summing up Eqs. (56), (57) and recalling the definition of $V$ in (28), we conclude that

$$V(z^{\mathrm{ag}}_{t-\frac{1}{2}} \mid \tilde{z}) - (1-\alpha_t)V(z^{\mathrm{ag}}_{t-\frac{3}{2}} \mid \tilde{z})$$

$$= \mathcal{F}(z^{\mathrm{ag}}_{t-\frac{1}{2}}) - (1-\alpha_t)\mathcal{F}(z^{\mathrm{ag}}_{t-\frac{3}{2}}) - \alpha_t\mathcal{F}(\tilde{z}) + \langle \mathcal{H}(\tilde{z}), z^{\mathrm{ag}}_{t-\frac{1}{2}} - \tilde{z}\rangle - (1-\alpha_t)\langle \mathcal{H}(\tilde{z}), z^{\mathrm{ag}}_{t-\frac{3}{2}} - \tilde{z}\rangle$$

$$\leq \alpha_t\langle \nabla\mathcal{F}(z^{\mathrm{md}}_{t-1}) + \mathcal{H}(z_{t-\frac{1}{2}}), z_{t-\frac{1}{2}} - \tilde{z}\rangle + \tfrac{\alpha_t^2 L}{2}\|z_{t-\frac{1}{2}} - z_{t-1}\|^2,$$

and hence conclude (32) and Lemma D.4.

### E.5 Proof of Lemma D.5

*Proof.*[Proof of Lemma D.5] To bound the inner-product terms in (32), by setting $\varphi_1 = z_{t-\frac{1}{2}}$, $\theta = z_{t-1}$, $\varphi_2 = z_t$, $\delta_1 = \eta_t\left(\nabla\tilde{\mathcal{F}}(z^{\mathrm{md}}_{t-1}; \xi_{-\frac{1}{2}}) + \tilde{\mathcal{H}}(z_{t-1}; \zeta_{-\frac{1}{2}})\right)$, $\delta_2 = \eta_t\left(\nabla\tilde{\mathcal{F}}(z^{\mathrm{md}}_{t-1}; \xi_{-\frac{1}{2}}) + \tilde{\mathcal{H}}(z_{t-\frac{1}{2}}; \zeta_t)\right)$ as in Lemma D.1 (with $\mathbf{z} = \tilde{z}$), we have

$$\eta_t\langle \nabla\tilde{\mathcal{F}}(z^{\mathrm{md}}_{t-1}; \xi_{-\frac{1}{2}}) + \tilde{\mathcal{H}}(z_{t-\frac{1}{2}}; \zeta_t), z_{t-\frac{1}{2}} - \tilde{z}\rangle$$

$$\leq \tfrac{1}{2}\left[\|z_{t-1} - \tilde{z}\|^2 - \|\mathbf{x}_t - \tilde{z}\|^2 - \left\|z_{t-\frac{1}{2}} - z_{t-1}\right\|^2\right] + \tfrac{\eta_t^2}{2}\|\tilde{\mathcal{H}}(z_{t-\frac{1}{2}}; \zeta_t) - \tilde{\mathcal{H}}(z_{t-1}; \zeta_{-\frac{1}{2}})\|^2,$$

where Young's inequality combined with the martingale structure yields (also noting (31))

$$\mathbb{E}\|\tilde{\mathcal{H}}(z_{t-\frac{1}{2}};\zeta_t) - \tilde{\mathcal{H}}(z_{t-1};\zeta_{t-\frac{1}{2}})\|^2$$

$$= \mathbb{E}\|\mathcal{H}(z_{t-\frac{1}{2}}) - \mathcal{H}(z_{t-1}) - \mathbf{\Delta}_{\mathcal{H}}^{t-\frac{1}{2}}\|^2 + \mathbb{E}\left\|\mathbf{\Delta}_{\mathcal{H}}^{t}\right\|^2$$

$$\leq (1+\beta)M^2\mathbb{E}\|z_{t-\frac{1}{2}} - z_{t-1}\|^2 + (1+\tfrac{1}{\beta})\mathbb{E}\left\|\mathbf{\Delta}_{\mathcal{H}}^{t-\frac{1}{2}}\right\|^2 + \mathbb{E}\left\|\mathbf{\Delta}_{\mathcal{H}}^{t}\right\|^2.$$

Combining the above two displays with expectation taken gives

$$\mathbb{E}\left[\eta_t\langle\nabla\tilde{\mathcal{F}}(z_{t-1}^{\mathrm{md}};\xi_{t-\frac{1}{2}}) + \tilde{\mathcal{H}}(z_{t-\frac{1}{2}};\zeta_t), z_{t-\frac{1}{2}} - \tilde{z}\rangle\right]$$

$$\leq \frac{1}{2}\mathbb{E}\left[\|z_{t-1} - \tilde{z}\|^2 - \|\mathbf{x}_t - \tilde{z}\|^2 - \left\|z_{t-\frac{1}{2}} - z_{t-1}\right\|^2\right] \tag{58}$$

$$+ \frac{\eta_t^2}{2}\left((1+\beta)M^2\mathbb{E}\left\|z_{t-\frac{1}{2}} - z_{t-1}\right\|^2 + (1+\tfrac{1}{\beta})\mathbb{E}\left\|\mathbf{\Delta}_{\mathcal{H}}^{t-\frac{1}{2}}\right\|^2 + \mathbb{E}\left\|\mathbf{\Delta}_{\mathcal{H}}^{t}\right\|^2\right).$$

Further, by definition of the primal-dual gap function and the definition of the noisy terms (31), by taking $\alpha_t = \frac{2}{t+1}$ in (32) of Lemma D.4 and taking expectations on both sides, we have

$$\mathbb{E}\left[V\big(z_{t-\frac{1}{2}}^{\mathrm{ag}} \mid \tilde{z}\big)\right] - \frac{t-1}{t+1}\mathbb{E}\left[V\big(z_{t-\frac{3}{2}}^{\mathrm{ag}} \mid \tilde{z}\big)\right]$$

$$\leq \frac{2}{t+1}\mathbb{E}\left\langle\nabla\mathcal{F}(z_{t-1}^{\mathrm{md}}) + \mathcal{H}(z_{t-\frac{1}{2}}), z_{t-\frac{1}{2}} - \tilde{z}\right\rangle + \frac{2L}{(t+1)^2}\mathbb{E}\left\|z_{t-\frac{1}{2}} - z_{t-1}\right\|^2$$

$$= \frac{2}{t+1}\mathbb{E}\left\langle\nabla\tilde{\mathcal{F}}(z_{t-1}^{\mathrm{md}};\xi_{t-\frac{1}{2}}) + \tilde{\mathcal{H}}(z_{t-\frac{1}{2}};\zeta_t), z_{t-\frac{1}{2}} - \tilde{z}\right\rangle + \frac{2L}{(t+1)^2}\mathbb{E}\left\|z_{t-\frac{1}{2}} - z_{t-1}\right\|^2$$

$$- \frac{2}{t+1}\mathbb{E}\left\langle\mathbf{\Delta}_{\mathcal{F}}^{t-\frac{1}{2}} + \mathbf{\Delta}_{\mathcal{H}}^{t}, z_{t-\frac{1}{2}} - \tilde{z}\right\rangle$$

Bringing in (58) into the above derivation, we obtain

$$\mathbb{E}\left[V\big(z_{t-\frac{1}{2}}^{\mathrm{ag}} \mid \tilde{z}\big)\right] - \frac{t-1}{t+1}\mathbb{E}\left[V\big(z_{t-\frac{3}{2}}^{\mathrm{ag}} \mid \tilde{z}\big)\right]$$

$$\leq \frac{1}{(t+1)\eta_t}\mathbb{E}\left[\|z_{t-1} - \tilde{z}\|^2 - \|z_t - \tilde{z}\|^2\right]$$

$$- \frac{1}{(t+1)\eta_t}\left(1 - \frac{2L}{t+1}\eta_t - (1+\beta)M^2\eta_t^2\right)\mathbb{E}\left\|z_{t-\frac{1}{2}} - z_{t-1}\right\|^2$$

$$+ \frac{\eta_t}{t+1}\left((1+\tfrac{1}{\beta})\mathbb{E}\left\|\mathbf{\Delta}_{\mathcal{H}}^{t-\frac{1}{2}}\right\|^2 + \mathbb{E}\left\|\mathbf{\Delta}_{\mathcal{H}}^{t}\right\|^2\right) - \frac{2}{t+1}\mathbb{E}\left\langle\mathbf{\Delta}_{\mathcal{F}}^{t-\frac{1}{2}} + \mathbf{\Delta}_{\mathcal{H}}^{t}, z_{t-\frac{1}{2}} - \tilde{z}\right\rangle.$$

Recalling that we use the choice of $\eta_t$ that satisfies for a given $r \in (0,1)$ that $r - \frac{2L}{t+1}\eta_t - (1+\beta)M^2\eta_t^2 \geq 0$. With some manipulations we obtain

$$\mathbb{E}\left[V\big(z_{t-\frac{1}{2}}^{\mathrm{ag}} \mid \tilde{z}\big)\right] - \frac{t-1}{t+1}\mathbb{E}\left[V\big(z_{t-\frac{3}{2}}^{\mathrm{ag}} \mid \tilde{z}\big)\right]$$

$$\leq \frac{1}{(t+1)\eta_t}\mathbb{E}\left[\|z_{t-1} - \tilde{z}\|^2 - \|z_t - \tilde{z}\|^2\right]$$

$$+ \frac{\eta_t}{(t+1)}\left((1+\tfrac{1}{\beta})\mathbb{E}\left\|\mathbf{\Delta}_{\mathcal{H}}^{t-\frac{1}{2}}\right\|^2 + \mathbb{E}\left\|\mathbf{\Delta}_{\mathcal{H}}^{t}\right\|^2\right) - \frac{(1-r)}{(t+1)\eta_t}\mathbb{E}\left[\left\|z_{t-\frac{1}{2}} - z_{t-1}\right\|^2\right]$$

$$- \frac{2}{t+1}\mathbb{E}\left\langle\mathbf{\Delta}_{\mathcal{F}}^{t-\frac{1}{2}}, z_{t-\frac{1}{2}} - z_{t-1}\right\rangle \underbrace{- \frac{2}{t+1}\mathbb{E}\left\langle\mathbf{\Delta}_{\mathcal{F}}^{t-\frac{1}{2}}, z_{t-1} - \tilde{z}\right\rangle - \frac{2}{t+1}\mathbb{E}\left\langle\mathbf{\Delta}_{\mathcal{H}}^{t}, z_{t-\frac{1}{2}} - \tilde{z}\right\rangle}_{\mathrm{I}}.$$

$$\tag{59}$$

Due to the law of iterated expectation applied to martingale difference conditions $\mathbb{E}\left[\mathbf{\Delta}_{\mathcal{F}}^{t-\frac{1}{2}} \mid \mathcal{F}_{t-1}\right] = \mathbf{0}$ and $\mathbb{E}\left[\mathbf{\Delta}_{\mathcal{H}}^{t} \mid \mathcal{F}_{t-\frac{1}{2}}\right] = \mathbf{0}$, $i = 1, 2$, we have

$$\mathrm{I} = 0.$$

Moreover, for the rest of the terms in (59), we note that there is a basic quadratic inequality that
$-\frac{1-r}{\eta_t}\left\|z_{t-1}-\mathbf{x}_{t-\frac{1}{2}}\right\|^2 - 2\left\langle \mathbf{\Delta}_{\mathcal{F}}^{t-\frac{1}{2}}, z_{t-\frac{1}{2}} - z_{t-1}\right\rangle \le \frac{\eta_t}{1-r}\left\|\mathbf{\Delta}_{\mathcal{F}}^{t-\frac{1}{2}}\right\|^2$. (59) reduces to

$$\mathbb{E}\left[V\left(z_{t-\frac{1}{2}}^{\mathrm{ag}} \mid \tilde{z}\right)\right] - \frac{t-1}{t+1}\mathbb{E}\left[V\left(z_{t-\frac{3}{2}}^{\mathrm{ag}} \mid \tilde{z}\right)\right] \le \frac{1}{(t+1)\eta_t}\mathbb{E}\left[\|z_{t-1}-\tilde{z}\|^2 - \|z_t - \tilde{z}\|^2\right]$$
$$+ \frac{\eta_t}{t+1}\left((1+\tfrac{1}{\beta})\mathbb{E}\left\|\mathbf{\Delta}_{\mathcal{H}}^{t-\frac{1}{2}}\right\|^2 + \mathbb{E}\left\|\mathbf{\Delta}_{\mathcal{H}}^{t}\right\|^2\right) + \frac{\eta_t}{(1-r)(t+1)}\mathbb{E}\left\|\mathbf{\Delta}_{\mathcal{F}}^{t-\frac{1}{2}}\right\|^2. \tag{60}$$

Multiplying both sides of (60) by $t(t+1)$, we obtain for all $t = 1, \ldots, T$

$$t(t+1)\mathbb{E}\left[V\left(z_{t-\frac{1}{2}}^{\mathrm{ag}} \mid \tilde{z}\right)\right] - (t-1)t\mathbb{E}\left[V\left(z_{t-\frac{3}{2}}^{\mathrm{ag}} \mid \tilde{z}\right)\right]$$
$$\le \frac{t}{\eta_t}\mathbb{E}\left[\|z_{t-1}-\tilde{z}\| - \|z_t - \tilde{z}\|^2\right] + t\eta_t\left(\frac{1}{1-r}\mathbb{E}\left\|\mathbf{\Delta}_{\mathcal{F}}^{t-\frac{1}{2}}\right\|^2 + (1+\tfrac{1}{\beta})\mathbb{E}\left\|\mathbf{\Delta}_{\mathcal{H}}^{t-\frac{1}{2}}\right\|^2 + \mathbb{E}\|\mathbf{\Delta}_{\mathcal{H}}^{t}\|^2\right)$$
$$\le \frac{t}{\eta_t}\mathbb{E}\left[\|z_{t-1}-\tilde{z}\|^2 - \|z_t - \tilde{z}\|^2\right] + \left(\frac{1}{1-r}\sigma_{\mathrm{Str}}^2 + (2+\tfrac{1}{\beta})\sigma_{\mathrm{Bil}}^2\right)t\eta_t,$$

where in the last line above we applied Assumption 2.2, so by law of iterated expectations

$$\mathbb{E}\left\|\mathbf{\Delta}_{\mathcal{F}}^{t-\frac{1}{2}}\right\|^2 = \mathbb{E}\left[\|\nabla\tilde{F}(z_{t-1}^{\mathrm{md}};\xi_{t-\frac{1}{2}}) - \nabla F(z_{t-1}^{\mathrm{md}})\|^2\right] \le \sigma_{\mathrm{Str}}^2,$$
$$\mathbb{E}\left\|\mathbf{\Delta}_{\mathcal{H}}^{t-\frac{1}{2}}\right\|^2 = \mathbb{E}\left[\|\tilde{H}(z_{t-1};\zeta_{t-\frac{1}{2}}) - H(z_{t-1})\|^2\right] \le \sigma_{\mathrm{Bil}}^2, \tag{61}$$
$$\mathbb{E}\|\mathbf{\Delta}_{\mathcal{H}}^{t}\|^2 = \mathbb{E}\left[\|\tilde{H}(z_{t-\frac{1}{2}};\zeta_t) - H(z_{t-\frac{1}{2}})\|^2\right] \le \sigma_{\mathrm{Bil}}^2.$$

## E.6  Proof of Lemma D.6

*Proof.*[Proof of Lemma D.6] The proof of Lemma D.6 goes in an analogous fashion as the proof of Lemma D.4, except that the display above (57) is replaced by

$$\langle \mathcal{H}(\tilde{z}), z_{t-\frac{1}{2}} - \tilde{z}\rangle \le \langle \mathcal{H}(z_{t-\frac{1}{2}}), z_{t-\frac{1}{2}} - \tilde{z}\rangle - \mu_\star\left\|z_{t-\frac{1}{2}} - \tilde{z}\right\|^2,$$

due to that $H$ is a $\mu_\star$-strongly-convex-$\mu_\star$-strongly-concave isotropic quadratic function after scaling reduction. Hence (57) becomes

$$\langle \mathcal{H}(\tilde{z}), z_{t-\frac{1}{2}}^{\mathrm{ag}} - \tilde{z}\rangle - (1-\alpha_t)\langle \mathcal{H}(\tilde{z}), z_{t-\frac{3}{2}}^{\mathrm{ag}} - \tilde{z}\rangle \le \alpha_t\left[\langle \mathcal{H}(z_{t-\frac{1}{2}}), z_{t-\frac{1}{2}} - \tilde{z}\rangle - \mu_\star\left\|z_{t-\frac{1}{2}} - \tilde{z}\right\|^2\right]. \tag{62}$$

Therefore, we omit its detailed proof.

## E.7  Proof of Lemma D.7

*Proof.*[Proof of Lemma D.7] From the convexity and $L$-smoothness of $F$ as in Assumption 2.1, we know that for arbitrary $\tilde{z}$:

$$\mathcal{F}(z_{t-\frac{1}{2}}^{\mathrm{ag}}) - \mathcal{F}(\tilde{z}) = \mathcal{F}(z_{t-\frac{1}{2}}^{\mathrm{ag}}) - \mathcal{F}(z_{t-1}^{\mathrm{md}}) - \left(\mathcal{F}(\tilde{z}) - \mathcal{F}(z_{t-1}^{\mathrm{md}})\right)$$
$$\le \langle \nabla\mathcal{F}(z_{t-1}^{\mathrm{md}}), z_{t-\frac{1}{2}}^{\mathrm{ag}} - z_{t-1}^{\mathrm{md}}\rangle + \frac{L}{2}\left\|z_{t-\frac{1}{2}}^{\mathrm{ag}} - z_{t-1}^{\mathrm{md}}\right\|^2 - \langle \nabla\mathcal{F}(z_{t-1}^{\mathrm{md}}), \tilde{z} - z_{t-1}^{\mathrm{md}}\rangle.$$

Taking $\tilde{z} = z_{t-\frac{3}{2}}^{\mathrm{ag}}$ in the above inequality, we have

$$\mathcal{F}(z_{t-\frac{1}{2}}^{\mathrm{ag}}) - \mathcal{F}(z_{t-\frac{3}{2}}^{\mathrm{ag}}) = \mathcal{F}(z_{t-\frac{1}{2}}^{\mathrm{ag}}) - \mathcal{F}(z_{t-1}^{\mathrm{md}}) - \left(\mathcal{F}(z_{t-\frac{3}{2}}^{\mathrm{ag}}) - \mathcal{F}(z_{t-1}^{\mathrm{md}})\right)$$
$$\le \langle \nabla\mathcal{F}(z_{t-1}^{\mathrm{md}}), z_{t-\frac{1}{2}}^{\mathrm{ag}} - z_{t-1}^{\mathrm{md}}\rangle + \frac{L}{2}\left\|z_{t-\frac{1}{2}}^{\mathrm{ag}} - z_{t-1}^{\mathrm{md}}\right\|^2 - \langle \nabla\mathcal{F}(z_{t-1}^{\mathrm{md}}), z_{t-\frac{3}{2}}^{\mathrm{ag}} - z_{t-1}^{\mathrm{md}}\rangle.$$

Multiplying the first display by $\alpha_t$ and the second display by $(1 - \alpha_t)$ and adding them up, we have

$$\mathcal{F}(z_{t-\frac{1}{2}}^{\mathrm{ag}}) - (1 - \alpha_t)\mathcal{F}(z_{t-\frac{3}{2}}^{\mathrm{ag}}) - \alpha_t\mathcal{F}(\tilde{z})$$

$$\leq \langle \nabla\mathcal{F}(z_{t-1}^{\mathrm{md}}), z_{t-\frac{1}{2}}^{\mathrm{ag}} - z_{t-1}^{\mathrm{md}} \rangle + \frac{L}{2}\left\| z_{t-\frac{1}{2}}^{\mathrm{ag}} - z_{t-1}^{\mathrm{md}} \right\|^2 - \langle \nabla\mathcal{F}(z_{t-1}^{\mathrm{md}}), (1 - \alpha_t)z_{t-\frac{3}{2}}^{\mathrm{ag}} + \alpha_t\tilde{z} - z_{t-1}^{\mathrm{md}} \rangle$$

$$\leq \langle \nabla\mathcal{F}(z_{t-1}^{\mathrm{md}}), \alpha_t(z_{t-\frac{1}{2}} - z_{t-1}) \rangle + \frac{L}{2}\|\alpha_t(z_{t-\frac{1}{2}} - z_{t-1})\|^2 - \langle \nabla\mathcal{F}(z_{t-1}^{\mathrm{md}}), \alpha_t(\tilde{z} - z_{t-1}) \rangle$$

$$= \alpha_t\langle \nabla\mathcal{F}(z_{t-1}^{\mathrm{md}}), z_{t-\frac{1}{2}} - \tilde{z} \rangle + \frac{\alpha_t^2 L}{2}\|z_{t-\frac{1}{2}} - z_{t-1}\|^2, \tag{63}$$

where we applied the fact from our update rules that $z_{t-\frac{1}{2}}^{\mathrm{ag}} - z_{t-1}^{\mathrm{md}} = \alpha_t(z_{t-\frac{1}{2}} - z_{t-1})$.

On the other hand, due to Line (6) in Algorithm 3 we have

$$\langle \mathcal{H}(\tilde{z}), z_{t-\frac{1}{2}}^{\mathrm{ag}} - \tilde{z} \rangle - (1 - \alpha_t)\langle \mathcal{H}(\tilde{z}), z_{t-\frac{3}{2}}^{\mathrm{ag}} - \tilde{z} \rangle = \langle \mathcal{H}(\tilde{z}), z_{t-\frac{1}{2}}^{\mathrm{ag}} - \tilde{z} - (1 - \alpha_t)(z_{t-\frac{3}{2}}^{\mathrm{ag}} - \tilde{z}) \rangle$$

$$= \alpha_t\langle \mathcal{H}(\tilde{z}), z_{t-\frac{1}{2}} - \tilde{z} \rangle.$$

Further, due to our monotonicity assumption on $\mathcal{H}$ we have

$$\langle \mathcal{H}(\tilde{z}), z_{t-\frac{1}{2}} - \tilde{z} \rangle \leq \langle \mathcal{H}(z_{t-\frac{1}{2}}), z_{t-\frac{1}{2}} - \tilde{z} \rangle.$$

Combining the above two displays together yields

$$\langle \mathcal{H}(\tilde{z}), z_{t-\frac{1}{2}}^{\mathrm{ag}} - \tilde{z} \rangle - (1 - \alpha_t)\langle \mathcal{H}(\tilde{z}), z_{t-\frac{3}{2}}^{\mathrm{ag}} - \tilde{z} \rangle \leq \alpha_t\langle \mathcal{H}(z_{t-\frac{1}{2}}), z_{t-\frac{1}{2}} - \tilde{z} \rangle. \tag{64}$$

Moreover, we have

$$J(z_{t-\frac{1}{2}}^{\mathrm{ag}}) - J(\tilde{z}) - (1 - \alpha_t)\left( J(z_{t-\frac{3}{2}}^{\mathrm{ag}}) - J(\tilde{z}) \right) = J(z_{t-\frac{1}{2}}^{\mathrm{ag}}) - (1 - \alpha_t)J(z_{t-\frac{3}{2}}^{\mathrm{ag}}) - \alpha_t J(\tilde{z})$$

$$\leq \alpha_t J(z_{t-\frac{1}{2}}) - \alpha_t J(\tilde{z}) \tag{65}$$

Now, summing up Eqs. (63), (64) and recalling the definition of $V$, we conclude that

$$V(z_{t-\frac{1}{2}}^{\mathrm{ag}} \mid \tilde{z}) - (1 - \alpha_t)V(z_{t-\frac{3}{2}}^{\mathrm{ag}} \mid \tilde{z})$$

$$= \mathcal{F}(z_{t-\frac{1}{2}}^{\mathrm{ag}}) - (1 - \alpha_t)\mathcal{F}(z_{t-\frac{3}{2}}^{\mathrm{ag}}) - \alpha_t\mathcal{F}(\tilde{z}) + \langle \mathcal{H}(\tilde{z}), z_{t-\frac{1}{2}}^{\mathrm{ag}} - \tilde{z} \rangle - (1 - \alpha_t)\langle \mathcal{H}(\tilde{z}), z_{t-\frac{3}{2}}^{\mathrm{ag}} - \tilde{z} \rangle$$

$$\leq \alpha_t\langle \nabla\mathcal{F}(z_{t-1}^{\mathrm{md}}) + \mathcal{H}(z_{t-\frac{1}{2}}), z_{t-\frac{1}{2}} - \tilde{z} \rangle + \frac{\alpha_t^2 L}{2}\|z_{t-\frac{1}{2}} - z_{t-1}\|^2,$$

and hence conclude (41) and Lemma D.7. $\qquad\square$

### E.8 Proof of Lemma D.8

*Proof.*[Proof of Lemma D.8] To bound the inner-product terms in (41), by setting $\varphi_1 = z_{t-\frac{1}{2}}$, $\theta = z_{t-1}$, $\varphi_2 = z_t$, $\delta_1 = \eta_t\left( \nabla\tilde{\mathcal{F}}(z_{t-1}^{\mathrm{md}}; \xi_{t-\frac{1}{2}}) + \tilde{\mathcal{H}}(z_{t-1}; \zeta_{t-\frac{1}{2}}) \right)$, $\delta_2 = \eta_t\left( \nabla\tilde{\mathcal{F}}(z_{t-1}^{\mathrm{md}}; \xi_{t-\frac{1}{2}}) + \tilde{\mathcal{H}}(z_{t-\frac{1}{2}}; \zeta_t) \right)$ and $J = \eta_t J$ as in Lemma D.1 (with $\mathbf{z} = \tilde{z}$), we have

$$\eta_t\langle \nabla\tilde{\mathcal{F}}(z_{t-1}^{\mathrm{md}}; \xi_{t-\frac{1}{2}}) + \tilde{\mathcal{H}}(z_{t-\frac{1}{2}}; \zeta_t), z_{t-\frac{1}{2}} - \tilde{z} \rangle + \eta_t\left( J(z_{t-\frac{1}{2}}) - J(\tilde{z}) \right)$$

$$\leq \mathcal{B}(z_{t-1}, \tilde{z}) - \mathcal{B}(\mathbf{x}_t, \tilde{z}) - \mathcal{B}(z_{t-\frac{1}{2}}, z_{t-1}) + \frac{\eta_t^2}{2\mu_{\mathcal{B}}}\|\tilde{\mathcal{H}}(z_{t-\frac{1}{2}}; \zeta_t) - \tilde{\mathcal{H}}(z_{t-1}; \zeta_{t-\frac{1}{2}})\|^2,$$

where Young's inequality combined with the martingale structure yields (also noting (31))

$$\mathbb{E}\|\tilde{\mathcal{H}}(z_{t-\frac{1}{2}}; \zeta_t) - \tilde{\mathcal{H}}(z_{t-1}; \zeta_{t-\frac{1}{2}})\|^2$$

$$= \mathbb{E}\|\mathcal{H}(z_{t-\frac{1}{2}}) - \mathcal{H}(z_{t-1}) - \Delta_{\mathcal{H}}^{t-\frac{1}{2}}\|^2 + \mathbb{E}\left\| \Delta_{\mathcal{H}}^t \right\|^2$$

$$\leq (1 + \beta)M^2\mathbb{E}\|z_{t-\frac{1}{2}} - z_{t-1}\|^2 + (1 + \tfrac{1}{\beta})\mathbb{E}\left\| \Delta_{\mathcal{H}}^{t-\frac{1}{2}} \right\|^2 + \mathbb{E}\left\| \Delta_{\mathcal{H}}^t \right\|^2.$$

Combining the above two displays with expectation taken gives

$$\mathbb{E}\left[\eta_t\langle\nabla\tilde{\mathcal{F}}(\boldsymbol{z}_{t-1}^{\mathrm{md}};\xi_{t-\frac{1}{2}})+\tilde{\mathcal{H}}(\boldsymbol{z}_{t-\frac{1}{2}};\zeta_t),\boldsymbol{z}_{t-\frac{1}{2}}-\tilde{\boldsymbol{z}}\rangle+\eta_t\left(J(\boldsymbol{z}_{t-\frac{1}{2}})-J(\tilde{\boldsymbol{z}})\right)\right]$$

$$\leq\mathbb{E}\left[\mathcal{B}(\boldsymbol{z}_{t-1},\tilde{\boldsymbol{z}})-\mathcal{B}(\mathbf{x}_t,\tilde{\boldsymbol{z}})-\mathcal{B}(\boldsymbol{z}_{t-\frac{1}{2}},\boldsymbol{z}_{t-1})\right]\tag{66}$$

$$+\frac{\eta_t^2}{2\mu_{\mathcal{B}}}\left((1+\beta)M^2\mathbb{E}\|\boldsymbol{z}_{t-\frac{1}{2}}-\boldsymbol{z}_{t-1}\|^2+(1+\tfrac{1}{\beta})\mathbb{E}\left\|\boldsymbol{\Delta}_{\mathcal{H}}^{t-\frac{1}{2}}\right\|^2+\mathbb{E}\left\|\boldsymbol{\Delta}_{\mathcal{H}}^t\right\|^2\right).$$

Applying the inequality $\mathcal{B}(\boldsymbol{z}_{t-\frac{1}{2}},\boldsymbol{z}_{t-1})\geq\frac{\mu_{\mathcal{B}}}{2}\left\|\boldsymbol{z}_{t-\frac{1}{2}}-\boldsymbol{z}_{t-1}\right\|^2$ again gives

$$\mathbb{E}\left[\eta_t\langle\nabla\tilde{\mathcal{F}}(\boldsymbol{z}_{t-1}^{\mathrm{md}};\xi_{t-\frac{1}{2}})+\mathcal{H}(\boldsymbol{z}_{t-\frac{1}{2}};\zeta_t),\boldsymbol{z}_{t-\frac{1}{2}}-\tilde{\boldsymbol{z}}\rangle+\eta_t\left(J(\boldsymbol{z}_{t-\frac{1}{2}})-J(\tilde{\boldsymbol{z}})\right)\right]$$

$$\leq\mathbb{E}\left[\mathcal{B}(\boldsymbol{z}_{t-1},\tilde{\boldsymbol{z}})-\mathcal{B}(\mathbf{x}_t,\tilde{\boldsymbol{z}})\right]$$

$$-\left(\frac{\mu_{\mathcal{B}}}{2}-\frac{\eta_t^2}{2\mu_{\mathcal{B}}}(1+\beta)M^2\right)\mathbb{E}\|\boldsymbol{z}_{t-\frac{1}{2}}-\boldsymbol{z}_{t-1}\|^2+\frac{\eta_t^2}{2\mu_{\mathcal{B}}}\left((1+\tfrac{1}{\beta})\mathbb{E}\left\|\boldsymbol{\Delta}_{\mathcal{H}}^{t-\frac{1}{2}}\right\|^2+\mathbb{E}\left\|\boldsymbol{\Delta}_{\mathcal{H}}^t\right\|^2\right).$$

Further, by definition of the primal-dual gap function and the definition of the noisy terms (31), by taking $\alpha_t=\frac{2}{t+1}$ in (41) of Lemma D.7 and taking expectations on both sides, we have

$$\mathbb{E}[V(\boldsymbol{z}_{t-\frac{1}{2}}^{\mathrm{ag}}\mid\tilde{\boldsymbol{z}})]-\frac{t-1}{t+1}\mathbb{E}[V(\boldsymbol{z}_{t-\frac{3}{2}}^{\mathrm{ag}}\mid\tilde{\boldsymbol{z}})]$$

$$\leq\frac{2}{t+1}\mathbb{E}\langle\nabla\mathcal{F}(\boldsymbol{z}_{t-1}^{\mathrm{md}})+\mathcal{H}(\boldsymbol{z}_{t-\frac{1}{2}}),\boldsymbol{z}_{t-\frac{1}{2}}-\tilde{\boldsymbol{z}}\rangle+\frac{2L}{(t+1)^2}\mathbb{E}\left\|\boldsymbol{z}_{t-\frac{1}{2}}-\boldsymbol{z}_{t-1}\right\|^2$$

$$+\frac{2}{t+1}\mathbb{E}\left(J(\boldsymbol{z}_{t-\frac{1}{2}})-J(\tilde{\boldsymbol{z}})\right)$$

$$=\frac{2}{t+1}\mathbb{E}\langle\nabla\tilde{\mathcal{F}}(\boldsymbol{z}_{t-1}^{\mathrm{md}};\xi_{t-\frac{1}{2}})+\tilde{\mathcal{H}}(\boldsymbol{z}_{t-\frac{1}{2}};\zeta_t),\boldsymbol{z}_{t-\frac{1}{2}}-\tilde{\boldsymbol{z}}\rangle+\frac{2L}{(t+1)^2}\mathbb{E}\left\|\boldsymbol{z}_{t-\frac{1}{2}}-\boldsymbol{z}_{t-1}\right\|^2$$

$$-\frac{2}{t+1}\mathbb{E}\langle\boldsymbol{\Delta}_{\mathcal{F}}^{t-\frac{1}{2}}+\boldsymbol{\Delta}_{\mathcal{H}}^t,\boldsymbol{z}_{t-\frac{1}{2}}-\tilde{\boldsymbol{z}}\rangle+\frac{2}{t+1}\mathbb{E}\left(J(\boldsymbol{z}_{t-\frac{1}{2}})-J(\tilde{\boldsymbol{z}})\right)$$

Bringing in (66) into the above derivation, we obtain

$$\mathbb{E}\left[V\big(\boldsymbol{z}_{t-\frac{1}{2}}^{\mathrm{ag}}\mid\tilde{\boldsymbol{z}}\big)\right]-\frac{t-1}{t+1}\mathbb{E}\left[V\big(\boldsymbol{z}_{t-\frac{3}{2}}^{\mathrm{ag}}\mid\tilde{\boldsymbol{z}}\big)\right]$$

$$\leq\frac{2}{(t+1)\eta_t}\left(\mathbb{E}\left[\mathcal{B}(\boldsymbol{z}_{t-1},\tilde{\boldsymbol{z}})\right]-\mathbb{E}\left[\mathcal{B}(\boldsymbol{z}_t,\tilde{\boldsymbol{z}})\right]\right)$$

$$-\frac{1}{(t+1)\eta_t}\left(\mu_{\mathcal{B}}-\frac{2L}{t+1}\eta_t-\frac{(1+\beta)M^2\eta_t^2}{\mu_{\mathcal{B}}}\right)\mathbb{E}\left\|\boldsymbol{z}_{t-\frac{1}{2}}-\boldsymbol{z}_{t-1}\right\|^2$$

$$+\frac{\eta_t}{(t+1)\mu_{\mathcal{B}}}\left((1+\tfrac{1}{\beta})\mathbb{E}\left\|\boldsymbol{\Delta}_{\mathcal{H}}^{t-\frac{1}{2}}\right\|^2+\mathbb{E}\left\|\boldsymbol{\Delta}_{\mathcal{H}}^t\right\|^2\right)-\frac{2}{t+1}\mathbb{E}\langle\boldsymbol{\Delta}_{\mathcal{F}}^{t-\frac{1}{2}}+\boldsymbol{\Delta}_{\mathcal{H}}^t,\boldsymbol{z}_{t-\frac{1}{2}}-\tilde{\boldsymbol{z}}\rangle.$$

Recalling that we use the choice of $\eta_t$ that satisfies for a given $r\in(0,1)$ that $r\mu_{\mathcal{B}}-\frac{2L}{t+1}\eta_t-\frac{(1+\beta)M^2\eta_t^2}{\mu_{\mathcal{B}}}\geq0$. With some manipulations we obtain

$$\mathbb{E}\left[V\big(\boldsymbol{z}_{t-\frac{1}{2}}^{\mathrm{ag}}\mid\tilde{\boldsymbol{z}}\big)\right]-\frac{t-1}{t+1}\mathbb{E}\left[V\big(\boldsymbol{z}_{t-\frac{3}{2}}^{\mathrm{ag}}\mid\tilde{\boldsymbol{z}}\big)\right]$$

$$\leq\frac{2}{(t+1)\eta_t}\left(\mathbb{E}\left[\mathcal{B}(\boldsymbol{z}_{t-1},\tilde{\boldsymbol{z}})\right]-\mathbb{E}\left[\mathcal{B}(\boldsymbol{z}_t,\tilde{\boldsymbol{z}})\right]\right)$$

$$+\frac{\eta_t}{(t+1)\mu_{\mathcal{B}}}\left((1+\tfrac{1}{\beta})\mathbb{E}\left\|\boldsymbol{\Delta}_{\mathcal{H}}^{t-\frac{1}{2}}\right\|^2+\mathbb{E}\left\|\boldsymbol{\Delta}_{\mathcal{H}}^t\right\|^2\right)-\frac{(1-r)\mu_{\mathcal{B}}}{(t+1)\eta_t}\mathbb{E}\left[\left\|\boldsymbol{z}_{t-\frac{1}{2}}-\boldsymbol{z}_{t-1}\right\|^2\right]$$

$$-\frac{2}{t+1}\mathbb{E}\langle\boldsymbol{\Delta}_{\mathcal{F}}^{t-\frac{1}{2}},\boldsymbol{z}_{t-\frac{1}{2}}-\boldsymbol{z}_{t-1}\rangle-\underbrace{\frac{2}{t+1}\mathbb{E}\langle\boldsymbol{\Delta}_{\mathcal{F}}^{t-\frac{1}{2}},\boldsymbol{z}_{t-1}-\tilde{\boldsymbol{z}}\rangle-\frac{2}{t+1}\mathbb{E}\langle\boldsymbol{\Delta}_{\mathcal{H}}^t,\boldsymbol{z}_{t-\frac{1}{2}}-\tilde{\boldsymbol{z}}\rangle}_{\mathrm{I}}.$$

$$\tag{67}$$

Due to the law of iterated expectation applied to martingale difference conditions $\mathbb{E}\left[\boldsymbol{\Delta}_{\mathcal{F}}^{t-\frac{1}{2}} \mid \mathcal{F}_{t-1}\right] = \mathbf{0}$ and $\mathbb{E}\left[\boldsymbol{\Delta}_{\mathcal{H}}^{t} \mid \mathcal{F}_{t-\frac{1}{2}}\right] = \mathbf{0}$, $i = 1, 2$, we have

$$I = 0.$$

Moreover, for the rest of the terms in (67), we note that there is a basic quadratic inequality that $-\frac{(1-r)\mu_{\mathcal{B}}}{\eta_t}\left\|\boldsymbol{z}_{t-1} - \mathbf{x}_{t-\frac{1}{2}}\right\|^2 - 2\left\langle\boldsymbol{\Delta}_{\mathcal{F}}^{t-\frac{1}{2}}, \boldsymbol{z}_{t-\frac{1}{2}} - \boldsymbol{z}_{t-1}\right\rangle \leq \frac{\eta_t}{(1-r)\mu_{\mathcal{B}}}\left\|\boldsymbol{\Delta}_{\mathcal{F}}^{t-\frac{1}{2}}\right\|^2$. (67) reduces to

$$\mathbb{E}\left[V\left(\boldsymbol{z}_{t-\frac{1}{2}}^{\mathrm{ag}} \mid \tilde{\boldsymbol{z}}\right)\right] - \frac{t-1}{t+1}\mathbb{E}\left[V\left(\boldsymbol{z}_{t-\frac{3}{2}}^{\mathrm{ag}} \mid \tilde{\boldsymbol{z}}\right)\right] \leq \frac{2}{(t+1)\eta_t}\left(\mathbb{E}\left[\mathcal{B}(\boldsymbol{z}_{t-1}, \tilde{\boldsymbol{z}})\right] - \mathbb{E}\left[\mathcal{B}(\boldsymbol{z}_t, \tilde{\boldsymbol{z}})\right]\right)$$
$$+ \frac{\eta_t}{(t+1)\mu_{\mathcal{B}}}\left((1+\tfrac{1}{\beta})\mathbb{E}\left\|\boldsymbol{\Delta}_{\mathcal{H}}^{t-\frac{1}{2}}\right\|^2 + \mathbb{E}\left\|\boldsymbol{\Delta}_{\mathcal{H}}^{t}\right\|^2\right) + \frac{\eta_t}{(1-r)(t+1)\mu_{\mathcal{B}}}\mathbb{E}\left\|\boldsymbol{\Delta}_{\mathcal{F}}^{t-\frac{1}{2}}\right\|^2. \quad (68)$$

Multiplying both sides of (68) by $t(t+1)$, we obtain for all $t = 1, \ldots, T$

$$t(t+1)\mathbb{E}\left[V\left(\boldsymbol{z}_{t-\frac{1}{2}}^{\mathrm{ag}} \mid \tilde{\boldsymbol{z}}\right)\right] - (t-1)t\mathbb{E}\left[V\left(\boldsymbol{z}_{t-\frac{3}{2}}^{\mathrm{ag}} \mid \tilde{\boldsymbol{z}}\right)\right]$$
$$\leq \frac{2t}{\eta_t}\left(\mathbb{E}\left[\mathcal{B}(\boldsymbol{z}_{t-1}, \tilde{\boldsymbol{z}})\right] - \mathbb{E}\left[\mathcal{B}(\boldsymbol{z}_t, \tilde{\boldsymbol{z}})\right]\right) + \frac{t\eta_t}{\mu_{\mathcal{B}}}\left(\tfrac{1}{1-r}\mathbb{E}\left\|\boldsymbol{\Delta}_{\mathcal{F}}^{t-\frac{1}{2}}\right\|^2 + (1+\tfrac{1}{\beta})\mathbb{E}\left\|\boldsymbol{\Delta}_{\mathcal{H}}^{t-\frac{1}{2}}\right\|^2 + \mathbb{E}\left\|\boldsymbol{\Delta}_{\mathcal{H}}^{t}\right\|^2\right)$$
$$\leq \frac{2t}{\eta_t}\left(\mathbb{E}\left[\mathcal{B}(\boldsymbol{z}_{t-1}, \tilde{\boldsymbol{z}})\right] - \mathbb{E}\left[\mathcal{B}(\boldsymbol{z}_t, \tilde{\boldsymbol{z}})\right]\right) + \left(\tfrac{1}{1-r}\sigma_{\mathrm{Str}}^2 + (2+\tfrac{1}{\beta})\sigma_{\mathrm{Bil}}^2\right)\frac{t\eta_t}{\mu_{\mathcal{B}}},$$

where in the last line above we applied Assumption 2.2, so by law of iterated expectations

$$\mathbb{E}\left\|\boldsymbol{\Delta}_{\mathcal{F}}^{t-\frac{1}{2}}\right\|^2 = \mathbb{E}\left[\left\|\nabla\tilde{F}(\boldsymbol{z}_{t-1}^{\mathrm{md}}; \xi_{t-\frac{1}{2}}) - \nabla F(\boldsymbol{z}_{t-1}^{\mathrm{md}})\right\|^2\right] \leq \sigma_{\mathrm{Str}}^2,$$
$$\mathbb{E}\left\|\boldsymbol{\Delta}_{\mathcal{H}}^{t-\frac{1}{2}}\right\|^2 = \mathbb{E}\left[\left\|\tilde{H}(\boldsymbol{z}_{t-1}; \zeta_{t-\frac{1}{2}}) - H(\boldsymbol{z}_{t-1})\right\|^2\right] \leq \sigma_{\mathrm{Bil}}^2, \quad (69)$$
$$\mathbb{E}\left\|\boldsymbol{\Delta}_{\mathcal{H}}^{t}\right\|^2 = \mathbb{E}\left[\left\|\tilde{H}(\boldsymbol{z}_{t-\frac{1}{2}}; \zeta_t) - H(\boldsymbol{z}_{t-\frac{1}{2}})\right\|^2\right] \leq \sigma_{\mathrm{Bil}}^2.$$