# OpenReview forum: "Optimal Extragradient-Based Algorithms for Stochastic Variational Inequalities with Separable Structure"
_NeurIPS.cc/2023/Conference — NeurIPS 2023 poster_

### Official Review · Reviewer_KTFh · 2023-06-29

**Soundness:** 4 excellent
**Presentation:** 3 good
**Contribution:** 2 fair
**Rating:** 6
**Confidence:** 4

**Summary:**

This paper studies the stochastic monotone variational inequality problem. They propose the Accelerated Gradient - Extragradient (AG-EG) algorithm which is shown to have the optimal convergence rates for the strongly monotone VI problem.

**Strengths:**

This papers studies the important problem of Stochastic VIs, the algorithm, especially the idea of restarting is interesting.

**Weaknesses:**

Dependence on the Diameter of the Set: A major differentiating point of this paper, as claimed by the authors in Remark 2.4, is that the convergence rate bounds have no dependence on the constraint set. However, there is a dependence on the initial distance of the iterate to the solution. Showing that the iterates are bounded is essentially the only requirement to convert the results of papers like [Chen et a. 2017] to the unconstrained setting. Can the authors describe the main difficulties and technical novelties involved in showing this?

Rate dependence on the noise term: As stated in the appendix, the optimal dependence on the noise term has been achieved by a multistage algorithm in [Fallah et al. 2020], where the  dependence on the noise term is $\mathcal{O}(\sigma^2/n)$, whereas this paper seems to have a dependence of the form  $\mathcal{O}(\sigma^2/\sqrt{n})$. Am I missing something? Is the optimality of the proposed algorithm only in terms of the first term, where the dependence on the problem parameters $L, \mu$ etc. are optimal?

**Questions:**

See above

**Limitations:**

See above

---

> ### Author Rebuttal · Authors · 2023-08-10
>
> Thank you for your constructive feedback. We address your comments and questions as follows.
>
> ---
>
> **Q1**: Showing that the iterates are bounded is essentially the only requirement to convert the results of papers like Chen et al. [2017] to the unconstrained setting. Can the authors describe the main difficulties and technical novelties involved in showing this?
>
> **A1**:  We appreciate your question. While the optimal rate for monotone variational inequalities (VIs) has been achieved by Chen et al. [2017], achieving the optimal rate for strongly-monotone VIs remains largely unsolved. The boundedness of iterates has been one of the main barriers to proving the optimal rate for strongly-monotone VIs.
>
> The standard gradient descent-ascent method is known to *diverge* for many convex-concave problems [1]. Moreover, in the noisy setting, *even extragradient may diverge* [2]. Thus it is critical to show that the iterates are bounded. It is a challenge as many seminal works on stochastic VIs only consider bounded domains [3,4]. In the VI literature, showing the boundedness of the iterates is often the crux of the proof (see for instance, [5]).
>
> The main proof technique used in our work is the “bootstrapping” argument shown in step 3 of Sec. D.3. By summing up Eq. (33), a rearrangement of the terms implies a relation of $\mathbb{E}||z_t - z^*||^2$ with a summation of itself, the result follows by the bootstrapping argument. In sharp contrast, Chen et al. [2017] adopt "the enlargement of a maximal monotone operator" technique to deal with unbounded domains, which seems complicated and unnecessary for our purpose. Our bootstrapping method leads to a simpler and more concise analysis. Moreover, Chen et al. [2017] only considered the monotone VI setting rather than the strongly-monotone VI setting.
>
> In addition, we also consider specific instances of VI problems (i.e., bilinear games, bilinearly coupled SC-SC) with lower-bound matching results by utilizing scaling reduction techniques. These results were not obtained by Chen et al. [2017] either.
>
> [1] Mescheder, Lars, Andreas Geiger, and Sebastian Nowozin. "Which training methods for GANs do actually converge?." International conference on machine learning. PMLR, 2018.
>
> [2] Chavdarova, Tatjana, et al. "Reducing noise in GAN training with variance reduced extragradient." Advances in Neural Information Processing Systems 32 (2019).
>
> [3] Nemirovski, Arkadi, et al. "Robust stochastic approximation approach to stochastic programming." SIAM Journal on optimization 19.4 (2009): 1574-1609.
>
> [4] Juditsky, Anatoli, Arkadi Nemirovski, and Claire Tauvel. "Solving variational inequalities with stochastic mirror-prox algorithm." Stochastic Systems 1.1 (2011): 17-58.
>
> [5] Gorbunov, Eduard, et al. "Clipped stochastic methods for variational inequalities with heavy-tailed noise." NeurIPS 2022.
>
> ---
>
> **Q2**: The optimal dependence on the noise term has been achieved by a multistage algorithm in Fallah et al. [2020], where the dependence on the noise term is $\mathcal{O}\left(\sigma^2 / n\right)$, whereas this paper seems to have a dependence of the form $\mathcal{O}\left(\sigma^2 / \sqrt{n}\right)$. Is the optimality of the proposed algorithm only in terms of the first term, where the dependence on the problem parameters
> $L, \mu$ are optimal?
>
> **A2**: We believe there is a misunderstanding by the reviewer. This is due to the different definitions of the $\varepsilon$-optimal point. In our paper, the $\varepsilon$-optimal point is defined by $||z - z^*|| \leq \varepsilon$. In contrast, Fallah et al. [2020] define the optimal point by $||z - z^*||^2 \leq \varepsilon$. Thus, our $\varepsilon^2$ is equivalent to their $\varepsilon$. Therefore, the complexity term that depends on the noise variance $\sigma^2$ in Corollary 2.8, i.e., $\frac{\sigma^2}{\mu^2 \varepsilon^2}$ should be translated into $\frac{\sigma^2}{\mu^2 n}$ rather than $\frac{\sigma^2}{\mu^2 \sqrt{n}}$. This also applies to Corollaries 2.9, 3.1, and 3.3.
> To sum up, the optimality of our proposed algorithm is not only in terms of the first term but also in the second noise term. We apologize for the confusion caused by the different definition of $\varepsilon$-optimal point, and will make it clear in the revision.

---

> > ### Comment · Reviewer_KTFh · 2023-08-13
> > **Reply to Rebuttal**
> >
> > I thank the authors for their rebuttal. I have increased my score.

---

> > > ### Author Response · Authors · 2023-08-13
> > >
> > > Thank you for considering our rebuttal and increasing the score. Your feedback is greatly appreciated.

---

### Official Review · Reviewer_oFAu · 2023-07-06

**Soundness:** 3 good
**Presentation:** 3 good
**Contribution:** 3 good
**Rating:** 6
**Confidence:** 2

**Summary:**

The author(s) studied the extragradient algorithm for the separable strongly monotone VI problems. It was shown that the new analysis can achieve optimal error bounds in various settings.

**Strengths:**

- The new analysis gives optimal error bounds in various settings, which is a decent contribution to the field.
- The paper is well-written and easy to follow in general.

**Weaknesses:**

- I encourage the author(s) to include a table to compare the error bounds and assumptions used with prior works. It seems that the error bound obtained by the author(s) is not entirely new, existing analysis can also achieve the same error bound under other conditions.
- There is no related work section, which makes it hard for readers to understand the position of this work in the literature.

**Questions:**

- I suggest the author(s) to include a table of error bounds and assumptions to compare with existing works.

**Limitations:**

I do not find any negative societal impact of this paper.

---

> ### Author Rebuttal · Authors · 2023-08-10
>
> Thank you for your positive feedback. We address your comments and questions as follows.
>
> ---
>
> **Q1**: I suggest the author(s) to include a table of error bounds and assumptions to compare with existing works.
>
> **A1**: Thank you for your suggestion. We have added a revised table comparing error bounds and assumptions with existing works in our uploaded pdf file.
>
> ---
>
> **Q2**: There is no related work section, which makes it hard for readers to understand the position of this work in the literature.
>
> **A2**: Due to the space limit in the submission, we have deferred the related work section to the supplementary material. We will restructure our content to accommodate a comprehensive related work section in the main content.

---

> > ### Comment · Reviewer_oFAu · 2023-08-15
> > **Thanks for your rebuttal**
> >
> > Thanks for adding the table, the contribution becomes more clear to me (as an non-expert in this field). I am keeping my score unchanged.

---

> > > ### Author Response · Authors · 2023-08-15
> > >
> > > Thank you for your positive feedback! We will be sure to integrate the table and all other revisions as promised into the final version.

---

### Official Review · Reviewer_qR4K · 2023-07-24

**Soundness:** 4 excellent
**Presentation:** 3 good
**Contribution:** 3 good
**Rating:** 6
**Confidence:** 3

**Summary:**

The paper studies variational inequalities with separable structures (sum of the gradient of a strongly convex function and a monotone operator). This class subsumes bilinear coupling SC-SC minimax optimization and bilinear games. The authors propose an extragradient algorithm with acceleration by shifting the convexity part from the gradient to the operator. This algorithm matches the lower bounds for strongly monotone VIs (up to some log factors). When specialized in bilinear problems, the algorithm can be coupled with scheduled restarting to match the existing lower bounds for bilinear SC-SC and bilinear games.

**Strengths:**

- The proposed algorithm matches several lower bounds at the same time.
- The algorithms do not require the bounded domain assumption as in some of the previous works.
- The technical details of the paper are excellent and easy to follow. I checked some of the proofs in the appendix and did not find any issues.
- The discussion provided in the paper is generally well written.

**Weaknesses:**

- Scheduled restart may not always be desirable, although this is not necessarily a weak point. For bilinear problems, the algorithm does need scheduled restart to achieve the optimality. The claim in the conclusion that all three lower bounds are matched in one algorithm is therefore not really accurate.

**Questions:**

1. In Jordan et al (2023) the separable structure means sum of the gradient of a general convex function and a strongly monotone operator. Could the authors discuss the difference between this and the one considered in the paper? How would shifting the convexity technique change?
2. Some existing papers talk about optimal bounds with the attention to the log factors. I am not sure to what extent the authors pointed this out in the related work and the paper. Could the authors discuss?

**Limitations:**

The authors discussed some limitations of the proposed algorithm.

---

> ### Author Rebuttal · Authors · 2023-08-10
>
> Thank you for your positive feedback. We address your comments and questions as follows.
>
> ---
>
> **Q1**: For bilinear problems, the algorithm does need scheduled restart to achieve optimality. The claim in the conclusion that all three lower bounds are matched in one algorithm is, therefore not really accurate.
>
> **A1**: We appreciate your feedback and the opportunity to clarify our claim. First, our algorithm with scheduled restarting can achieve optimal results for general variational inequality, bilinear games, and bilinearly coupled strongly-convex-strongly concave (SC-SC) settings. Please refer to the table in the uploaded pdf file for the corresponding upper and lower bounds. So our claim that all three lower bounds are matched in one algorithm in the conclusion is valid.
>
> In addition, we agree that the scheduled restarting technique may not be ideal. For bilinear game problems, techniques such as scheduled restarting is essential in matching the lower bound.  Alternative approaches exist, such as the work by Azizian et al. [2020b], which uses EG with momentum. However, the accelerated rate in their work is limited to scenarios with large condition numbers. It remains an open question whether a technique can simultaneously accelerate the bilinear part and the minimization part without scheduled restarting. This question is left for future work.
>
> ---
>
> **Q2**: In Jordan et al. [2023] the separable structure means sum of the gradient of a general convex function and a strongly monotone operator.
> Could the authors discuss the difference between this and the one considered in the paper?
>
> **A2**: Thanks for your question. Jordan et al. [2023] uses separable structure to refer to the sum of the gradients of a general convex function and a strongly monotone operator, while we use separable structure to refer to the sum of the gradients of a strongly convex function and a general monotone operator.  These two settings are equivalent because we can shift the “strong convexity'' component from the strongly convex function to the monotone operator.
>
> ---
>
> **Q3**: Some existing papers talk about optimal bounds with the attention to the log factors.
>
> **A3**: Our upper bounds and existing lower bounds all contain log factors. For example, in Corollary 2.8, Eq. (17), Corollary 3.1, and Corollary 3.3, the complexity is presented with log factors being explicitly considered. In Table 1 in our original supplementary material, and the table in our uploaded pdf file, we have omitted dependency on log factors due to layout reasons. We have added discussions on the log factors in the caption of our table in the uploaded pdf file.

---

> > ### Comment · Reviewer_qR4K · 2023-08-12
> >
> > I thank the authors for their thoughtful rebuttal. I maintain my current score.

---

> > > ### Author Response · Authors · 2023-08-13
> > >
> > > We greatly appreciate your positive feedback. We're glad to know that our response has effectively addressed your questions.

---

### Official Review · Reviewer_iB8j · 2023-07-25

**Soundness:** 3 good
**Presentation:** 3 good
**Contribution:** 3 good
**Rating:** 6
**Confidence:** 3

**Summary:**

The work presents a stochastic accelerated gradient-extra gradient (AG-EG) algorithm for strongly monotone variational inequalities (VI) that is designed by combining extra gradient and Nesterov acceleration. The major of the work is extending the formulation to the case when the constraint set is convex but can be unbounded and still match the best-known convergence result in the literature. Two variants of the AG-EG algorithm are presented by the authors, a direct approach and another with scheduled restarting. The convergence rates also match the lower bounds for the special case of VI's in bilinearly coupled strongly convex strongly concave saddle point problems and bilinear games with AG-EG Scheduled Restarting.

**Strengths:**

The optimal convergence rate is established for strongly monotone VI's.

The paper is written well for most parts except for a few issues which are listed in the weakness section.



**Weaknesses:**


- The notation in algorithm 1 is confusing for the update equations.

- The work does not demonstrate the efficacy of the approach through empirical studies on an application eg. Reinforcemt Learning, Regularized empirical risk minimization, quadratic games, etc.

**Questions:**

-  In section 2 a simplified deterministic setting is shown although the final results are presented for the stochastic case. This included the Assumption 1 as well. It leads to confusion about the assumptions, is it in expectation or holds for each random realization of function?

- The authors may add a table comparing the complexity results with the SOTA algorithms for solving stochastic VI problems for better readability and presentation.

---

> ### Author Rebuttal · Authors · 2023-08-09
>
> Thank you for your positive feedback. We address your comments and questions as follows.
>
> ---
>
> **Q1**: The notation in algorithm 1 is confusing for the update equations
>
> **A1**: In Algorithm 1, we introduce notation to distinguish different points in the AGEG method. Specifically, $z_{t-\frac{1}{2}}$ refers to the extrapolated point, while $z^{\text{md}}$ and $z^{\text{ag}}$ denote the "middle" and "aggregated" points, respectively. In addition, we use notations $\zeta_{t-\frac{1}{2}}$, $\zeta_{t}$, and $\xi_{t - \frac{1}{2}}$ to account for noise during various steps. With our notation, $z_x$ depends on $\zeta_{y\leq x}$ and $\xi_{y \leq x}$ but remains independent of $\zeta_{y > x}$ and $\xi_{y > x}$. We provide definitions of $f(x; \xi), h(x, y, \zeta)$, and $g(y; \xi)$ in Sec. 1 as stochastic oracles of $F(x), H(x, y)$, and $G(y)$, respectively.
>
> ---
>
> **Q2**: The work does not demonstrate the efficacy of the approach through empirical studies on an application
>
> **A2**: During the rebuttal period, we conducted a comparison between our stochastic AGEG algorithms (with restarting) and stochastic extragradient (SEG) algorithms (with restarting) on quadratic games. Our experiments focused on synthetic quadratic games with varying parameter settings and noise scales. Detailed results have been provided in our uploaded pdf file. We can see that stochastic AGEG with restarting outperforms SEG with restarting by a large margin. Due to the time limit, we will add more empirical results in our final version.
>
> ---
>
> **Q3**: Section 2 presents a simplified deterministic setting, although the final results are given for the stochastic case. This included the Assumption 1 as well. It leads to confusion about the assumptions, is it in expectation or holds for each random realization of function?
>
> **A3**: Throughout this paper, our notation $\mathcal{F}$ and $\mathcal{H}$ represents functions in their expected forms. Consequently, Assumption 2.1 holds in expectation.

---

> > ### Comment · Reviewer_iB8j · 2023-08-15
> >
> > Thanks for clarifying the notation and the additional empirical results. I am keeping my score unchanged.

---

### Author Rebuttal · Authors · 2023-08-10

Dear reviewers,

Thank you for your time and constructive comments. Based on the feedback of Reviewer iB8j and Reviewer oFAu, we have added a table comparing the complexity results and assumptions with the prior works. In addition, to address the feedback of Reviewer iB8j on empirical studies, we conducted an experiment to examine the performance of stochastic AGEG vs stochastic EG. Please find the table and figure in the uploaded PDF file. For all other comments and feedbacks, we have provided point-by-point responses in the rebuttal separately.

---

### Decision · Program_Chairs · 2023-09-21

**Decision:**

Accept (poster)

**Comment:**

The author response addressed the reviewers' main concerns. The reviewers all agree that the paper makes solid contributions to this line of work, and it provides optimal results in several settings that remained open.